# Trans-Dimensional Generative Modeling via Jump Diffusion Models

**Andrew Campbell**[1]      **William Harvey**[2]      **Christian Weilbach**[2]

**Valentin De Bortoli**[3]      **Tom Rainforth**[1]      **Arnaud Doucet**[1]

[1]Department of Statistics, University of Oxford, UK
[2] Department of Computer Science, University of British Columbia, Vancouver, Canada
[3]CNRS ENS Ulm, Paris, France
{campbell, rainforth, doucet}@stats.ox.ac.uk
{wsgh, weilbach}@cs.ubc.ca
valentin.debortoli@gmail.com

## Abstract

We propose a new class of generative models that naturally handle data of varying dimensionality by jointly modeling the state and dimension of each datapoint. The generative process is formulated as a jump diffusion process that makes jumps between different dimensional spaces. We first define a dimension destroying forward noising process, before deriving the dimension creating time-reversed generative process along with a novel evidence lower bound training objective for learning to approximate it. Simulating our learned approximation to the time-reversed generative process then provides an effective way of sampling data of varying dimensionality by jointly generating state values and dimensions. We demonstrate our approach on molecular and video datasets of varying dimensionality, reporting better compatibility with test-time diffusion guidance imputation tasks and improved interpolation capabilities versus fixed dimensional models that generate state values and dimensions separately.

## 1 Introduction

Generative models based on diffusion processes [1–3] have become widely used in solving a range of problems including text-to-image generation [4, 5], audio synthesis [6] and protein design [7]. These models define a forward noising diffusion process that corrupts data to noise and then learn the corresponding time-reversed backward generative process that generates novel datapoints from noise.

In many applications, for example generating novel molecules [8] or videos [9, 10], the dimension of the data can also vary. For example, a molecule can contain a varying number of atoms and a video can contain a varying number of frames. When defining a generative model over these data-types, it is therefore necessary to model the number of dimensions along with the raw values of each of its dimensions (the state). Previous approaches to modeling such data have relied on first sampling the number of dimensions from the empirical distribution obtained from the training data, and then sampling data using a fixed dimension diffusion model (FDDM) conditioned on this number of dimensions [8]. For conditional modeling, where the number of dimensions may depend on the observations, this approach does not apply and we are forced to first train an auxiliary model that predicts the number of dimensions given the observations [11].

37th Conference on Neural Information Processing Systems (NeurIPS 2023).

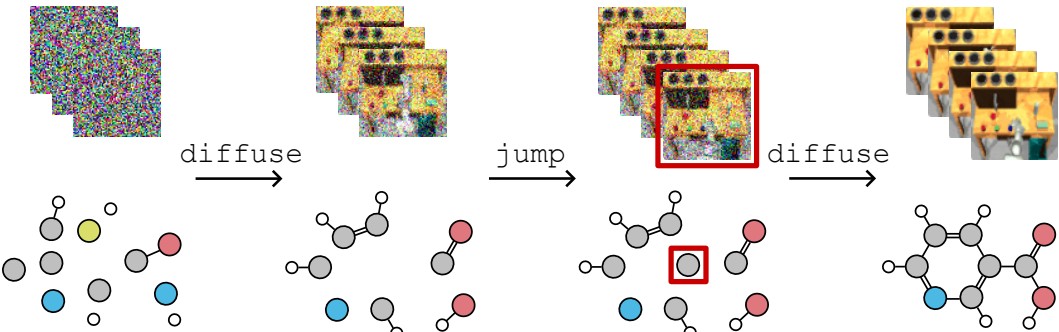

Figure 1: Illustration of the jump diffusion generative process on videos and molecules. The generative process consists of two parts: a diffusion part which denoises the current set of frames/atoms and a jump part which adds on a suitable number of new frames/atoms such that the final generation is a clean synthetic datapoint of an appropriate size.

This approach to trans-dimensional generative modeling is fundamentally limited due to the complete separation of dimension generation and state value generation. This is exemplified in the common use case of conditional diffusion guidance. Here, an unconditional diffusion model is trained that end-users can then easily and cheaply condition on their task of interest through guiding the generative diffusion process [3, 12–14] without needing to perform any further training or fine-tuning of the model on their task of interest. Since the diffusion occurs in a fixed dimensional space, there is no way for the guidance to appropriately guide the dimension of the generated datapoint. This can lead to incorrect generations for datasets where the dimension greatly affects the nature of the datapoint created, e.g. small molecules have completely different properties to large molecules.

To generate data of varying dimensionality, we propose a jump diffusion based generative model that jointly generates both the dimension and the state. Our model can be seen as a unification of diffusion models which generate all dimensions in parallel with autoregressive type models which generate dimensions sequentially. We derive the model through constructing a forward noising process that adds noise and removes dimensions and a backward generative process that denoises and adds dimensions, see Figure 1. We derive the optimum backward generative process as the time-reversal of the forward noising process and derive a novel learning objective to learn this backward process from data. We demonstrate the advantages of our method on molecular and video datasets finding our method achieves superior guided generation performance and produces more representative data interpolations across dimensions.

## 2  Background

Standard continuous-time diffusion models [3, 15–17] define a forward diffusion process through a stochastic differential equation (SDE) where $\mathbf{x}_0 \sim p_{\text{data}}$ and, for $t > 0$,

$$\mathrm{d}\mathbf{x}_t = \overrightarrow{\mathbf{b}}_t(\mathbf{x}_t)\mathrm{d}t + g_t\mathrm{d}\mathbf{w}_t, \tag{1}$$

where $\mathbf{x}_t \in \mathbb{R}^d$ is the current state, $\overrightarrow{\mathbf{b}}_t : \mathbb{R}^d \to \mathbb{R}^d$ is the drift and $g_t \in \mathbb{R}$ is the diffusion coefficient. $\mathrm{d}\mathbf{w}_t$ is a Brownian motion increment on $\mathbb{R}^d$. This SDE can be understood intuitively by noting that in each infinitesimal timestep, we move slightly in the direction of the drift $\overrightarrow{\mathbf{b}}_t$ and inject a small amount of Gaussian noised governed by $g_t$. Let $p_t(\mathbf{x}_t)$ denote the distribution of $\mathbf{x}_t$ for the forward diffusion process (1) so that $p_0(\mathbf{x}_0) = p_{\text{data}}(\mathbf{x}_0)$. $\overrightarrow{\mathbf{b}}_t$ and $g_t$ are set such that at time $t = T$, $p_T(\mathbf{x}_T)$ is close to $p_{\text{ref}}(\mathbf{x}_T) = \mathcal{N}(\mathbf{x}_T; 0, I_d)$; e.g. $\overrightarrow{\mathbf{b}}_t(\mathbf{x}_t) = -\frac{1}{2}\beta_t\mathbf{x}_t$, $g_t = \sqrt{\beta_t}$ for $\beta_t > 0$ [2, 3].

The time-reversal of the forward diffusion (1) is also a diffusion [18, 19] which runs backwards in time from $p_T(\mathbf{x}_T)$ to $p_0(\mathbf{x}_0)$ and satisfies the following reverse time SDE

$$\mathrm{d}\mathbf{x}_t = \overleftarrow{\mathbf{b}}_t(\mathbf{x}_t)\mathrm{d}t + g_t\mathrm{d}\hat{\mathbf{w}}_t,$$

where $\overleftarrow{\mathbf{b}}_t(\mathbf{x}_t) = \overrightarrow{\mathbf{b}}_t(\mathbf{x}_t) - g_t^2\nabla_{\mathbf{x}_t}\log p_t(\mathbf{x}_t)$, $\mathrm{d}t$ is a negative infinitesimal time step and $\mathrm{d}\hat{\mathbf{w}}_t$ is a Brownian motion increment when time flows backwards. Unfortunately, both the terminal

distribution, $p_T(\mathbf{x}_T)$, and the score, $\nabla_{\mathbf{x}_t} \log p_t(\mathbf{x}_t)$, are unknown in practice. A generative model is obtained by approximating $p_T$ with $p_{\text{ref}}$ and learning an approximation $s_t^\theta(\mathbf{x}_t)$ to $\nabla_{\mathbf{x}_t} \log p_t(\mathbf{x}_t)$ typically using denoising score matching [20], i.e.

$$\min_\theta \quad \mathbb{E}_{\mathcal{U}(t;0,T)p_{0,t}(\mathbf{x}_0,\mathbf{x}_t)}[\|s_t^\theta(\mathbf{x}_t) - \nabla_{\mathbf{x}_t} \log p_{t|0}(\mathbf{x}_t|\mathbf{x}_0)\|^2]. \tag{2}$$

For a flexible model class, $s^\theta$, we get $s_t^\theta(\mathbf{x}_t) \approx \nabla_{\mathbf{x}_t} \log p_t(\mathbf{x}_t)$ at the minimizing parameter.

# 3 Trans-Dimensional Generative Model

Instead of working with fixed dimension datapoints, we will instead assume our datapoints consist of a variable number of components. A datapoint $\mathbf{X}$ consists of $n$ components each of dimension $d$. For ease of notation, each datapoint will explicitly store both the number of components, $n$, and the state values, $\mathbf{x}$, giving $\mathbf{X} = (n, \mathbf{x})$. Since each datapoint can have a variable number of components from $n = 1$ to $n = N$, our overall space that our datapoints live in is the union of all these possibilities, $\mathbf{X} \in \mathcal{X} = \bigcup_{n=1}^N \{n\} \times \mathbb{R}^{nd}$. For example, for a varying size point cloud dataset, components would refer to points in the cloud, each containing $(x, y, z)$ coordinates giving $d = 3$ and the maximum possible number of points in the cloud is $N$.

Broadly speaking, our approach will follow the same framework as previous diffusion generative models. We will first define a forward noising process that both corrupts state values with Gaussian noise and progressively deletes dimensions. We then learn an approximation to the time-reversal giving a backward generative process that simultaneously denoises whilst also progressively adding dimensions back until a synthetic datapoint of appropriate dimensionality has been constructed.

## 3.1 Forward Process

Our forward and backward processes will be defined through jump diffusions. A jump diffusion process has two components, the diffusion part and the jump part. Between jumps, the process evolves according to a standard SDE. When a jump occurs, the process transitions to a different dimensional space with the new value for the process being drawn from a transition kernel $K_t(\mathbf{Y}|\mathbf{X}) : \mathcal{X} \times \mathcal{X} \to \mathbb{R}_{\geq 0}$. Letting $\mathbf{Y} = (m, \mathbf{y})$, the transition kernel satisfies $\sum_m \int_{\mathbf{y}} K_t(m, \mathbf{y}|\mathbf{X})d\mathbf{y} = 1$ and $\int_{\mathbf{y}} K_t(m = n, \mathbf{y}|\mathbf{X})d\mathbf{y} = 0$. The rate at which jumps occur (jumps per unit time) is given by a rate function $\lambda_t(\mathbf{X}) : \mathcal{X} \to \mathbb{R}_{\geq 0}$. For an infinitesimal timestep $\mathrm{d}t$, the jump diffusion can be written as

$$\textbf{Jump} \qquad \mathbf{X}_t' = \begin{cases} \mathbf{X}_t & \text{with probability } 1 - \lambda_t(\mathbf{X}_t)\mathrm{d}t \\ \mathbf{Y} \sim K_t(\mathbf{Y}|\mathbf{X}_t) & \text{with probability } \lambda_t(\mathbf{X}_t)\mathrm{d}t \end{cases}$$

$$\textbf{Diffusion} \qquad \mathbf{x}_{t+\mathrm{d}t} = \mathbf{x}_t' + \mathbf{b}_t(\mathbf{X}_t')\mathrm{d}t + g_t\mathrm{d}\mathbf{w}_t \qquad n_{t+\mathrm{d}t} = n_t'$$

with $\mathbf{X}_t \triangleq (n_t, \mathbf{x}_t)$ and $\mathbf{X}_{t+\mathrm{d}t} \triangleq (n_{t+\mathrm{d}t}, \mathbf{x}_{t+\mathrm{d}t})$ and $\mathrm{d}\mathbf{w}_t$ being a Brownian motion increment on $\mathbb{R}^{n_t'd}$. We provide a more formal definition in Appendix A.

With the jump diffusion formalism in hand, we can now construct our forward noising process. We will use the diffusion part to corrupt existing state values with Gaussian noise and the jump part to destroy dimensions. For the diffusion part, we use the VP-SDE introduced in [2, 3] with $\overrightarrow{\mathbf{b}}_t(\mathbf{X}) = -\frac{1}{2}\beta_t\mathbf{x}$ and $\overrightarrow{g}_t = \sqrt{\beta_t}$ with $\beta_t \geq 0$.

When a jump occurs in the forward process, one component of the current state will be deleted. For example, one point in a point cloud or a single frame in a video is deleted. The rate at which these deletions occur is set by a user-defined forward rate $\overrightarrow{\lambda}_t(\mathbf{X})$. To formalize the deletion, we need to introduce some more notation. We let $K^{\text{del}}(i|n)$ be a user-defined distribution over which component of the current state to delete. We also define del $: \mathcal{X} \times \mathbb{N} \to \mathcal{X}$ to be the deletion operator that deletes a specified component. Specifically, $(n-1, \mathbf{y}) = \text{del}((n, \mathbf{x}), i)$ where $\mathbf{y} \in \mathbb{R}^{(n-1)d}$ has the same values as $\mathbf{x} \in \mathbb{R}^{nd}$ except for the $d$ values corresponding to the $i$th component which have been removed. We can now define the forward jump transition kernel as $\overrightarrow{K}_t(\mathbf{Y}|\mathbf{X}) = \sum_{i=1}^n K^{\text{del}}(i|n)\delta_{\text{del}(\mathbf{X},i)}(\mathbf{Y})$. We note that only one component is ever deleted at a time meaning $\overrightarrow{K}_t(m, \mathbf{y}|\mathbf{X}) = 0$ for $m \neq n - 1$. Further, the choice of $K^{\text{del}}(i|n)$ will dictate the behaviour of the reverse generative process. If we set $K^{\text{del}}(i|n) = \mathbb{I}\{i = n\}$ then we only ever delete the final component and so in the reverse generative

Table 1: Summary of forward and parameterized backward processes

| Direction | $\mathbf{b}_t$ | $g_t$ | $\lambda_t(\mathbf{X})$ | $K_t(\mathbf{Y}|\mathbf{X})$ |
|---|---|---|---|---|
| **Forward** | $-\frac{1}{2}\beta_t\mathbf{x}$ | $\sqrt{\beta_t}$ | $\overrightarrow{\lambda}_t(n)$ | $\sum_{i=1}^{n} K^{\mathrm{del}}(i|n)\delta_{\mathrm{del}(\mathbf{X},i)}(\mathbf{Y})$ |
| **Backward** | $-\frac{1}{2}\beta_t\mathbf{x} - \beta_t s_t^\theta(\mathbf{X})$ | $\sqrt{\beta_t}$ | $\overleftarrow{\lambda}_t^\theta(\mathbf{X})$ | $\int_{\mathbf{y}^{\mathrm{add}}} \sum_{i=1}^{n+1} A_t^\theta(\mathbf{y}^{\mathrm{add}},i|\mathbf{X})\delta_{\mathrm{ins}(\mathbf{X},\mathbf{y}^{\mathrm{add}},i)}(\mathbf{Y})\mathrm{d}\mathbf{y}^{\mathrm{add}}$ |

direction, datapoints are created additively, appending components onto the end of the current state. Alternatively, if we set $K^{\mathrm{del}}(i|n) = 1/n$ then components are deleted uniformly at random during forward corruption and in the reverse generative process, the model will need to pick the most suitable location for a new component from all possible positions.

The forward noising process is simulated from $t = 0$ to $t = T$ and should be such that at time $t = T$, the marginal probability $p_t(\mathbf{X})$ should be close to a reference measure $p_{\mathrm{ref}}(\mathbf{X})$ that can be sampled from. We set $p_{\mathrm{ref}}(\mathbf{X}) = \mathbb{I}\{n = 1\}\mathcal{N}(\mathbf{x}; 0, I_d)$ where $\mathbb{I}\{n = 1\}$ is 1 when $n = 1$ and 0 otherwise. To be close to $p_{\mathrm{ref}}$, for the jump part, we set $\overrightarrow{\lambda}_t$ high enough such that at time $t = T$ there is a high probability that all but one of the components in the original datapoint have been deleted. For simplicity, we also set $\overrightarrow{\lambda}_t$ to depend only on the current dimension $\overrightarrow{\lambda}_t(\mathbf{X}) = \overrightarrow{\lambda}_t(n)$ with $\overrightarrow{\lambda}_t(n = 1) = 0$ so that the forward process stops deleting components when there is only 1 left. In our experiments, we demonstrate the trade-offs between different rate schedules in time. For the diffusion part, we use the standard diffusion $\beta_t$ schedule [2, 3] so that we are close to $\mathcal{N}(\mathbf{x}; 0, I_d)$.

## 3.2 Backward Process

The backward generative process will simultaneously denoise and add dimensions back in order to construct the final datapoint. It will consist of a backward drift $\overleftarrow{\mathbf{b}}_t(\mathbf{X})$, diffusion coefficient $\overleftarrow{g}_t$, rate $\overleftarrow{\lambda}_t(\mathbf{X})$ and transition kernel $\overleftarrow{K}_t(\mathbf{Y}|\mathbf{X})$. We would like these quantities to be such that the backward process is the time-reversal of the forward process. In order to find the time-reversal of the forward process, we must first introduce some notation to describe $\overleftarrow{K}_t(\mathbf{Y}|\mathbf{X})$. $\overleftarrow{K}_t(\mathbf{Y}|\mathbf{X})$ should undo the forward deletion operation. Since $\overrightarrow{K}_t(\mathbf{Y}|\mathbf{X})$ chooses a component and then deletes it, $\overleftarrow{K}_t(\mathbf{Y}|\mathbf{X})$ will need to generate the state values for a new component, decide where the component should be placed and then insert it at this location. Our new component will be denoted $\mathbf{y}^{\mathrm{add}} \in \mathbb{R}^d$. The insertion operator is defined as ins : $\mathcal{X} \times \mathbb{R}^d \times \mathbb{N} \to \mathcal{X}$. It takes in the current value $\mathbf{X}$, the new component $\mathbf{y}^{\mathrm{add}}$ and an index $i \in \{1, \ldots, n + 1\}$ and inserts $\mathbf{y}^{\mathrm{add}}$ into $\mathbf{X}$ at location $i$ such that the resulting value $\mathbf{Y} = \mathrm{ins}(\mathbf{X}, \mathbf{y}^{\mathrm{add}}, i)$ has $\mathrm{del}(\mathbf{Y}, i) = \mathbf{X}$. We denote the joint conditional distribution over the newly added component and the index at which it is inserted as $A_t(\mathbf{y}^{\mathrm{add}}, i|\mathbf{X})$. We therefore have $\overleftarrow{K}_t(\mathbf{Y}|\mathbf{X}) = \int_{\mathbf{y}^{\mathrm{add}}} \sum_{i=1}^{n+1} A_t(\mathbf{y}^{\mathrm{add}}, i|\mathbf{X})\delta_{\mathrm{ins}(\mathbf{X}, \mathbf{y}^{\mathrm{add}}, i)}(\mathbf{Y})\mathrm{d}\mathbf{y}^{\mathrm{add}}$. Noting that only one component is ever added at a time, we have $\overleftarrow{K}_t(m, \mathbf{y}|\mathbf{X}) = 0$ for $m \neq n + 1$.

This backward process formalism can be seen as a unification of diffusion models with autoregressive models. The diffusion part $\overleftarrow{\mathbf{b}}_t$ denoises the current set of components in parallel, whilst the autoregressive part $A_t(\mathbf{y}^{\mathrm{add}}, i|\mathbf{X})$ predicts a new component and its location. $\overleftarrow{\lambda}_t(\mathbf{X})$ is the glue between these parts controlling when and how many new components are added during generation.

We now give the optimum values for $\overleftarrow{\mathbf{b}}_t(\mathbf{X})$, $\overleftarrow{g}_t$, $\overleftarrow{\lambda}_t(\mathbf{X})$ and $A_t(\mathbf{y}^{\mathrm{add}}, i|\mathbf{X})$ such that the backward process is the time-reversal of the forward process.

**Proposition 1.** *The time reversal of a forward jump diffusion process given by drift $\overrightarrow{\mathbf{b}}_t$, diffusion coefficient $\overrightarrow{g}_t$, rate $\overrightarrow{\lambda}_t(n)$ and transition kernel $\sum_{i=1}^{n} K^{\mathrm{del}}(i|n)\delta_{\mathrm{del}(\mathbf{X},i)}(\mathbf{Y})$ is given by a jump diffusion process with drift $\overleftarrow{\mathbf{b}}_t^*(\mathbf{X})$, diffusion coefficient $\overleftarrow{g}_t^*$, rate $\overleftarrow{\lambda}_t^*(\mathbf{X})$ and transition kernel*

$\int_{\mathbf{y}^{\text{add}}} \sum_{i=1}^{n+1} A_t^*(\mathbf{y}^{\text{add}}, i|\mathbf{X})\delta_{\text{ins}(\mathbf{X},\mathbf{y}^{\text{add}},i)}(\mathbf{Y})\mathrm{d}\mathbf{y}^{\text{add}}$ *as defined below*

$$\overleftarrow{\mathbf{b}}_t^*(\mathbf{X}) = \overrightarrow{\mathbf{b}}_t(\mathbf{X}) - \overrightarrow{g}_t^2 \nabla_{\mathbf{x}} \log p_t(\mathbf{X}), \quad \overleftarrow{g}_t^* = \overrightarrow{g}_t,$$

$$\overleftarrow{\lambda}_t^*(\mathbf{X}) = \overrightarrow{\lambda}_t(n+1)\frac{\sum_{i=1}^{n+1} K^{\text{del}}(i|n+1) \int_{\mathbf{y}^{\text{add}}} p_t(\text{ins}(\mathbf{X}, \mathbf{y}^{\text{add}}, i))\mathrm{d}\mathbf{y}^{\text{add}}}{p_t(\mathbf{X})},$$

$$A_t^*(\mathbf{y}^{\text{add}}, i|\mathbf{X}) \propto p_t(\text{ins}(\mathbf{X}, \mathbf{y}^{\text{add}}, i))K^{\text{del}}(i|n+1).$$

All proofs are given in Appendix A. The expressions for $\overleftarrow{\mathbf{b}}_t^*$ and $\overleftarrow{g}_t^*$ are the same as for a standard diffusion except for replacing $\nabla_{\mathbf{x}} \log p_t(\mathbf{x})$ with $\nabla_{\mathbf{x}} \log p_t(\mathbf{X}) = \nabla_{\mathbf{x}} \log p_t(\mathbf{x}|n)$ which is simply the score in the current dimension. The expression for $\overleftarrow{\lambda}_t^*$ can be understood intuitively by noting that the numerator in the probability ratio is the probability that at time $t$, given a deletion occurs, the forward process will arrive at $\mathbf{X}$. If this is higher than the raw probability at time $t$ that the forward process is at $\mathbf{X}$ (the denominator) then we should have high $\overleftarrow{\lambda}_t^*$ because $\mathbf{X}$ is likely the result of a deletion of a larger datapoint. Finally the optimum $A_t^*(\mathbf{y}^{\text{add}}, i|\mathbf{X})$ is simply the conditional distribution of $\mathbf{y}^{\text{add}}$ and $i$ given $\mathbf{X}$ when the joint distribution over $\mathbf{y}^{\text{add}}, i, \mathbf{X}$ is given by $p_t(\text{ins}(\mathbf{X}, \mathbf{y}^{\text{add}}, i))K^{\text{del}}(i|n+1)$.

### 3.3 Objective for Learning the Backward Process

The true $\overleftarrow{\mathbf{b}}_t^*$, $\overleftarrow{\lambda}_t^*$ and $A_t^*$ are unknown so we need to learn approximations to them, $\overleftarrow{\mathbf{b}}_t^\theta$, $\overleftarrow{\lambda}_t^\theta$ and $A_t^\theta$. Following Proposition 1, we set $\overleftarrow{\mathbf{b}}_t^\theta(\mathbf{X}) = \overrightarrow{\mathbf{b}}_t(\mathbf{X}) - \overrightarrow{g}_t^2 s_t^\theta(\mathbf{X})$ where $s_t^\theta(\mathbf{X})$ approximates $\nabla_{\mathbf{x}} \log p_t(\mathbf{X})$. The forward and parameterized backward processes are summarized in Table 1.

Standard diffusion models are trained using a denoising score matching loss which can be derived from maximizing an evidence lower bound on the model probability for $\mathbb{E}_{p_{\text{data}}(\mathbf{x}_0)}[\log p_0^\theta(\mathbf{x}_0)]$ [21]. We derive here an equivalent loss to learn $s_t^\theta$, $\overleftarrow{\lambda}_t^\theta$ and $A_t^\theta$ for our jump diffusion process by leveraging the results of [17] and [22]. Before presenting this loss, we first introduce some notation. Our objective for $s_t^\theta(\mathbf{X}_t)$ will resemble denoising score matching (2) but instead involve the conditional score $\nabla_{\mathbf{x}_t} \log p_{t|0}(\mathbf{X}_t|\mathbf{X}_0) = \nabla_{\mathbf{x}_t} \log p_{t|0}(\mathbf{x}_t|\mathbf{X}_0, n_t)$. This is difficult to calculate directly due to a combinatorial sum over the different ways the components of $\mathbf{X}_0$ can be deleted to get to $\mathbf{X}_t$. We avoid this problem by equivalently conditioning on a mask variable $M_t \in \{0,1\}^{n_0}$ that is 0 for components of $\mathbf{X}_0$ that have been deleted to get to $\mathbf{X}_t$ and 1 for components that remain in $\mathbf{X}_t$. This makes our denoising score matching target easy to calculate: $\nabla_{\mathbf{x}_t} \log p_{t|0}(\mathbf{x}_t|\mathbf{X}_0, n_t, M_t) = \frac{\sqrt{\alpha_t}M_t(\mathbf{x}_0) - \mathbf{x}_t}{1 - \alpha_t}$ where $\alpha_t = \exp(-\int_0^t \beta(s)\mathrm{d}s)$ [3]. Here $M_t(\mathbf{x}_0)$ is the vector removing any components in $\mathbf{x}_0$ for which $M_t$ is 0, thus $M_t(\mathbf{x}_0)$ and $\mathbf{x}_t$ have the same dimensionality. We now state our full objective.

**Proposition 2.** *For the backward generative jump diffusion process starting at $p_{\text{ref}}(\mathbf{X}_T)$ and finishing at $p_0^\theta(\mathbf{X}_0)$, an evidence lower bound on the model log-likelihood $\mathbb{E}_{\mathbf{x}_0 \sim p_{\text{data}}}[\log p_0^\theta(\mathbf{x}_0)]$ is given by*

$$\mathcal{L}(\theta) = -\frac{T}{2}\mathbb{E}\Big[g_t^2 \|s_t^\theta(\mathbf{X}_t) - \nabla_{\mathbf{x}_t} \log p_{t|0}(\mathbf{x}_t|\mathbf{X}_0, n_t, M_t)\|^2\Big] + \tag{3}$$

$$T\mathbb{E}\Big[-\overleftarrow{\lambda}_t^\theta(\mathbf{X}_t) + \overrightarrow{\lambda}_t(n_t) \log \overleftarrow{\lambda}_t^\theta(\mathbf{Y}) + \overrightarrow{\lambda}_t(n_t) \log A_t^\theta(\mathbf{x}_t^{\text{add}}, i|\mathbf{Y})\Big] + C, \tag{4}$$

*where expectations are with respect to $\mathcal{U}(t; 0, T)p_{0,t}(\mathbf{X}_0, \mathbf{X}_t, M_t)K^{\text{del}}(i|n_t)\delta_{\text{del}(\mathbf{X}_t, i)}(\mathbf{Y})$, $C$ is a constant term independent of $\theta$ and $\mathbf{X}_t = \text{ins}(\mathbf{Y}, \mathbf{x}_t^{\text{add}}, i)$. This evidence lower bound is equal to the log-likelihood when $\overleftarrow{\mathbf{b}}_t^\theta = \overleftarrow{\mathbf{b}}_t^*$, $\overleftarrow{\lambda}_t^\theta = \overleftarrow{\lambda}_t^*$ and $A_t^\theta = A_t^*$.*

We now examine the objective to gain an intuition into the learning signal. Our first term (3) is an $L_2$ regression to a target that, as we have seen, is a scaled vector between $\mathbf{x}_t$ and $\sqrt{\alpha_t}M_t(\mathbf{x}_0)$. As the solution to an $L_2$ regression problem is the conditional expectation of the target, $s_t^\theta(\mathbf{X}_t)$ will learn to predict vectors pointing towards $\mathbf{x}_0$ averaged over the possible correspondences between dimensions of $\mathbf{x}_t$ and dimensions of $\mathbf{x}_0$. Thus, during sampling, $s_t^\theta(\mathbf{X}_t)$ provides a suitable direction to adjust the current value $\mathbf{X}_t$ taking into account the fact $\mathbf{X}_t$ represents only a noisy subpart of a clean whole $\mathbf{X}_0$.

The second term (4) gives a learning signal for $\overleftarrow{\lambda}_t^\theta$ and $A_t^\theta$. For $A_t^\theta$, we simply have a maximum likelihood objective, predicting the missing part of $\mathbf{X}_t$ (i.e. $\mathbf{x}_t^{\text{add}}$) given the observed part of $\mathbf{X}_t$ (i.e. $\mathbf{Y}$). The signal for $\overleftarrow{\lambda}_t^\theta$ comes from balancing two terms: $-\overleftarrow{\lambda}_t^\theta(\mathbf{X}_t)$ and $\overrightarrow{\lambda}_t(n_t) \log \overleftarrow{\lambda}_t^\theta(\mathbf{Y})$

which encourage the value of $\overleftarrow{\lambda}_t^\theta$ to move in opposite directions. For a new test input $\mathbf{Z}$, $\overleftarrow{\lambda}_t^\theta(\mathbf{Z})$'s value needs to trade off between the two terms by learning the relative probability between $\mathbf{Z}$ being the entirety of a genuine sample from the forward process, corresponding to the $\overleftarrow{\lambda}_t^\theta(\mathbf{X}_t)$ term in (4), or $\mathbf{Z}$ being a substructure of a genuine sample, corresponding to the $\overleftarrow{\lambda}_t^\theta(\mathbf{Y})$ term in (4). The optimum trade-off is found exactly at the time reversal $\overleftarrow{\lambda}_t^*$ as we show in Appendix A.5.

We optimize $\mathcal{L}(\theta)$ using stochastic gradient ascent, generating minibatches by first sampling $t \sim \mathcal{U}(0, T)$, $\mathbf{X}_0 \sim p_{\text{data}}$ and then computing $\mathbf{X}_t$ from the forward process. This can be done analytically for the $\overrightarrow{\lambda}_t(n)$ functions used in our experiments. We first sample $n_t$ by analytic integration of the dimension deletion Poisson process with time inhomogeneous rate $\overrightarrow{\lambda}_t(n)$. We then add Gaussian noise independently to each dimension under $p_{t|0}(\mathbf{x}_t|\mathbf{X}_0, n_t, M_t)$ using a randomly drawn mask variable $M_t$. See Appendix B for further details on the efficient evaluation of our objective.

### 3.4 Parameterization

$s_t^\theta(\mathbf{X}_t)$, $A_t^\theta(\mathbf{y}^{\text{add}}, i|\mathbf{X}_t)$ and $\overleftarrow{\lambda}_t^\theta(\mathbf{X}_t)$ will all be parameterized by neural networks. In practice, we have a single backbone network suited to the problem of interest e.g. a Transformer [23], an EGNN [24] or a UNet [25] onto which we add prediction heads for $s_t^\theta(\mathbf{X}_t)$, $A_t^\theta(\mathbf{y}^{\text{add}}, i|\mathbf{X}_t)$ and $\overleftarrow{\lambda}_t^\theta(\mathbf{X}_t)$. $s_t^\theta(\mathbf{X}_t)$ outputs a vector in $\mathbb{R}^{n_t d}$. $A_t^\theta(\mathbf{y}^{\text{add}}, i|\mathbf{X}_t)$ outputs a distribution over $i$ and mean and standard deviation statistics for a Gaussian distribution over $\mathbf{y}^{\text{add}}$. Finally, having $\overleftarrow{\lambda}_t^\theta(\mathbf{X}_t) \in \mathbb{R}_{\geq 0}$ be the raw output of a neural network can cause optimization issues due to the optimum $\overleftarrow{\lambda}_t^*$ including a probability ratio which can take on very large values. Instead, we learn a component prediction network $p_{0|t}^\theta(n_0|\mathbf{X}_t)$ that predicts the number of components in $\mathbf{X}_0$ given $\mathbf{X}_t$. To convert this into $\overleftarrow{\lambda}^\theta(\mathbf{X}_t)$, we show in Proposition 3 how the optimum $\overleftarrow{\lambda}_t^*(\mathbf{X}_t)$ is an analytic function of the true $p_{0|t}(n_0|\mathbf{X}_t)$. We then plug $p_{0|t}^\theta(n_0|\mathbf{X}_t)$ into Proposition 3 to obtain an approximation of $\overleftarrow{\lambda}_t^*(\mathbf{X}_t)$.

**Proposition 3.** *We have*

$$\overleftarrow{\lambda}_t^*(\mathbf{X}_t) = \overrightarrow{\lambda}_t(n_t + 1) \sum_{n_0=1}^{N} \frac{p_{t|0}(n_t + 1|n_0)}{p_{t|0}(n_t|n_0)} p_{0|t}(n_0|\mathbf{X}_t),$$

*where $\mathbf{X}_t = (n_t, \mathbf{x}_t)$ and $p_{t|0}(n_t + 1|n_0)$ and $p_{t|0}(n_t|n_0)$ are both easily calculable distributions from the forward dimension deletion process.*

### 3.5 Sampling

To sample the generative process, we numerically integrate the learned backward jump diffusion process using time-step $\delta t$. Intuitively, it is simply the standard continuous time diffusion sampling scheme [3] but at each timestep we check whether a jump has occurred and if it has, sample the new component and insert it at the chosen index as explained by Algorithm 1.

---

**Algorithm 1:** Sampling the Generative Process

$t \leftarrow T$
$\mathbf{X} \sim p_{\text{ref}}(\mathbf{X}) = \mathbb{I}\{n = 1\}\mathcal{N}(\mathbf{x}; 0, I_d)$
**while** $t > 0$ **do**
    **if** $u < \overleftarrow{\lambda}_t^\theta(\mathbf{X})\delta t$ with $u \sim \mathcal{U}(0, 1)$ **then**
        Sample $\mathbf{x}^{\text{add}}, i \sim A_t^\theta(\mathbf{x}^{\text{add}}, i|\mathbf{X})$
        $\mathbf{X} \leftarrow \text{ins}(\mathbf{X}, \mathbf{x}^{\text{add}}, i)$
    **end**
    $\mathbf{x} \leftarrow \mathbf{x} - \overleftarrow{\mathbf{b}}_t^\theta(\mathbf{X})\delta t + g_t\sqrt{\delta t}\epsilon$ with $\epsilon \sim \mathcal{N}(0, I_{nd})$
    $\mathbf{X} \leftarrow (n, \mathbf{x}), t \leftarrow t - \delta t$
**end**

---

## 4 Related Work

Our method jointly generates both dimensions and state values during the generative process whereas prior approaches [8, 11] are forced to first sample the number of dimensions and then run the diffusion process in this fixed dimension. When diffusion guidance is applied to these unconditional models [14, 26], users need to pick by hand the number of dimensions independent of the conditioning information even though the number of dimensions can be correlated with the conditioning parameter.

Instead of automatically learning when and how many dimensions to add during the generative process, previous work focusing on images [27, 28] hand pick dimension jump points such that the

resolution of images is increased during sampling and reaches a certain pre-defined desired resolution at the end of the generative process. Further, rather than using any equivalent of $A_t^\theta$, the values for new dimensions are simply filled in with Gaussian noise. These approaches mainly focus on efficiency rather than flexible generation as we do here.

The first term in our learning objective in Proposition 2 corresponds to learning the continuous part of our process (the diffusion) and the second corresponds to learning the discrete part of our process (the jumps). The first term can be seen as a trans-dimensional extension of standard denoising score matching [20] whilst the second bears similarity to the discrete space ELBO derived in [29].

Finally, jump diffusions also have a long history of use in Bayesian inference, where one aims to draw samples from a trans-dimensional target posterior distribution based on an unnormalized version of its density [30]: an ergodic jump diffusion is designed which admits the target as the invariant distribution [30–32]. The invariant distribution is not preserved when time-discretizing the process. However, it was shown in [33, 34] how general jump proposals could be built and how this process could be "Metropolized" to obtain a discrete-time Markov process admitting the correct invariant distribution, yielding the popular Reversible Jump Markov Chain Monte Carlo algorithm. Our setup differs significantly as we only have access to samples in the form of data, not an unnormalized target.

# 5 Experiments

## 5.1 Molecules

We now show how our model provides significant benefits for diffusion guidance and interpolation tasks. We model the QM9 dataset [35, 36] of 100K varying size molecules. Following [8], we consider each molecule as a 3-dimensional point cloud of atoms, each atom having the features: $(x, y, z)$ coordinates, a one-hot encoded atom type, and an integer charge value. Bonds are inferred from inter-atomic distances. We use an EGNN [24] backbone with three heads to predict $s_t^\theta$, $p_{0|t}^\theta(n_0|\mathbf{X}_t)$, and $A_t^\theta$. We uniformly delete dimensions, $K^{\mathrm{del}}(i|n) = 1/n$, and since a point cloud is permutation invariant, $A_t^\theta(\mathbf{y}^{\mathrm{add}}|\mathbf{X}_t)$ need only predict new dimension values. We set $\overrightarrow{\lambda}_t$ to a constant except for $t < 0.1T$, where we set $\overrightarrow{\lambda}_{t<0.1T} = 0$. This ensures that all dimensions are added with enough generation time remaining for the diffusion process to finalize all state values.

We visualize sampling from our learned generative process in Figure 2; note how the process jointly creates a suitable number of atoms whilst adjusting their positions and identities. Before moving on to apply diffusion guidance which is the focus of our experiments, we first verify our unconditional sample quality in Table 2 and find we perform comparably to the results reported in [8] which use an FDDM. We ablate our choice of $\overrightarrow{\lambda}_t$ by comparing with setting $\overrightarrow{\lambda}_t$ to a constant for all $t$ and with setting $\overrightarrow{\lambda}_t = 0$ for $t < 0.9T$ (rather than just for $t < 0.1T$). We find that the constant $\overrightarrow{\lambda}_t$ performs worse due to the occasional component being added late in the generation process without enough time for the diffusion process to finalize its value. We find the $\overrightarrow{\lambda}_{t<0.9T} = 0$ setting to have satisfactory sample quality however this choice of $\overrightarrow{\lambda}_t$ introduces issues during diffusion guided generation as we see next. Finally, we ablate the parameterization of Proposition 3 by learning $\overleftarrow{\lambda}_t^\theta(\mathbf{X}_t) \in \mathbb{R}$ directly as the output of a neural network head. We find that this reduces sample quality due to the more well-behaved nature of the target, $p_{0|t}^\theta(n_0|\mathbf{X}_t)$ when using Proposition 3. We note pure autoregressive models perform significantly worse than diffusion based models as found in [8].

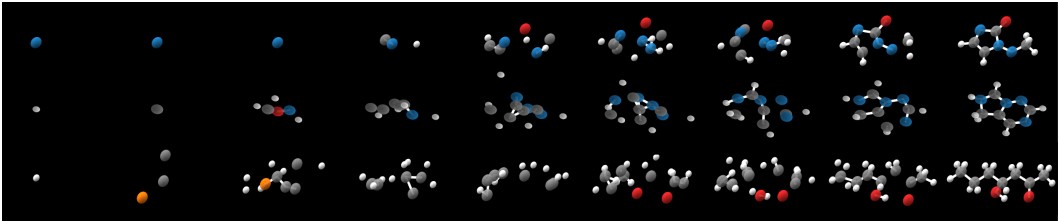

Figure 2: Visualization of the jump-diffusion backward generative process on molecules.

Table 2: Sample quality metrics for unconditional molecule generation. An atom is stable if it has the correct valency whilst a molecule is considered stable if all of its atoms are stable. Molecular validity is measured using RDKit [37]. All methods use 1000 simulation steps and draw 10000 samples.

| Method | % Atom Stable ($\uparrow$) | % Molecule Stable ($\uparrow$) | % Valid ($\uparrow$) |
|---|---|---|---|
| FDDM [8] | **98.7** | 82.0 | 91.9 |
| TDDM (ours) | 98.3 | **87.2** | **92.3** |
| TDDM, const $\overrightarrow{\lambda}_t$ | 96.7 | 79.1 | 86.7 |
| TDDM, $\overrightarrow{\lambda}_{t<0.9T} = 0$ | 97.7 | 82.6 | 89.4 |
| TDDM w/o Prop. 3 | 97.0 | 66.9 | 87.1 |

Table 3: Conditional Molecule Generation for 10 conditioning tasks that each result in a different dimension distribution. We report dimension error as the average Hellinger distance between the generated and ground truth dimension distributions for that property as well as average sample quality metrics. Standard deviations are given across the 10 conditioning tasks. We report in bold values that are statistically indistinguishable from the best result at the 5% level using a two-sided Wilcoxon signed rank test across the 10 conditioning tasks.

| Method | Dimension Error ($\downarrow$) | % Atom Stable ($\uparrow$) | % Molecule Stable ($\uparrow$) | % Valid ($\uparrow$) |
|---|---|---|---|---|
| FDDM | $0.511_{\pm 0.19}$ | $93.5_{\pm 1.1}$ | $31.3_{\pm 6.3}$ | $65.2_{\pm 10.3}$ |
| TDDM | $\mathbf{0.134_{\pm 0.076}}$ | $93.5_{\pm 2.6}$ | $\mathbf{59.1_{\pm 11}}$ | $\mathbf{74.8_{\pm 9.3}}$ |
| TDDM, const $\overrightarrow{\lambda}_t$ | $0.226_{\pm 0.17}$ | $88.9_{\pm 4.8}$ | $43.6_{\pm 15}$ | $63.4_{\pm 14}$ |
| TDDM, $\overrightarrow{\lambda}_{t<0.9T} = 0$ | $0.390_{\pm 0.38}$ | $\mathbf{95.0_{\pm 2.1}}$ | $\mathbf{61.7_{\pm 17}}$ | $\mathbf{77.8_{\pm 13}}$ |
| TDDM w/o Prop. 3 | $0.219_{\pm 0.12}$ | $\mathbf{93.8_{\pm 3.2}}$ | $55.0_{\pm 19}$ | $73.8_{\pm 13}$ |

### 5.1.1 Trans-Dimensional Diffusion Guidance

We now apply diffusion guidance to our unconditional model in order to generate molecules that contain a certain number of desired atom types, e.g. 3 carbons or 1 oxygen and 2 nitrogens. The distribution of molecule sizes changes depending on these conditions. We generate molecules conditioned on these properties by using the reconstruction guided sampling approach introduced in [9]. This method augments the score $s_t^\theta(\mathbf{X}_t)$ such that it approximates $\nabla_{\mathbf{x}_t} \log p_t(\mathbf{X}_t|y)$ rather than $\nabla_{\mathbf{x}_t} \log p_t(\mathbf{X}_t)$ (where $y$ is the conditioning information) by adding on a term approximating $\nabla_{\mathbf{x}_t} \log p_t(y|\mathbf{X}_t)$ with $p_t(y|\mathbf{X}_t) = \sum_{n_0} \int_{\mathbf{x}_0} p(y|\mathbf{X}_0)p_{0|t}(\mathbf{X}_0|\mathbf{X}_t)\mathrm{d}\mathbf{x}_0$. This guides $\mathbf{x}_t$ such that it is consistent with $y$. Since $\lambda_t^\theta(\mathbf{X}_t)$ has access to $\mathbf{x}_t$, it will cause $n_t$ to automatically also be consistent with $y$ without the user needing to input any information on how the conditioning information relates to the size of the datapoints. We give further details on diffusion guidance in Appendix C.

We show our results in Table 3. In order to perform guidance on the FDDM baseline, we implement the model from [8] in continuous time and initialize the dimension from the empirically observed dimension distribution in the dataset. This accounts for the case of an end user attempting to guide a unconditional model with access to no further information. We find that TDDM produces samples whose dimensions much more accurately reflect the true conditional distribution of dimensions given the conditioning information. The $\overrightarrow{\lambda}_{t<0.9T} = 0$ ablation on the other hand only marginally improves the dimension error over FDDM because all dimensions are added in the generative process at a time when $\mathbf{X}_t$ is noisy and has little relation to the conditioning information. This highlights the necessity of allowing dimensions to be added throughout the generative process to gain the trans-dimensional diffusion guidance ability. The ablation with constant $\overrightarrow{\lambda}_t$ has increased dimension error over TDDM as we find that when $\overrightarrow{\lambda}_t > 0$ for all $t$, $\overleftarrow{\lambda}_t^\theta$ can become very large when $t$ is close to 0 when the model has perceived a lack of dimensions. This occasionally results in too many dimensions being added hence an increased dimension error. Not using the Proposition 3 parameterization also increases dimension error due to the increased difficulty in learning $\overleftarrow{\lambda}_t^\theta$.

### 5.1.2 Trans-Dimensional Interpolation

Interpolations are a unique way of gaining insights into the effect of some conditioning parameter on a dataset of interest. To create an interpolation, a conditional generative model is first trained and then sampled with a sweep of the conditioning parameter but using fixed random noise [8]. The resulting series of synthetic datapoints share similar features due to the fixed random noise but vary in ways that are very informative as to the effect of the conditioning parameter. Attempting to interpolate with an FDDM is fundamentally limited because the entire interpolation occurs in the same dimension which is unrealistic when the conditioning parameter is heavily correlated with the dimension of the datapoint. We demonstrate this by following the setup of [8] who train a conditional FDDM conditioned on polarizability. Polariz-

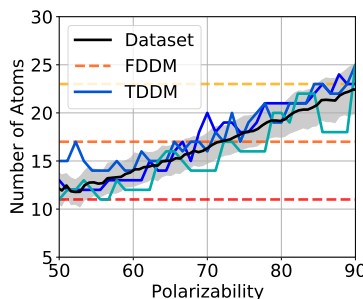

Figure 3: Number of atoms versus polarizability for 3 interpolations with fixed random noise. The dataset mean and standard deviation for the number of atoms is also shown. FDDM interpolates entirely in a fixed dimensional space hence the number of atoms is fixed for all polarizabilities.

ability is the ability of a molecule's electron cloud to distort in response to an external electric field [38] with larger molecules tending to have higher polarizability. To enable us to perform a trans-dimensional interpolation, we also train a conditional version of our model conditioned on polarizability. An example interpolation with this model is shown in Figure 4. We find that indeed the size of the molecule increases with increasing polarizability, with some molecular substructures e.g. rings, being maintained across dimensions. We show how the dimension changes with polarizability during 3 interpolations in Figure 3. We find that these match the true dataset statistics much more accurately than interpolations using FDDM which first pick a dimension and carry out the entire interpolation in that fixed dimension.

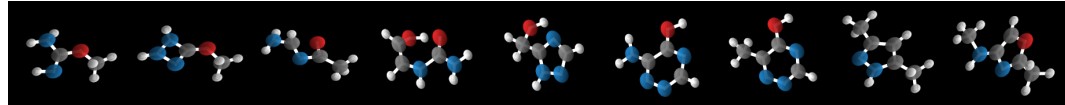

Figure 4: Sequence of generations for linearly increasing polarizability from $39\,\mathrm{Bohr}^3$ to $66\,\mathrm{Bohr}^3$ with fixed random noise. Note how molecular size generally increases with polarizability and how some molecular substructures are maintained between sequential generations of differing dimension. For example, between molecules 6 and 7, the single change is a nitrogen (blue) to a carbon (gray) and an extra hydrogen (white) is added to maintain the correct valency.

### 5.2 Video

We finally demonstrate our model on a video modeling task. Specifically we model the RoboDesk dataset [39], a video benchmark to measure the applicability of video models for planning and control problems. The videos are renderings of a robotic arm [40] performing a variety of different tasks including opening drawers and moving objects. We first train an unconditional model on videos of varying length and then perform planning by applying diffusion guidance to generate videos conditioned on an initial starting frame and a final goal frame [41]. The planning problem is then reduced to "filling in" the frames in between. Our trans-dimensional model automatically varies the number of in-filled frames during generation so that the final length of video matches the length of time the task should take, whereas the fixed dimension model relies on the unrealistic assumption that the length of time the task should take is known before generation.

We model videos at $32 \times 32$ resolution and with varying length from 2 to 35 frames. For the network backbone, we use a UNet adapted for video [42]. In contrast to molecular point clouds, our data is no longer permutation invariant hence $A_t^\theta(\mathbf{y}^{\mathrm{add}}, i|\mathbf{X}_t)$ includes a prediction over the location to insert the new frame. Full experimental details are provided in Appendix D. We evaluate our approach on three planning tasks, holding stationary, sliding a door and pushing an object. An example generation conditioned on the first and last frame for the slide door task is shown in Figure 5, with

the model in-filling a plausible trajectory. We quantify our model's ability to generate videos of a length appropriate to the task in Table 4 finding on all three tasks we generate a more accurate length of video than FDDM which is forced to sample video lengths from the unconditional empirically observed length distribution in the training dataset.

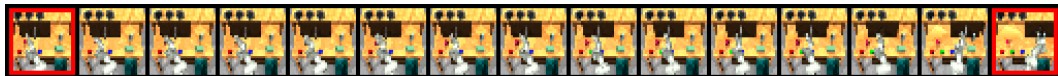

Figure 5: A sample for the slide door task conditioned on the first and last frame (highlighted).

Table 4: Dimension prediction mean absolute error for three planning tasks with standard deviations estimated over 45 samples.

| Method | Stationary ($\downarrow$) | Slide Door ($\downarrow$) | Push Object ($\downarrow$) | Average ($\downarrow$) |
|---|---|---|---|---|
| FDDM | $14.16_{\pm 1.41}$ | $13.39_{\pm 1.34}$ | $17.06_{\pm 1.47}$ | 14.87 |
| TDDM | $\mathbf{9.70_{\pm 0.99}}$ | $\mathbf{11.47_{\pm 0.74}}$ | $\mathbf{15.43_{\pm 0.90}}$ | $\mathbf{12.2}$ |

# 6  Discussion

In this work, we highlighted the pitfalls of performing generative modeling on varying dimensional datasets when treating state values and dimensions completely separately. We instead proposed a trans-dimensional generative model that generates both state values and dimensions jointly during the generative process. We detailed how this process can be formalized with the time-reversal of a jump diffusion and derived a novel evidence lower bound training objective for learning the generative process from data. In our experiments, we found our trans-dimensional model to provide significantly better dimension generation performance for diffusion guidance and interpolations when conditioning on properties that are heavily correlated with the dimension of a datapoint. We believe our approach can further enable generative models to be applied in a wider variety of domains where previous restrictive fixed dimension assumptions have been unsuitable.

# 7  Acknowledgements

The authors are grateful to Martin Buttenschoen for helpful discussions. AC acknowledges support from the EPSRC CDT in Modern Statistics and Statistical Machine Learning (EP/S023151/1). AD acknowledges support of the UK Dstl and EPSRC grant EP/R013616/1. This is part of the collaboration between US DOD, UK MOD and UK EPSRC under the Multidisciplinary University Research Initiative. He also acknowledges support from the EPSRC grants CoSines (EP/R034710/1) and Bayes4Health (EP/R018561/1). WH and CW acknowledge the support of the Natural Sciences and Engineering Research Council of Canada (NSERC), the Canada CIFAR AI Chairs Program. This material is based upon work supported by the United States Air Force Research Laboratory (AFRL) under the Defense Advanced Research Projects Agency (DARPA) Data Driven Discovery Models (D3M) program (Contract No. FA8750-19-2-0222) and Learning with Less Labels (LwLL) program (Contract No.FA8750-19-C-0515). Additional support was provided by UBC's Composites Research Network (CRN), Data Science Institute (DSI) and Support for Teams to Advance Interdisciplinary Research (STAIR) Grants. This research was enabled in part by technical support and computational resources provided by WestGrid (`https://www.westgrid.ca/`) and Compute Canada (`www.computecanada.ca`). The authors would like to acknowledge the use of the University of Oxford Advanced Research Computing (ARC) facility in carrying out this work. `http://dx.doi.org/10.5281/zenodo.22558`

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

# Appendix

This appendix is organized as follows. In Section A, we present proofs for all of our propositions. Section A.1 presents a rigorous definition of our forward process using a more specific notation. This is then used in Section A.2.1 to prove the time reversal for our jump diffusions. We also present an intuitive proof of the time reversal using notation from the main text in Section A.2.2. In Section A.3 we prove Proposition 2 using the notation from the main text. We prove Proposition 3 in Section A.4 and we analyse the optimum of our objective directly without using stochastic process theory in Section A.5. In Section B we give more details on our objective and in Section C we detail how we apply diffusion guidance to our model. We give the full details for our experiments in Section D and finally, in Section E, we discuss the broader impacts of our work.

## A  Proofs

### A.1  Notation and Setup

We here introduce a more rigorous notation for defining our trans-dimensional notation that will be used in a rigorous proof for the time-reversal of our jump diffusion. First, while it makes sense from a methodological and experimental point of view to present our setting as a *transdimensional* one, we slightly change the point of view in order to derive our theoretical results. We extend the space $\mathbb{R}^d$ to $\hat{\mathbb{R}}^d = \mathbb{R}^d \cup \{\infty\}$ using the *one-point compactification* of the space. We refer to [43] for details on this space. The point $\infty$ will be understood as a mask. For instance, let $x_1, x_2, x_3 \in \mathbb{R}^d$. Then $X = (x_1, x_2, x_3) \in (\hat{\mathbb{R}}^d)^N$ with $N = 3$ corresponds to a vector for which all components are *observed* whereas $X' = (x_1, \infty, x_3) \in (\hat{\mathbb{R}}^d)^N$ corresponds to a vector for which only the components on the first and third dimension are observed. The second dimension is *masked* in that case. Doing so, we will consider diffusion models on the space $\mathsf{X} = (\hat{\mathbb{R}}^d)^N$ with $d, N \in \mathbb{N}$. In the case of a video diffusion model, $N$ can be seen as the max number of frames. We will always consider that this space is equipped with its Borelian sigma-field $\mathcal{X}$ and all probability measures will be defined on $\mathcal{X}$.

We denote $\dim : \mathsf{X} \to \{0,1\}^N$ which is given for any $X = \{x_i\}_{i=1}^N \in \mathsf{X}$ by

$$\dim(X) = \{\delta_{\mathbb{R}^d}(x_i)\}_{i=1}^N.$$

In other words, $\dim(X)$ is a binary vector identifying the "dimension" of the vector $X$, i.e. which frames are observed. Going back to our example $X = (x_1, x_2, x_3) \in (\hat{\mathbb{R}}^d)^N$ and $X' = (x_1, \infty, x_3) \in (\hat{\mathbb{R}}^d)^N$, we have that $\dim(X) = \{1,1,1\}$ and $\dim(X') = \{1,0,1\}$. For any vector $u \in \{0,1\}^N$ we denote $|u| = \sum_{i=1}^N u_i$, i.e. the *active dimensions* of $u$ (or equivalently the non-masked frames). For any $X \in \mathsf{X}$ and $\mathsf{D} \in \{0,1\}^N$, we denote $X_\mathsf{D} = \{X_i'\}_{i=1}^N$ with $X_i' = X_i$ if $\mathsf{D}_i = 1$ and $X_i' = \infty$ if $\mathsf{D}_i = 0$.

We denote $\mathrm{C}_b^k(\mathbb{R}^d, \mathbb{R})$ the set of functions which are $k$ differentiable and bounded. Similarly, we denote $\mathrm{C}_b^k(\mathbb{R}^d, \mathbb{R})$ the set of functions which are $k$ differentiable and compactly supported. The set $\mathrm{C}_0^k(\mathbb{R}^d, \mathbb{R})$ denotes the functions which are $k$ differentiable and vanish when $\|x\| \to +\infty$. We note that $f \in \mathrm{C}(\hat{\mathbb{R}}^d)$, if $f \in \mathrm{C}(\mathbb{R}^d)$ and $f - f(\infty) \in \mathrm{C}_0(\mathbb{R}^d)$ and that $f \in \mathrm{C}^k(\hat{\mathbb{R}}^d)$ for any $k \in \mathbb{N}$ if the restriction of $f$ to $\mathbb{R}^d$ is in $\mathrm{C}^k(\mathbb{R}^d)$ and $f \in \mathrm{C}(\hat{\mathbb{R}}^d)$.

#### A.1.1  Transdimensional infinitesimal generator

To introduce rigorously the *transdimensional* diffusion model defined in Section 3.1, we will introduce its *infinitesimal generator*. The infinitesimal generator of a stochastic process can be roughly defined as its "probabilistic derivative". More precisely, assume that a stochastic process $(\mathbf{X}_t)_{t \geq 0}$ admits a transition semigroup $(\mathrm{P}_t)_{t \geq 0}$, i.e. for any $t \geq 0$, $\mathsf{A} \in \mathcal{X}$ and $X \in \mathsf{X}$ we have $\mathbb{P}(\mathbf{X}_t \in \mathsf{A} \mid \mathbf{X}_0 = x) = \mathrm{P}_t(x, \mathsf{A})$, then the infinitesimal generator is defined as $\mathcal{A}(f) = \lim_{t \to 0}(\mathrm{P}_t(f) - f)/t$, for every $f$ for which this quantity is well-defined.

Here, we start by introducing the infinitesimal generator of interest and give some intuition about its form. Then, we prove a time-reversal formula for this infinitesimal generator.

We consider $b : \mathbb{R}^d \to \mathbb{R}^d$, $\alpha : \{0,1\}^{NM} \to \mathbb{R}_+$. For any $f \in \mathrm{C}^2(\mathsf{X})$ and $X \in \mathsf{X}$ we define

$$\mathcal{A}(f)(X) = \sum_{i=1}^{N}\{\langle b(X_i), \nabla_{x_i} f(X)\rangle + \tfrac{1}{2}\Delta_{x_i} f(X)\}\delta_{\mathbb{R}^d}(X_i) \tag{5}$$
$$- \sum_{\mathsf{D}_1 \subset \mathsf{D}_0^{\Delta_0}} \cdots \sum_{\mathsf{D}_M \subset \mathsf{D}_{M-1}^{\Delta_{M-1}}} \alpha(\mathsf{D}_0, \dots, \mathsf{D}_M) \sum_{i=0}^{M-1}(f(X) - f(X_{\mathsf{D}_{i+1}}))\delta_{\mathsf{D}_i}(\dim(X)),$$

where $M \in \mathbb{N}$, $\mathsf{D}_0 = \{1\}^N$, $\{\Delta_j\}_{j=0}^{M-1} \in \mathbb{N}^M$ such that $\sum_{j=0}^{M-1}\Delta_j < N$ and for any $j \in \{0, \dots, M-1\}$, $\mathsf{D}_j^{\Delta_j}$ is the subset of $\{0,1\}^{\{1,\dots,N\}}$ such that $\mathsf{D}_{j+1} \in \mathsf{D}_j^{\Delta_j}$ if and only if $\mathsf{D}_j \cdot \mathsf{D}_{j+1} = \mathsf{D}_{j+1}$, where $\cdot$ is the pointwise multiplication operator, and $|\mathsf{D}_j| = |\mathsf{D}_{j+1}| + \Delta_j$. The condition $\mathsf{D}_j \cdot \mathsf{D}_{j+1} = \mathsf{D}_{j+1}$ means that the non-masked dimensions in $\mathsf{D}_{j+1}$ are also non-masked dimensions in $\mathsf{D}_j$. The condition $|\mathsf{D}_j| = |\mathsf{D}_{j+1}| + \Delta_j$ means that in order to go from $\mathsf{D}_j$ to $\mathsf{D}_{j+1}$, one needs to mask exactly $\Delta_j$ dimensions.

Therefore, a sequence $\{\Delta_j\}_{j=0}^{M-1} \in \mathbb{N}^M$ such that $\sum_{j=0}^{M-1}\Delta_j < N$ can be interpreted as a sequence of *drops* in dimension. At the core level, we have that $|\mathsf{D}_M| = N - \sum_{j=0}^{M-1}\Delta_j$. For instance if $|\mathsf{D}_M| = 1$, we have that at the end of the process, only one dimension is considered.

We choose $\alpha$ such that $\sum_{\mathsf{D}_1 \subset \mathsf{D}_0^{\Delta_0}} \cdots \sum_{\mathsf{D}_M \subset \mathsf{D}_{M-1}^{\Delta_{M-1}}} \alpha(\mathsf{D}_0, \dots, \mathsf{D}_M) = 1$. Therefore, $\alpha(\mathsf{D}_0, \dots, \mathsf{D}_M)$ corresponds to the probability to choose the *dimension path* $\mathsf{D}_0 \to \cdots \to \mathsf{D}_M$.

The part $X \mapsto \langle b(X_i), \nabla_{x_i} f(X)\rangle + \tfrac{1}{2}\Delta_{x_i} f(X)$ is more classical and corresponds to the *continuous part* of the diffusion process. We refer to [44] for a thorough introduction on infinitesimal generators. For simplicity, we omit the schedule coefficients in (5).

### A.1.2 Justification of the form of the infinitesimal generator

For any *dimension path* $\mathsf{P} = \mathsf{D}_0 \to \cdots \to \mathsf{D}_M$ (recall that $\mathsf{D}_0 = \{1\}^N$), we define the *jump kernel* $\mathbb{J}^{\mathsf{P}}$ as follows. For any $x \in \mathsf{X}$, we have $\mathbb{J}^{\mathsf{P}}(X, \mathrm{d}Y) = \sum_{i=0}^{M-1} \delta_{\mathsf{D}_i}(\dim(X))\delta_{X_{\mathsf{D}_{i+1}}}(\mathrm{d}Y)$. This operator corresponds to the *deletion* operator introduced in Section 3.1 . Hence, for any *dimension path* $\mathsf{P} = \mathsf{D}_0 \to \cdots \to \mathsf{D}_M$, we can define the associated infinitesimal generator: for any $f \in \mathrm{C}^2(\mathsf{X})$ and $X \in \mathsf{X}$ we define

$$\mathcal{A}^{\mathsf{P}}(f)(X) = \sum_{i=1}^{N}\{\langle b(x_i), \nabla_{x_i} f(X)\rangle + \tfrac{1}{2}\Delta_{x_i} f(X)\}\delta_{\mathbb{R}^d}(X_i) + \int_{\mathsf{X}}(f(Y) - f(X))\mathbb{J}^{\mathsf{P}}(X, \mathrm{d}Y).$$

We can define the following *jump kernel*

$$\mathbb{J} = \sum_{\mathsf{D}_1 \subset \mathsf{D}_0^{\Delta_0}} \cdots \sum_{\mathsf{D}_M \subset \mathsf{D}_{M-1}^{\Delta_{M-1}}} \alpha(\mathsf{D}_0, \dots, \mathsf{D}_M)\mathbb{J}^{\mathsf{P}}.$$

This corresponds to averaging the jump kernel over the different possible dimension paths. We have that for any $f \in \mathrm{C}^2(\mathsf{X})$ and $X \in \mathsf{X}$

$$\mathcal{A}(f)(X) = \sum_{i=1}^{N}\{\langle b(x_i), \nabla_{x_i} f(X)\rangle + \tfrac{1}{2}\Delta_{x_i} f(X)\}\delta_{\mathbb{R}^d}(X_i) + \int_{\mathsf{X}}(f(Y) - f(X))\mathbb{J}(X, \mathrm{d}Y). \tag{6}$$

In other words, $\mathcal{A} = \sum_{\mathsf{D}_1 \subset \mathsf{D}_0^{\Delta_0}} \cdots \sum_{\mathsf{D}_M \subset \mathsf{D}_{M-1}^{\Delta_{M-1}}} \alpha(\mathsf{D}_0, \dots, \mathsf{D}_M)\mathcal{A}^{\mathsf{P}}$.

In what follows, we assume that there exists a Markov process $(\mathbf{X}_t)_{t \geq 0}$ with infinitesimal generator $\mathcal{A}$. In order to sample from $(\mathbf{X}_t)_{t \geq 0}$, one choice is to first sample the dimension path $\mathsf{P}$ according to the probability $\alpha$. Second sample from the Markov process associated with the infinitesimal generator $\mathcal{A}^{\mathsf{P}}$. We can approximately sample from this process using the Lie-Trotter-Kato formula [44, Corollary 6.7, p.33].

Denote $(P_t)_{t \geq 0}$ the semigroup associated with $\mathcal{A}^{\mathsf{P}}$, $(Q_t)_{t \geq 0}$ the semigroup associated with the continuous part of $\mathcal{A}^{\mathsf{P}}$ and $(J_t)_{t \geq 0}$ the semigroup associated with the jump part of $\mathcal{A}^{\mathsf{P}}$. More precisely, we have that, $(Q_t)_{t \geq 0}$ is associated with $\mathcal{A}_{\mathrm{cont}}$ such that for any $f \in \mathrm{C}^2(\mathsf{X})$ and $X \in \mathsf{X}$

$$\mathcal{A}_{\mathrm{cont}}(f)(X) = \sum_{i=1}^{N}\{\langle b(X_i), \nabla_{x_i} f(X)\rangle + \tfrac{1}{2}\Delta_{x_i} f(X)\}.$$

In addition, we have that, $(Q_t)_{t \geq 0}$ is associated with $\mathcal{A}_{\mathrm{jump}}^{\mathsf{P}}$ such that for any $f \in \mathrm{C}^2(\mathsf{X})$ and $X \in \mathsf{X}$

$$\mathcal{A}_{\mathrm{jump}}^{\mathsf{P}}(f)(X) = \int_{\mathsf{X}}(f(Y) - f(X))\mathbb{J}^{\mathsf{P}}(X, \mathrm{d}Y).$$

First, note that $\mathcal{A}_{\mathrm{cont}}$ corresponds to the infinitesimal generator of a classical diffusion on the components which are not set to $\infty$. Hence, we can approximately sample from $(Q_t)_{t\geq 0}$ by sampling according to the Euler-Maruyama discretization of the associated diffusion, i.e. by setting

$$\mathbf{X}_t \approx \mathbf{X}_0 + tb(\mathbf{X}_0) + \sqrt{t}Z, \tag{7}$$

where $Z$ is a Gaussian random variable.

Similarly, in order to sample from $(J_t)_{t\geq 0}$, one should sample from the jump process defined as follows. On the interval $[0, \tau)$, we have $\bar{\mathbf{X}}_t = \mathbf{X}_0$. At time $\tau$, we define $\mathbf{X}_1 \sim \mathbb{J}(\mathbf{X}_0, \cdot)$ and repeat the procedure. In this case $\tau$ is defined as an exponential random variable with parameter 1. For $t > 0$ small enough the probability that $t > \tau$ is of order $t$. Therefore, we sample from J, i.e. the deletion kernel, with probability $t$. Combining this approximation and (7), we get approximate samplers for $(Q_t)_{t\geq 0}$ and $(J_t)_{t\geq 0}$. Under mild assumptions, the Lie-Trotter-Kato formula ensures that for any $t \geq 0$

$$P_t = \lim_{n\to+\infty} (Q_{t/n}J_{t/n})^n.$$

This justifies sampling according to Algorithm 1 (in the case of the forward process).

### A.2 Proof of Proposition 1

For the proof of Proposition 1, we first provide a rigorous proof using the notation introduced in A.1. We then follow this with a second proof that aims to be more intuitive using the notation used in the main paper.

#### A.2.1 Time-reversal for the transdimensional infinitesimal generator and Proof of Proposition 1

We are now going to derive the formula for the time-reversal of the transdimensional infinitesimal generator $\mathcal{A}$, see (5). This corresponds to a rigorous proof of Proposition 1. We refer to Section A.2.2 for a more intuitive, albeit less-rigorous, proof. We start by introducing the kernel $\mathbb{K}^{\mathsf{P}}$ given for any dimension path $\mathsf{D}_0 \to \cdots \to \mathsf{D}_M$, for any $i \in \{0, \ldots, M-1\}$, $Y \in \mathsf{D}_{i+1}$ and $\mathsf{A} \in \mathcal{X}$ by

$$\mathbb{K}^{\mathsf{P}}(Y, \mathsf{A}) = \sum_{i=0}^{M-1} \delta_{\mathsf{D}_{i+1}}(\dim(Y)) \int_{\mathsf{A}\cap\mathsf{D}_i} \frac{p_t((X_{\mathsf{D}_i\setminus\mathsf{D}_{i+1}}, Y_{\mathsf{D}_{i+1}})|\dim(\mathbf{X}_t)=\mathsf{D}_i)\mathbb{P}(\dim(\mathbf{X}_t)=\mathsf{D}_i)}{p_t(Y_{\mathsf{D}_{i+1}}|\dim(\mathbf{X}_t)=\mathsf{D}_{i+1})\mathbb{P}(\dim(\mathbf{X}_t)=\mathsf{D}_{i+1})} \mathrm{d}X_{\mathsf{D}_i\setminus\mathsf{D}_{i+1}}.$$

Note that this kernel is the same as the one considered in Proposition 1. It is well-defined under the following assumption.

**Assumption 1.** *For any $t > 0$ and $\mathsf{D} \subset \{0, 1\}^N$, we have that $\mathbf{X}_t$ conditioned to $\dim(\mathbf{X}_t) = \mathsf{D}$ admits a density w.r.t. the $|\mathsf{D}|d$-dimensional Lebesgue measure, denoted $p_t(\cdot|\dim(\mathbf{X}_t) = \mathsf{D})$.*

The following result will be key to establish the time-reversal formula.

**Lemma 1.** *Assume A1. Let $\mathsf{A}, \mathsf{B} \in \mathcal{X}$. Let $\mathsf{P}$ be a dimension path $\mathsf{D}_0 \to \cdots \to \mathsf{D}_M$ with $M \in \mathbb{N}$. Then, we have*

$$\mathbb{E}[\mathbf{1}_{\mathsf{A}}(\mathbf{X}_t)\mathbb{J}^{\mathsf{P}}(\mathbf{X}_t, \mathsf{B})] = \mathbb{E}[\mathbf{1}_{\mathsf{B}}(\mathbf{X}_t)\mathbb{K}^{\mathsf{P}}(\mathbf{X}_t, \mathsf{A})].$$

*Proof.* Let $\mathsf{A}, \mathsf{B} \in \mathcal{X}$. We have

$$\mathbb{E}[\mathbf{1}_{\mathsf{A}}(\mathbf{X}_t)\mathbb{J}^{\mathsf{P}}(\mathbf{X}_t, \mathsf{B})] = \sum_{i=0}^{M-1} \mathbb{E}[\mathbf{1}_{\mathsf{A}}(\mathbf{X}_t)\delta_{\mathsf{D}_i}(\dim(\mathbf{X}_t))\mathbf{1}_{\mathsf{B}}((\mathbf{X}_t)_{\mathsf{D}_{i+1}})]$$

$$= \sum_{i=0}^{M-1} \int_{\mathsf{A}\cap\mathsf{D}_i} p_t(X_{\mathsf{D}_i}|\dim(\mathbf{X}_t) = \mathsf{D}_i)\mathbb{P}(\dim(\mathbf{X}_t) = \mathsf{D}_i)\mathbf{1}_{\mathsf{B}}(X_{\mathsf{D}_{i+1}})\mathrm{d}X_{\mathsf{D}_i}$$

$$= \sum_{i=0}^{M-1} \int_{\mathsf{A}\cap\mathsf{D}_i} p_t(X_{\mathsf{D}_i}|\dim(\mathbf{X}_t) = \mathsf{D}_i)\mathbb{P}(\dim(\mathbf{X}_t) = \mathsf{D}_i)\mathbf{1}_{\mathsf{B}}(X_{\mathsf{D}_{i+1}})\mathrm{d}X_{\mathsf{D}_{i+1}}\mathrm{d}X_{\mathsf{D}_i\setminus\mathsf{D}_{i+1}}$$

$$= \sum_{i=0}^{M-1} \int_{\mathsf{B}\cap\mathsf{D}_{i+1}} \mathbf{1}_{\mathsf{B}}(X_{\mathsf{D}_{i+1}})$$
$$\times (\int_{\mathsf{A}\cap\mathsf{D}_i} \mathbf{1}_{\mathsf{A}}(X_{\mathsf{D}_i})p_t(X_{\mathsf{D}_i}|\dim(\mathbf{X}_t) = \mathsf{D}_i)\mathbb{P}(\dim(\mathbf{X}_t) = \mathsf{D}_i)\mathrm{d}X_{\mathsf{D}_i\setminus\mathsf{D}_{i+1}})\mathrm{d}X_{\mathsf{D}_{i+1}}$$

$$= \sum_{i=0}^{M-1} \int_{\mathsf{B}\cap\mathsf{D}_{i+1}} \mathbf{1}_{\mathsf{B}}(X_{\mathsf{D}_{i+1}})$$
$$\times \mathbb{K}^{\mathsf{P}}(X_{\mathsf{D}_{i+1}}, \mathsf{A})p_t(X_{\mathsf{D}_{i+1}}|\dim(\mathbf{X}_t) = \mathsf{D}_{i+1})\mathbb{P}(\dim(\mathbf{X}_t) = \mathsf{D}_{i+1})\mathrm{d}X_{\mathsf{D}_{i+1}}$$

$$= \sum_{i=0}^{M-1} \mathbb{E}[\delta_{\mathsf{D}_{i+1}}(\dim(\mathbf{X}_t))\mathbb{K}^{\mathsf{P}}(\mathbf{X}_t, \mathsf{A})\mathbf{1}_{\mathsf{B}}(\mathbf{X}_t)],$$

which concludes the proof. $\square$

Lemma 1 shows that $\mathbb{K}^{\mathsf{P}}$ verifies the *flux equation* associated with $\mathbb{J}^{\mathsf{P}}$. The flux equation is the discrete state-space equivalent of the classical time-reversal formula for continuous state-space. We refer to [45] for a rigorous treatment of time-reversal with jumps under entropic conditions.

We are also going to consider the following assumption which ensures that the integration by part formula is valid.

**Assumption 2.** *For any $t > 0$ and $i \in \{1, \ldots, N\}$, $\mathbf{X}_t$ admits a smooth density w.r.t. the $Nd$-dimensional Lebesgue measure denoted $p_t$ and we have that for any $f, h \in \mathrm{C}_b^2((\mathbb{R}^d)^N)$ for any $u \in [0, t]$ and $i \in \{1, \ldots, N\}$*

$$
\begin{aligned}
\mathbb{E}[\delta_{\mathbb{R}^d}((\mathbf{X}_u)_i) &\langle \nabla_{x_i} f(\mathbf{X}_u), \nabla_{x_i} h(\mathbf{X}_u) \rangle] \\
&= -\mathbb{E}[\delta_{\mathbb{R}^d}((\mathbf{X}_u)_i) h(\mathbf{X}_u)(\Delta_{x_i} f(\mathbf{X}_u) + \langle \nabla_{x_i} \log p_u(\mathbf{X}_u), \nabla_{x_i} f(\mathbf{X}_u) \rangle)].
\end{aligned}
$$

The second assumption ensures that we can apply the backward Kolmogorov evolution equation.

**Assumption 3.** *For any $g \in \mathrm{C}^2(\mathsf{X})$ and $t > 0$, we have that for any $u \in [0, t]$ and $X \in \mathsf{X}$, $\partial_u g(u, X) + \mathcal{A}(g)(u, X) = 0$, where for any $u \in [0, t]$ and $X \in \mathsf{X}$, $g(u, X) = \mathbb{E}[g(\mathbf{X}_t) \mid \mathbf{X}_u = X]$.*

We refer to [19] for conditions under A2 and A3 are valid in the setting of diffusion processes.

**Proposition 4.** *Assume* A1*,* A2 *and* A3*. Assume that there exists a Markov process $(\mathbf{X}_t)_{t \geq 0}$ solution of the martingale problem associated with (6). Let $T > 0$ and consider $(\mathbf{Y}_t)_{t \in [0,T]} = (\mathbf{X}_{T-t})_{t \in [0,T]}$. Then $(\mathbf{Y}_t)_{t \in [0,T]}$ is solution to the martingale problem associated with $\mathcal{R}$, where for any $f \in \mathrm{C}^2(\mathsf{X})$, $t \in (0, T)$ and $x \in \mathsf{X}$ we have*

$$
\begin{aligned}
\mathcal{R}(f)(t, X) = \sum_{i=1}^N &\{ -\langle b(X_i) + \nabla_{x_i} \log p_t(X), \nabla_{x_i} f(X) \rangle + \tfrac{1}{2}\Delta_{x_i} f(X) \} \delta_{\mathbb{R}^d}(X_i) \\
&+ \int_{\mathsf{X}} (f(Y) - f(X)) \mathbb{K}(X, \mathrm{d}Y).
\end{aligned}
$$

*Proof.* Let $f, g \in \mathrm{C}^2(\mathsf{X})$. In what follows, we show that for any $s, t \in [0, T]$ with $t \geq s$

$$
\mathbb{E}[(f(\mathbf{Y}_t) - f(\mathbf{Y}_s))g(\mathbf{Y}_s)] = \mathbb{E}[g(\mathbf{Y}_s) \textstyle\int_s^t \mathcal{R}(f)(u, \mathbf{Y}_u)\mathrm{d}u].
$$

More precisely, we show that for any $s, t \in [0, T]$ with $t \geq s$

$$
\mathbb{E}[(f(\mathbf{X}_t) - f(\mathbf{X}_s))g(\mathbf{X}_t)] = \mathbb{E}[-g(\mathbf{X}_t) \textstyle\int_s^t \mathcal{R}(f)(u, \mathbf{X}_u)\mathrm{d}u].
$$

Let $s, t \in [0, T]$, with $t \geq s$. Next, we denote for any $u \in [0, t]$ and $X \in \mathsf{X}$, $g(u, X) = \mathbb{E}[g(\mathbf{X}_t) \mid \mathbf{X}_u = X]$. Using A3, we have that for any $u \in [0, t]$ and $X \in \mathsf{X}$, $\partial_u g(u, X) + \mathcal{A}(g)(u, X) = 0$, i.e. $g$ satisfies the backward Kolmogorov equation. For any $u \in [0, t]$ and $X \in \mathsf{X}$, we have

$$
\begin{aligned}
\mathcal{A}(fg)(u, X) &= \partial_u g(u, X)f(X) + \sum_{i=1}^N (\langle b(X_i), \nabla_{x_i} g(u, X) \rangle + \tfrac{1}{2}\Delta_{x_i} g(u, X_i))f(X)\delta_{\mathbb{R}^d}(X_i) \\
&\quad + \sum_{i=1}^N (\langle b(X_i), \nabla_{x_i} f(X) \rangle + \tfrac{1}{2}\Delta_{x_i} f(X))g(u, X)\delta_{\mathbb{R}^d}(X_i) \\
&\quad + \sum_{i=1}^N \delta_{\mathbb{R}^d}(X_i)\langle \nabla_{x_i} f(X), \nabla_{x_i} g(u, X) \rangle + \mathbb{J}(X, fg) \\
&= \partial_u g(u, X)f(X) + \mathcal{A}(g)(u, X)f(X) + \mathbb{J}(X, fg) - \mathbb{J}(X, g)f(X) \\
&\quad + \sum_{i=1}^N (\langle b(X_i), \nabla_{x_i} f(X) \rangle + \tfrac{1}{2}\Delta_{x_i} f(X))g(u, X)\delta_{\mathbb{R}^d}(X_i) \\
&\quad + \sum_{i=1}^N \delta_{\mathbb{R}^d}(X_i)\langle \nabla_{x_i} f(X), \nabla_{x_i} g(u, X) \rangle \\
&= \sum_{i=1}^N (\langle b(X_i), \nabla_{x_i} f(X) \rangle + \tfrac{1}{2}\Delta_{x_i} f(X))g(u, X)\delta_{\mathbb{R}^d}(X_i) \\
&\quad + \sum_{i=1}^N \delta_{\mathbb{R}^d}(X_i)\langle \nabla_{x_i} f(X), \nabla_{x_i} g(u, X) \rangle + \mathbb{J}(X, fg) - \mathbb{J}(X, g)f(X). \quad (8)
\end{aligned}
$$

Using A2, we have that for any $u \in [0, t]$ and $i \in \{1, \ldots, N\}$

$$
\begin{aligned}
\mathbb{E}[\delta_{\mathbb{R}^d}((\mathbf{X}_u)_i) &\langle \nabla_{x_i} f(\mathbf{X}_u), \nabla_{x_i} g(u, \mathbf{X}_u) \rangle] \\
&= -\mathbb{E}[\delta_{\mathbb{R}^d}((\mathbf{X}_u)_i) g(u, \mathbf{X}_u)(\Delta_{x_i} f(\mathbf{X}_u) + \langle \nabla_{x_i} \log p_u(\mathbf{X}_u), \nabla_{x_i} f(\mathbf{X}_u) \rangle)]. \quad (9)
\end{aligned}
$$

In addition, we have that for any $X \in \mathsf{X}$ and $u \in [0, t]$, $\mathbb{J}(X, fg) - \mathbb{J}(X, g)f(X) = \int_{\mathsf{X}} g(u, Y)(f(Y) - f(X))\mathbb{J}(X, \mathrm{d}Y)$. Using Lemma 1, we get

$$
\mathbb{E}[\mathbb{J}(\mathbf{X}_u, fg) - \mathbb{J}(\mathbf{X}_u, f)g(u, \mathbf{X}_u)] = -\mathbb{E}[g(u, \mathbf{X}_u)\mathbb{K}(\mathbf{X}_u, f)]. \quad (10)
$$

Therefore, using (8), (9) and (10), we have
$$\mathbb{E}[\mathcal{A}(fg)(u, \mathbf{X}_u)] = \mathbb{E}[-\mathcal{R}(f)(u, \mathbf{X}_u)g(u, \mathbf{X}_u)].$$
Finally, we have
$$\begin{aligned}
\mathbb{E}[(f(\mathbf{X}_t) - f(\mathbf{X}_s))g(\mathbf{X}_t)] &= \mathbb{E}[g(t, \mathbf{X}_t)f(\mathbf{X}_t) - f(\mathbf{X}_s)g(s, \mathbf{X}_s)] \\
&= \mathbb{E}[\int_s^t \mathcal{A}(fg)(u, \mathbf{X}_u)\mathrm{d}u] \\
&= -\mathbb{E}[\int_s^t \mathcal{R}(f)(u, \mathbf{X}_u)g(u, \mathbf{X}_u)\mathrm{d}u] = -\mathbb{E}[g(\mathbf{X}_t)\int_s^t \mathcal{R}(f)(u, \mathbf{X}_u)\mathrm{d}u],
\end{aligned}$$
which concludes the proof. $\qquad\square$

### A.2.2 Intuitive Proof of Proposition 1

We recall Proposition 1.

**Proposition 1.** *The time reversal of a forward jump diffusion process given by drift $\overrightarrow{\mathbf{b}}_t$, diffusion coefficient $\overrightarrow{g}_t$, rate $\overrightarrow{\lambda}_t(n)$ and transition kernel $\sum_{i=1}^n K^{\mathrm{del}}(i|n)\delta_{\mathrm{del}(\mathbf{X},i)}(\mathbf{Y})$ is given by a jump diffusion process with drift $\overleftarrow{\mathbf{b}}_t^*(\mathbf{X})$, diffusion coefficient $\overleftarrow{g}_t^*$, rate $\overleftarrow{\lambda}_t^*(\mathbf{X})$ and transition kernel $\int_{\mathbf{y}^{\mathrm{add}}} \sum_{i=1}^{n+1} A_t^*(\mathbf{y}^{\mathrm{add}}, i|\mathbf{X})\delta_{\mathrm{ins}(\mathbf{X},\mathbf{y}^{\mathrm{add}},i)}(\mathbf{Y})\mathrm{d}\mathbf{y}^{\mathrm{add}}$ as defined below*
$$\overleftarrow{\mathbf{b}}_t^*(\mathbf{X}) = \overrightarrow{\mathbf{b}}_t(\mathbf{X}) - \overrightarrow{g}_t^2 \nabla_{\mathbf{x}} \log p_t(\mathbf{X}), \quad \overleftarrow{g}_t^* = \overrightarrow{g}_t,$$
$$\overleftarrow{\lambda}_t^*(\mathbf{X}) = \overrightarrow{\lambda}_t(n+1)\frac{\sum_{i=1}^{n+1} K^{\mathrm{del}}(i|n+1) \int_{\mathbf{y}^{\mathrm{add}}} p_t(\mathrm{ins}(\mathbf{X}, \mathbf{y}^{\mathrm{add}}, i))\mathrm{d}\mathbf{y}^{\mathrm{add}}}{p_t(\mathbf{X})},$$
$$A_t^*(\mathbf{y}^{\mathrm{add}}, i|\mathbf{X}) \propto p_t(\mathrm{ins}(\mathbf{X}, \mathbf{y}^{\mathrm{add}}, i))K^{\mathrm{del}}(i|n+1).$$

**Diffusion part.** Using standard diffusion models arguments such as [18] [45], we get
$$\overleftarrow{\mathbf{b}}_t^*(\mathbf{X}) = \overrightarrow{\mathbf{b}}_t(\mathbf{X}) - \overrightarrow{g}_t^2 \nabla_{\mathbf{x}} \log p_t(\mathbf{X}|n).$$

**Jump part.** We use the flux equation from [45] which intuitively relates the probability flow going in the forward direction with the probability flow going the backward direction with equality being achieved at the time reversal.
$$p_t(\mathbf{X})\overleftarrow{\lambda}_t^*(\mathbf{X})\overleftarrow{K}_t^*(\mathbf{Y}|\mathbf{X}) = p_t(\mathbf{Y})\overrightarrow{\lambda}_t(\mathbf{Y})\overrightarrow{K}_t(\mathbf{X}|\mathbf{Y})$$
$$p_t(\mathbf{X})\overleftarrow{\lambda}_t^*(\mathbf{X}) \int_{\mathbf{y}^{\mathrm{add}}} \sum_{i=1}^{n+1} A_t^*(\mathbf{y}^{\mathrm{add}}, i|\mathbf{X})\delta_{\mathrm{ins}(\mathbf{X},\mathbf{y}^{\mathrm{add}},i)}(\mathbf{Y})\mathrm{d}\mathbf{y}^{\mathrm{add}}$$
$$= p_t(\mathbf{Y})\overrightarrow{\lambda}_t(\mathbf{Y}) \sum_{i=1}^{n+1} K^{\mathrm{del}}(i|n+1)\delta_{\mathrm{del}(\mathbf{Y},i)}(\mathbf{X}). \qquad (11)$$
To find $\overleftarrow{\lambda}_t^*(\mathbf{X})$, we sum and integrate both sides over $m$ and $\mathbf{y}$, with $\mathbf{Y} = (m, \mathbf{y})$,
$$\sum_{m=1}^N \int_{\mathbf{y}\in\mathbb{R}^{md}} p_t(\mathbf{X})\overleftarrow{\lambda}_t^*(\mathbf{X}) \int_{\mathbf{y}^{\mathrm{add}}} \sum_{i=1}^{n+1} A_t^*(\mathbf{y}^{\mathrm{add}}, i|\mathbf{X})\delta_{\mathrm{ins}(\mathbf{X},\mathbf{y}^{\mathrm{add}},i)}(\mathbf{Y})\mathrm{d}\mathbf{y}^{\mathrm{add}}\mathrm{d}\mathbf{y}$$
$$= \sum_{m=1}^N \int_{\mathbf{y}\in\mathbb{R}^{md}} p_t(\mathbf{Y})\overrightarrow{\lambda}_t(\mathbf{Y}) \sum_{i=1}^{n+1} K^{\mathrm{del}}(i|n+1)\delta_{\mathrm{del}(\mathbf{Y},i)}(\mathbf{X})\mathrm{d}\mathbf{y}.$$
Now we use the fact that $\delta_{\mathrm{del}(\mathbf{Y},i)}(\mathbf{X})$ is 0 for any $m \neq n+1$,
$$p_t(\mathbf{X})\overleftarrow{\lambda}_t^*(\mathbf{X}) = \overrightarrow{\lambda}_t(n+1) \int_{\mathbf{y}\in\mathbb{R}^{(n+1)d}} p_t(\mathbf{Y}) \sum_{i=1}^{n+1} K^{\mathrm{del}}(i|n+1)\delta_{\mathrm{del}(\mathbf{Y},i)}(\mathbf{X})\mathrm{d}\mathbf{y}$$
$$= \overrightarrow{\lambda}_t(n+1) \sum_{i=1}^{n+1} K^{\mathrm{del}}(i|n+1) \int_{\mathbf{y}\in\mathbb{R}^{(n+1)d}} p_t(\mathbf{Y})\delta_{\mathrm{del}(\mathbf{Y},i)}(\mathbf{X})\mathrm{d}\mathbf{y}.$$
Now letting $\mathbf{Y} = \mathrm{ins}(\mathbf{X}, \mathbf{y}^{\mathrm{add}}, i)$,
$$p_t(\mathbf{X})\overleftarrow{\lambda}_t^*(\mathbf{X}) = \overrightarrow{\lambda}_t(n+1) \sum_{i=1}^{n+1} K^{\mathrm{del}}(i|n+1) \int_{\mathbf{y}^{\mathrm{add}}} p_t(\mathrm{ins}(\mathbf{X}, \mathbf{y}^{\mathrm{add}}, i))\mathrm{d}\mathbf{y}^{\mathrm{add}}$$
$$\overleftarrow{\lambda}_t^*(\mathbf{X})x = \overrightarrow{\lambda}_t(n+1)\frac{\sum_{i=1}^{n+1} K^{\mathrm{del}}(i|n+1) \int_{\mathbf{y}^{\mathrm{add}}} p_t(\mathrm{ins}(\mathbf{X}, \mathbf{y}^{\mathrm{add}}, i))\mathrm{d}\mathbf{y}^{\mathrm{add}}}{p_t(\mathbf{X})}.$$
To find $A_t^*(\mathbf{y}^{\mathrm{add}}, i|\mathbf{X})$, we start from (11) and set $\mathbf{Y} = \mathrm{ins}(\mathbf{X}, \mathbf{z}^{\mathrm{add}}, j)$ to get
$$p_t(\mathbf{X})\overleftarrow{\lambda}_t^*(\mathbf{X})A_t^*(\mathbf{z}^{\mathrm{add}}, j|\mathbf{X}) = p_t(\mathbf{Y})\overrightarrow{\lambda}_t(n+1)K^{\mathrm{del}}(j|n+1).$$
By inspection, we see immediately that
$$A_t^*(\mathbf{z}^{\mathrm{add}}, j|\mathbf{X}) \propto p_t(\mathrm{ins}(\mathbf{X}, \mathbf{z}^{\mathrm{add}}, j))K^{\mathrm{del}}(j|n+1).$$
With a re-labeling of $\mathbf{z}^{\mathrm{add}}$ and $j$ we achieve the desired form
$$A_t^*(\mathbf{y}^{\mathrm{add}}, i|\mathbf{X}) \propto p_t(\mathrm{ins}(\mathbf{X}, \mathbf{y}^{\mathrm{add}}, i))K^{\mathrm{del}}(i|n+1).$$

## A.3 Proof of Proposition 2

In this section we prove Proposition 2 using the notation from the main paper by following the framework of [17]. We operate on a state space $\mathcal{X} = \bigcup_{n=1}^{N}\{n\} \times \mathbb{R}^{nd}$. On this space the gradient operator $\nabla : \mathcal{C}(\mathcal{X}, \mathbb{R}) \to \mathcal{C}(\mathcal{X}, \mathcal{X})$ is defined as $\nabla f(\mathbf{X}) = \nabla_{\mathbf{x}}^{(nd)} f(\mathbf{X})$ where $\nabla_{\mathbf{x}}^{(nd)}$ is the standard gradient operator defined as $\mathcal{C}(\mathbb{R}^{nd}, \mathbb{R}) \to \mathcal{C}(\mathbb{R}^{nd}, \mathbb{R}^{nd})$ with respect to $\mathbf{x} \in \mathbb{R}^{nd}$. We will write integration with respect to a probability measure defined on $\mathcal{X}$ as an explicit sum over the number of components and integral over $\mathbb{R}^{nd}$ with respect to a probability density defined on $\mathbb{R}^{nd}$ i.e. $\int_{\mathbf{X}} f(\mathbf{X})\mu(\mathrm{d}\mathbf{X}) = \sum_{n=1}^{N} \int_{\mathbf{x}\in\mathbb{R}^{nd}} f(\mathbf{X})p(n)p(\mathbf{x}|n)\mathrm{d}\mathbf{x}$ where, for $A \subset \mathbb{R}^{nd}$, $\int_{(n,A)} \mu(d\mathbf{X}) = \int_{\mathbf{x}\in A} p(n)p(\mathbf{x}|n)\mathrm{d}\mathbf{x}$. We will write $p(\mathbf{X})$ as shorthand for $p(n)p(\mathbf{x}|n)$.

Following, [17], we start by augmenting our space with a time variable so that operators become time inhomogeneous on the extended space. We write this as $\bar{\mathbf{X}} = (\mathbf{X}, t)$ where $\bar{\mathbf{X}}$ lives in the extended space $\mathcal{S} = \mathcal{X} \times \mathbb{R}_{\geq 0}$. In the proof, we use the infinitesimal generators for the the forward and backward processes. An infinitesimal generator is defined as

$$\mathcal{A}(f)(\bar{\mathbf{X}}) = \lim_{t \to 0} \frac{\mathbb{E}_{p_{t|0}(\bar{\mathbf{Y}}|\bar{\mathbf{X}})}[f(\bar{\mathbf{Y}})] - f(\bar{\mathbf{X}})}{t}$$

and can be understood as a probabilistic version of a derivative. For our process on the augmented space $\mathcal{S}$, our generators decompose as $\mathcal{A} = \partial_t + \hat{\mathcal{A}}_t$ where $\hat{\mathcal{A}}_t$ operates only on the spatial components of $\bar{\mathbf{X}}$ i.e. $\mathbf{X}$ [17].

We now define the spatial infinitesimal generators for our forward and backward process. We will change our treatment of the time variable compared to the main text. Both our forward and backward processes will run from $t = 0$ to $t = T$, with the true time reversal of $\mathbf{X}$ following the forward process satisfying $(\mathbf{Y}_t)_{t\in[0,T]} = (\mathbf{X}_{T-t})_{t\in[0,T]}$. Further, we will write $\overrightarrow{g}_t$ as $g_t$ and $\overleftarrow{g}_t = g_{T-t}$ as we do not learn $g$ and this is the optimal relation from the time reversal. We define

$$\hat{\mathcal{L}}_t(f)(\mathbf{X}) = \overrightarrow{\mathbf{b}}_t(\mathbf{X})\cdot\nabla f(\mathbf{X}) + \tfrac{1}{2}g_t^2\Delta f(\mathbf{X}) + \overrightarrow{\lambda}_t(\mathbf{X})\sum_{m=1}^{N}\int_{\mathbf{y}\in\mathbb{R}^{md}} f(\mathbf{Y})(\overrightarrow{K}_t(\mathbf{Y}|\mathbf{X}) - \delta_{\mathbf{X}}(\mathbf{Y}))\mathrm{d}\mathbf{y},$$

as well as

$$\hat{\mathcal{K}}_t(f)(\mathbf{X}) = \overleftarrow{\mathbf{b}}_t^{\theta}(\mathbf{X})\cdot\nabla f(\mathbf{X}) + \tfrac{1}{2}g_{T-t}^2\Delta f(\mathbf{X}) + \overleftarrow{\lambda}_t^{\theta}(\mathbf{X})\sum_{m=1}^{N}\int_{\mathbf{y}\in\mathbb{R}^{md}} f(\mathbf{Y})(\overleftarrow{K}_t^{\theta}(\mathbf{Y}|\mathbf{X}) - \delta_{\mathbf{X}}(\mathbf{Y}))\mathrm{d}\mathbf{y}$$

where $\Delta = (\nabla\cdot\nabla)$ is the Laplace operator and $\delta$ is a dirac delta on $\mathcal{X}$ i.e. $\sum_{m=1}^{N}\int_{\mathbf{y}\in\mathbb{R}^{md}}\delta_{\mathbf{X}}(\mathbf{Y})\mathrm{d}\mathbf{y} = 1$ and $\sum_{m=1}^{N}\int_{\mathbf{y}\in\mathbb{R}^{md}} f(\mathbf{Y})\delta_{\mathbf{X}}(\mathbf{Y})\mathrm{d}\mathbf{y} = f(\mathbf{X})$.

**Verifying Assumption 1.** The first step in the proof is to verify Assumption 1 in [17]. Letting $\nu_t(\mathbf{X}) = p_{T-t}(\mathbf{X})$, we assume we can write $\partial_t p_t(\mathbf{X}) = \hat{\mathcal{K}}_t^* p_t(\mathbf{X})$ in the form $\mathcal{M}\nu + c\nu = 0$ for some function $c : \mathcal{S} \to \mathbb{R}$, where $\mathcal{M}$ is the generator of another auxiliary process on $\mathcal{S}$ and $\hat{\mathcal{K}}_t^*$ is the adjoint operator which satisfies $\langle\hat{\mathcal{K}}_t^* f, h\rangle = \langle f, \hat{\mathcal{K}}_t h\rangle$ i.e.

$$\sum_{n=1}^{N}\int_{\mathbf{x}\in\mathbb{R}^{nd}} h(\mathbf{X})\hat{\mathcal{K}}_t^*(f)(\mathbf{X})\mathrm{d}\mathbf{x} = \sum_{n=1}^{N}\int_{\mathbf{x}\in\mathbb{R}^{nd}} f(\mathbf{X})\hat{\mathcal{K}}_t(h)(\mathbf{X})\mathrm{d}\mathbf{x}$$

We now find $\hat{\mathcal{K}}_t^*$. We start by substituting in the form for $\hat{\mathcal{K}}_t$,

$$\sum_{n=1}^{N}\int_{\mathbf{x}\in\mathbb{R}^{nd}} f(\mathbf{X})\hat{\mathcal{K}}_t(h)(\mathbf{X})\mathrm{d}\mathbf{x} = \sum_{n=1}^{N}\int_{\mathbf{x}\in\mathbb{R}^{nd}} f(\mathbf{X})\{(\overleftarrow{\mathbf{b}}_t^{\theta}(\mathbf{x})\cdot\nabla h)(\mathbf{X}) + \tfrac{1}{2}g_{T-t}^2\Delta h(\mathbf{X}) + $$
$$\overleftarrow{\lambda}_t^{\theta}(\mathbf{X})\sum_{m=1}^{N}\int_{\mathbf{y}\in\mathbb{R}^{md}} h(\mathbf{Y})(\overleftarrow{K}_t^{\theta}(\mathbf{Y}|\mathbf{X}) - \delta_{\mathbf{X}}(\mathbf{Y}))\mathrm{d}\mathbf{y}\}\mathrm{d}\mathbf{x}$$

We first focus on the RHS terms corresponding to the diffusion part of the process

$$\sum_{n=1}^{N}\int_{\mathbf{x}\in\mathbb{R}^{nd}} f(\mathbf{X})\{(\overleftarrow{\mathbf{b}}_t^{\theta}\cdot\nabla h)(\mathbf{X}) + \tfrac{1}{2}g_{T-t}^2\Delta h(\mathbf{X})\}\mathrm{d}\mathbf{x}$$
$$= \sum_{n=1}^{N}\int_{\mathbf{x}\in\mathbb{R}^{nd}} f(\mathbf{X})(\overleftarrow{\mathbf{b}}_t^{\theta}\cdot\nabla h)(\mathbf{X}) + \tfrac{1}{2}g_{T-t}^2 f(\mathbf{X})\nabla\cdot\nabla h(\mathbf{X})\mathrm{d}\mathbf{x}$$
$$= \sum_{n=1}^{N}\int_{\mathbf{x}\in\mathbb{R}^{nd}} f(\mathbf{X})(\overleftarrow{\mathbf{b}}_t^{\theta}\cdot\nabla h)(\mathbf{X}) + \tfrac{1}{2}g_{T-t}^2 h(\mathbf{X})\nabla\cdot\nabla f(\mathbf{X})\mathrm{d}\mathbf{x}$$
$$= \sum_{n=1}^{N}\int_{\mathbf{x}\in\mathbb{R}^{nd}} -h(\mathbf{X})\nabla\cdot(f\overleftarrow{\mathbf{b}}_t^{\theta})(\mathbf{X}) + \tfrac{1}{2}g_{T-t}^2 h(\mathbf{X})\nabla\cdot\nabla f(\mathbf{X})\mathrm{d}\mathbf{x}$$
$$= \sum_{n=1}^{N}\int_{\mathbf{x}\in\mathbb{R}^{nd}} h(\mathbf{X})\{-\nabla\cdot(f\overleftarrow{\mathbf{b}}_t^{\theta})(\mathbf{X}) + \tfrac{1}{2}g_{T-t}^2\nabla\cdot\nabla f(\mathbf{X})\}\mathrm{d}\mathbf{x}$$
$$= \sum_{n=1}^{N}\int_{\mathbf{x}\in\mathbb{R}^{nd}} h(\mathbf{X})\{-f(\mathbf{X})\nabla\cdot\overleftarrow{\mathbf{b}}_t^{\theta}(\mathbf{X}) - \nabla f(\mathbf{X})\cdot\overleftarrow{\mathbf{b}}_t^{\theta}(\mathbf{X}) + \tfrac{1}{2}g_{T-t}^2\nabla\cdot\nabla f(\mathbf{X})\}\mathrm{d}\mathbf{x}.$$

where we apply integration by parts twice to arrive at the third line and once to arrive at the fourth line. We now focus on the RHS term corresponding to the jump part of the process

$$\sum_{n=1}^{N} \int_{\mathbf{x}\in\mathbb{R}^{nd}} f(\mathbf{X})\{\overleftarrow{\lambda}_t^\theta(\mathbf{X}) \sum_{m=1}^{N} \int_{\mathbf{y}\in\mathbb{R}^{md}} h(\mathbf{Y})(\overleftarrow{K}_t^\theta(\mathbf{Y}|\mathbf{X}) - \delta_{\mathbf{X}}(\mathbf{Y}))\mathrm{d}\mathbf{y}\}\mathrm{d}\mathbf{x}$$

$$= \sum_{m=1}^{N} \int_{\mathbf{y}\in\mathbb{R}^{md}} h(\mathbf{Y})\{\sum_{n=1}^{N} \int_{\mathbf{x}\in\mathbb{R}^{nd}} f(\mathbf{X})\overleftarrow{\lambda}_t^\theta(\mathbf{X})(\overleftarrow{K}_t^\theta(\mathbf{Y}|\mathbf{X}) - \delta_{\mathbf{X}}(\mathbf{Y}))\mathrm{d}\mathbf{x}\}\mathrm{d}\mathbf{y}$$

$$= \sum_{n=1}^{N} \int_{\mathbf{x}\in\mathbb{R}^{nd}} h(\mathbf{X})\{\sum_{m=1}^{N} \int_{\mathbf{y}\in\mathbb{R}^{md}} f(\mathbf{Y})\overleftarrow{\lambda}_t^\theta(\mathbf{Y})(\overleftarrow{K}_t^\theta(\mathbf{X}|\mathbf{Y}) - \delta_{\mathbf{Y}}(\mathbf{X}))\mathrm{d}\mathbf{y}\}\mathrm{d}\mathbf{x},$$

where on the last line we have relabelled $\mathbf{X}$ to $\mathbf{Y}$ and $\mathbf{Y}$ to $\mathbf{X}$. Putting both re-arranged forms for the RHS together, we obtain

$$\sum_{n=1}^{N} \int_{\mathbf{x}\in\mathbb{R}^{nd}} h(\mathbf{X})\hat{\mathcal{K}}_t^*(f)(\mathbf{X})\mathrm{d}\mathbf{x} =$$

$$\sum_{n=1}^{N} \int_{\mathbf{x}\in\mathbb{R}^{nd}} h(\mathbf{X})\{-f(\mathbf{X})\nabla \cdot \overleftarrow{\mathbf{b}}_t^\theta(\mathbf{X}) - \nabla f(\mathbf{X}) \cdot \overleftarrow{\mathbf{b}}_t^\theta(\mathbf{X}) + \tfrac{1}{2}g_{T-t}^2\nabla \cdot \nabla f(\mathbf{X})+$$

$$\sum_{m=1}^{N} \int_{\mathbf{y}\in\mathbb{R}^{md}} f(\mathbf{Y})\overleftarrow{\lambda}_t^\theta(\mathbf{Y})(\overleftarrow{K}_t^\theta(\mathbf{X}|\mathbf{Y}) - \delta_{\mathbf{Y}}(\mathbf{X}))\mathrm{d}\mathbf{y}\}\mathrm{d}\mathbf{x}.$$

We therefore have

$$\hat{\mathcal{K}}_t^*(f)(\mathbf{X}) = -f(\mathbf{X})\nabla \cdot \overleftarrow{\mathbf{b}}_t^\theta(\mathbf{X}) - \nabla f(\mathbf{X}) \cdot \overleftarrow{\mathbf{b}}_t^\theta(\mathbf{X}) + \tfrac{1}{2}g_{T-t}^2\Delta f(\mathbf{X})+$$

$$\sum_{m=1}^{N} \int_{\mathbf{y}\in\mathbb{R}^{md}} f(\mathbf{Y})\overleftarrow{\lambda}_t^\theta(\mathbf{Y})(\overleftarrow{K}_t^\theta(\mathbf{X}|\mathbf{Y}) - \delta_{\mathbf{X}}(\mathbf{Y}))\mathrm{d}\mathbf{y}.$$

Now we re-write $\partial_t p_t(\mathbf{X}) = \hat{\mathcal{K}}_t^* p_t(\mathbf{x})$ in the form $\mathcal{M}\nu + c\nu = 0$. We start by re-arranging

$$\partial_t p_t(\mathbf{X}) = \hat{\mathcal{K}}_t^* p_t(\mathbf{X}) \implies 0 = \partial_t \nu_t(\mathbf{X}) + \hat{\mathcal{K}}_{T-t}^* \nu_t(\mathbf{X}).$$

Substituting in our form for $\hat{\mathcal{K}}_t^*$ we obtain

$$0 = \partial_t \nu_t(\mathbf{X}) - \nu_t(\mathbf{X})\nabla \cdot \overleftarrow{\mathbf{b}}_{T-t}^\theta(\mathbf{X}) - \overleftarrow{\mathbf{b}}_{T-t}^\theta(\mathbf{X}) \cdot \nabla \nu_t(\mathbf{X}) + \tfrac{1}{2}g_t^2\Delta \nu_t(\mathbf{X})$$

$$+ \sum_{m=1}^{N} \int_{\mathbf{y}\in\mathbb{R}^{md}} \nu_t(\mathbf{Y})\overleftarrow{\lambda}_{T-t}^\theta(\mathbf{Y})(\overleftarrow{K}_{T-t}^\theta(\mathbf{X}|\mathbf{Y}) - \delta_{\mathbf{X}}(\mathbf{Y}))\mathrm{d}\mathbf{y} \qquad (12)$$

We define our auxiliary process to have generator $\mathcal{M} = \partial_t + \hat{\mathcal{M}}_t$ with

$$\hat{\mathcal{M}}_t(f)(\mathbf{X}) = \mathbf{b}_t^M(\mathbf{X})\cdot\nabla f(\mathbf{X}) + \tfrac{1}{2}g_t^2\Delta f(\mathbf{X}) + \lambda_t^M(\mathbf{X}) \sum_{m=1}^{N} \int_{\mathbf{y}\in\mathbb{R}^{md}} f(\mathbf{Y})(K_t^M(\mathbf{Y}|\mathbf{X}) - \delta_{\mathbf{X}}(\mathbf{Y}))\mathrm{d}\mathbf{y}$$

which is a jump diffusion process with drift $\mathbf{b}_t^M$, diffusion coefficient $g_t$, rate $\lambda_t^M$ and transition kernel $K_t^M$. Then if we have $\mathbf{b}_t^M = -\overleftarrow{\mathbf{b}}_{T-t}^\theta$,

$$\lambda_t^M(\mathbf{X}) = \sum_{m=1}^{N} \int_{\mathbf{y}\in\mathbb{R}^{md}} \overleftarrow{\lambda}_{T-t}^\theta(\mathbf{Y})\overleftarrow{K}_{T-t}^\theta(\mathbf{X}|\mathbf{Y})\mathrm{d}\mathbf{y}, \qquad (13)$$

and

$$K_t^M(\mathbf{Y}|\mathbf{X}) \propto \overleftarrow{\lambda}_{T-t}^\theta(\mathbf{Y})\overleftarrow{K}_{T-t}^\theta(\mathbf{X}|\mathbf{Y}). \qquad (14)$$

Then we have (12) can be rewritten as

$$0 = \partial_t \nu_t(\mathbf{X}) - \nu_t(\mathbf{X})\nabla \cdot \overleftarrow{\mathbf{b}}_{T-t}^\theta(\mathbf{X}) + \mathbf{b}_t^M(\mathbf{X}) \cdot \nabla \nu_t(\mathbf{X}) + \tfrac{1}{2}g_t^2\Delta \nu_t(\mathbf{X})$$

$$- \nu_t(\mathbf{X})\overleftarrow{\lambda}_{T-t}^\theta(\mathbf{X}) + \nu_t(\mathbf{X}) \sum_{m=1}^{N} \int_{\mathbf{y}\in\mathbb{R}^{md}} \overleftarrow{\lambda}_{T-t}^\theta(\mathbf{Y})\overleftarrow{K}_{T-t}^\theta(\mathbf{X}|\mathbf{Y})\mathrm{d}\mathbf{y}+$$

$$\lambda_t^M(\mathbf{x}) \sum_{m=1}^{N} \int_{\mathbf{y}\in\mathbb{R}^{md}} \nu_t(\mathbf{Y})(K_t^M(\mathbf{Y}|\mathbf{X}) - \delta_{\mathbf{X}}(\mathbf{Y}))\mathrm{d}\mathbf{y}$$

which is in the form $\mathcal{M}(\nu)(\bar{\mathbf{X}}) + c(\bar{\mathbf{X}})\nu(\bar{\mathbf{X}}) = 0$ if we let

$$c(\bar{\mathbf{X}}) = -\nabla \cdot \overleftarrow{\mathbf{b}}_{T-t}^\theta(\mathbf{X}) - \overleftarrow{\lambda}_{T-t}^\theta(\mathbf{X}) + \sum_{m=1}^{N} \int_{\mathbf{y}\in\mathbb{R}^{md}} \overleftarrow{\lambda}_{T-t}^\theta(\mathbf{Y})\overleftarrow{K}_{T-t}^\theta(\mathbf{X}|\mathbf{Y})\mathrm{d}\mathbf{y}.$$

**Verifying Assumption 2.** Now that we have verified Assumption 1, the second step in the proof is Assumption 2 from [17]. We assume there is a bounded measurable function $\alpha : \mathcal{S} \to (0, \infty)$ such

that $\alpha\mathcal{M}f = \mathcal{L}(f\alpha) - f\mathcal{L}\alpha$ for all functions $f : \mathcal{X} \to \mathbb{R}$ such that $f \in \mathcal{D}(\mathcal{M})$ and $f\alpha \in \mathcal{D}(\mathcal{L})$. Substituting in $\mathcal{M}$ and $\mathcal{L}$ we get

$$\alpha_t(\mathbf{X})[\partial_t f(\mathbf{X}) - \overleftarrow{\mathbf{b}}^\theta_{T-t}(\mathbf{X}) \cdot \nabla f(\mathbf{X})$$
$$+ \tfrac{1}{2}g_t^2 \Delta f(\mathbf{X}) + \lambda_t^M(\mathbf{X}) \textstyle\sum_{m=1}^N \int_{\mathbf{y} \in \mathbb{R}^{md}} f(\mathbf{Y})(K_t^M(\mathbf{Y}|\mathbf{X}) - \delta_{\mathbf{X}}(\mathbf{Y}))\mathrm{d}\mathbf{y}]$$
$$= \partial_t(f\alpha_t)(\mathbf{X}) + \overrightarrow{\mathbf{b}}_t(\mathbf{X}) \cdot \nabla(f\alpha_t)(\mathbf{X}) + \tfrac{1}{2}g_t^2\Delta(f\alpha_t)(\mathbf{X})$$
$$+ \overrightarrow{\lambda}_t(\mathbf{X}) \textstyle\sum_{m=1}^N \int_{\mathbf{y} \in \mathbb{R}^{md}} f(\mathbf{Y})\alpha_t(\mathbf{Y})(\overrightarrow{K}_t(\mathbf{Y}|\mathbf{X}) - \delta_{\mathbf{X}}(\mathbf{Y}))\mathrm{d}\mathbf{y}$$
$$- f(\mathbf{X})[\partial_t\alpha_t(\mathbf{X}) + \overrightarrow{\mathbf{b}}_t(\mathbf{X}) \cdot \nabla\alpha_t(\mathbf{X}) + \tfrac{1}{2}g_t^2\Delta\alpha_t(\mathbf{X})$$
$$+ \overrightarrow{\lambda}_t(\mathbf{X}) \textstyle\sum_{m=1}^N \int_{\mathbf{y} \in \mathbb{R}^{md}} \alpha_t(\mathbf{Y})(\overrightarrow{K}_t(\mathbf{Y}|\mathbf{X}) - \delta_{\mathbf{X}}(\mathbf{Y}))\mathrm{d}\mathbf{y}] \tag{15}$$

Since $f$ does not depend on time, $\partial_t f(\mathbf{X}) = 0$ and $\partial_t(f\alpha_t)(\mathbf{X}) = f(\mathbf{X})\partial_t\alpha_t(\mathbf{X})$ thus the $\partial_t$ terms on the RHS also cancel out. Comparing terms on the LHS and RHS relating to the diffusion part of the process we obtain

$$- \alpha_t(\mathbf{X})(\overleftarrow{\mathbf{b}}^\theta_{T-t}(\mathbf{X}) \cdot \nabla f(\mathbf{X})) + \tfrac{1}{2}\alpha_t(\mathbf{X})g_t^2\Delta f(\mathbf{X}) =$$
$$\overrightarrow{\mathbf{b}}_t(\mathbf{X}) \cdot \nabla(f\alpha_t)(\mathbf{X}) + \tfrac{1}{2}g_t^2\Delta(f\alpha_t)(\mathbf{X}) - f(\mathbf{X})\overrightarrow{\mathbf{b}}_t(\mathbf{X}) \cdot \nabla\alpha_t(\mathbf{X}) - \tfrac{1}{2}f(\mathbf{X})g_t^2\Delta\alpha_t(\mathbf{X}).$$

Therefore, we get

$$- \alpha_t(\mathbf{X})(\overleftarrow{\mathbf{b}}^\theta_{T-t}(\mathbf{X}) \cdot \nabla f(\mathbf{X})) + \tfrac{1}{2}\alpha_t(\mathbf{X})g_t^2\Delta f(\mathbf{X}) =$$
$$\overrightarrow{\mathbf{b}}_t(\mathbf{X}) \cdot (f(\mathbf{X})\nabla\alpha_t(\mathbf{X}) + \alpha_t(\mathbf{X})\nabla f(\mathbf{X}))$$
$$+ \tfrac{1}{2}g_t^2\big(2\nabla f(\mathbf{X}) \cdot \nabla\alpha_t(\mathbf{X}) + f(\mathbf{X})\Delta\alpha_t(\mathbf{X}) + \alpha_t(\mathbf{X})\Delta f(\mathbf{X})\big)$$
$$- f(\mathbf{X})\overrightarrow{\mathbf{b}}_t((\mathbf{X}) \cdot \nabla\alpha_t(\mathbf{X}) - \tfrac{1}{2}f(\mathbf{X})g_t^2\Delta\alpha_t(\mathbf{X}).$$

Simplifying the above expression, we get

$$-\alpha_t(\mathbf{X})(\overleftarrow{\mathbf{b}}^\theta_{T-t}(\mathbf{X}) \cdot \nabla f(\mathbf{X})) = \alpha_t(\mathbf{X})\overrightarrow{\mathbf{b}}_t(\mathbf{X}) \cdot \nabla f(\mathbf{X}) + g_t^2\nabla f(\mathbf{X}) \cdot \nabla\alpha_t(\mathbf{X})$$
$$(-\alpha_t(\mathbf{X})\overleftarrow{\mathbf{b}}^\theta_{T-t}(\mathbf{X})) \cdot \nabla f(\mathbf{X}) = (\alpha_t(\mathbf{X})\overrightarrow{\mathbf{b}}_t(\mathbf{X}) + g_t^2\nabla\alpha_t(\mathbf{X})) \cdot \nabla f(\mathbf{X}).$$

This is true for any $f$ implying

$$-\alpha_t(\mathbf{X})\overleftarrow{\mathbf{b}}^\theta_{T-t}(\mathbf{X}) = \alpha_t(\mathbf{X})\overrightarrow{\mathbf{b}}_t(\mathbf{X}) + g_t^2\nabla\alpha_t(\mathbf{X}).$$

This implies that $\alpha_t(\mathbf{X})$ satisfies

$$\nabla \log \alpha_t(\mathbf{X}) = -\tfrac{1}{g_t^2}(\overrightarrow{\mathbf{b}}_t(\mathbf{X}) + \overleftarrow{\mathbf{b}}^\theta_{T-t}(\mathbf{X})) \tag{16}$$

Comparing terms from the LHS and RHS of (15) relating to the jump part of the process we obtain

$$\alpha_t(\mathbf{X})\lambda_t^M(\mathbf{X}) \textstyle\sum_{m=1}^N \int_{\mathbf{y} \in \mathbb{R}^{md}} f(\mathbf{Y})(K_t^M(\mathbf{Y}|\mathbf{X}) - \delta_{\mathbf{X}}(\mathbf{Y}))\mathrm{d}\mathbf{y} =$$
$$\overrightarrow{\lambda}_t(\mathbf{X}) \textstyle\sum_{m=1}^N \int_{\mathbf{y} \in \mathbb{R}^{md}} f(\mathbf{Y})\alpha_t(\mathbf{Y})(\overrightarrow{K}_t(\mathbf{Y}|\mathbf{X}) - \delta_{\mathbf{X}}(\mathbf{Y}))\mathrm{d}\mathbf{y}$$
$$- f(\mathbf{X})\overrightarrow{\lambda}_t(\mathbf{X}) \textstyle\sum_{m=1}^N \int_{\mathbf{y} \in \mathbb{R}^{md}} \alpha_t(\mathbf{Y})(\overrightarrow{K}_t(\mathbf{Y}|\mathbf{X}) - \delta_{\mathbf{X}}(\mathbf{Y}))\mathrm{d}\mathbf{y}.$$

Hence, we have

$$\alpha_t(\mathbf{X}) \textstyle\sum_{m=1}^N \int_{\mathbf{y} \in \mathbb{R}^{md}} f(\mathbf{Y})\lambda_t^M(\mathbf{X})K_t^M(\mathbf{Y}|\mathbf{X})\mathrm{d}\mathbf{y} - \alpha_t(\mathbf{X})\lambda_t^M(\mathbf{X})f(\mathbf{X}) =$$
$$\overrightarrow{\lambda}_t(\mathbf{X}) \textstyle\sum_{m=1}^N \int_{\mathbf{y} \in \mathbb{R}^{md}} f(\mathbf{Y})\alpha_t(\mathbf{Y})\overrightarrow{K}_t(\mathbf{Y}|\mathbf{X})\mathrm{d}\mathbf{y}$$
$$- f(\mathbf{X})\overrightarrow{\lambda}_t(\mathbf{X}) \textstyle\sum_{m=1}^N \int_{\mathbf{y} \in \mathbb{R}^{md}} \alpha_t(\mathbf{Y})\overrightarrow{K}_t(\mathbf{Y}|\mathbf{X})\mathrm{d}\mathbf{y}.$$

Recalling the definitions of $\lambda_t^M(\mathbf{X})$ and $K^M(\mathbf{Y}|\mathbf{X})$, (13) and (14), we get

$$\alpha_t(\mathbf{X}) \textstyle\sum_{m=1}^N \int_{\mathbf{y} \in \mathbb{R}^{md}} f(\mathbf{Y})\overleftarrow{\lambda}^\theta_{T-t}(\mathbf{Y})\overleftarrow{K}^\theta_{T-t}(\mathbf{X}|\mathbf{Y})\mathrm{d}\mathbf{y}$$
$$- \alpha_t(\mathbf{X})f(\mathbf{X}) \textstyle\sum_{m=1}^N \int_{\mathbf{y} \in \mathbb{R}^{md}} \overleftarrow{\lambda}^\theta_{T-t}(\mathbf{Y})\overleftarrow{K}^\theta_{T-t}(\mathbf{X}|\mathbf{Y})\mathrm{d}\mathbf{y} =$$
$$\overrightarrow{\lambda}_t(\mathbf{X}) \textstyle\sum_{m=1}^N \int_{\mathbf{y} \in \mathbb{R}^{md}} f(\mathbf{Y})\alpha_t(\mathbf{Y})\overrightarrow{K}_t(\mathbf{Y}|\mathbf{X})\mathrm{d}\mathbf{y}$$
$$- f(\mathbf{X})\overrightarrow{\lambda}_t(\mathbf{X}) \textstyle\sum_{m=1}^N \int_{\mathbf{y} \in \mathbb{R}^{md}} \alpha_t(\mathbf{Y})\overrightarrow{K}_t(\mathbf{Y}|\mathbf{X})\mathrm{d}\mathbf{y}.$$

This equality is satisfied if $\alpha_t(\mathbf{X})$ follows the following relation

$$\alpha_t(\mathbf{Y}) = \alpha_t(\mathbf{X})\frac{\overleftarrow{\lambda}^\theta_{T-t}(\mathbf{Y})\overleftarrow{K}^\theta_{T-t}(\mathbf{X}|\mathbf{Y})}{\overrightarrow{\lambda}_t(\mathbf{X})\overrightarrow{K}_t(\mathbf{Y}|\mathbf{X})} \quad \text{for } n \neq m \tag{17}$$

We only require this relation to be satisfied for $n \neq m$ because both $\overrightarrow{K}_t(\mathbf{Y}|\mathbf{X})$ and $\overleftarrow{K}^\theta_{T-t}(\mathbf{X}|\mathbf{Y})$ are 0 for $n = m$. We note at this point, as in [17], that if we have $\alpha_t(\mathbf{X}) = 1/p_t(\mathbf{X})$ and $\overleftarrow{\lambda}^\theta_{T-t}$ and $\overleftarrow{K}^\theta_{T-t}(\mathbf{X}|\mathbf{Y})$ equal to the true time-reversals, then both (16), and (17) are satisfied. However, $\alpha_t(\mathbf{X}) = 1/p_t(\mathbf{X})$ is not the only $\alpha_t$ to satisfy these equations. (16) and (17) can be thought of as enforcing a certain parameterization of the generative process in terms of $\alpha_t$ [17].

**Concluding the proof.** Now for the final part of the proof, we substitute our value for $\alpha$ into the $\mathcal{I}_{\text{ISM}}$ loss from [17] which is equal to the negative of the evidence lower bound on $\mathbb{E}_{p_{\text{data}}(\mathbf{X}_0)}[\log p^\theta_0(\mathbf{X}_0)]$ up to a constant independent of $\theta$. Defining $\beta_t(\mathbf{X}_t) = 1/\alpha_t(\mathbf{X}_t)$, we have

$$\mathcal{I}_{\text{ISM}}(\beta) = \int_0^T \mathbb{E}_{p_t(\mathbf{X}_t)}[\tfrac{\hat{\mathcal{L}}^*_t\beta_t(\mathbf{X}_t)}{\beta_t(\mathbf{X}_t)} + \hat{\mathcal{L}}_t \log \beta_t(\mathbf{X}_t)]\mathrm{d}t.$$

We split the spatial infintesimal generator of the forward process into the generator corresponding to the diffusion and the generator corresponding to the jump part, $\hat{\mathcal{L}} = \hat{\mathcal{L}}^{\text{diff}}_t + \hat{\mathcal{L}}^{\text{J}}_t$ with

$$\hat{\mathcal{L}}^{\text{diff}}_t(f)(\mathbf{X}) = \overrightarrow{\mathbf{b}}_t(\mathbf{X}) \cdot \nabla f(\mathbf{X}) + \tfrac{1}{2}g_t^2\Delta f(\mathbf{X}).$$

and

$$\hat{\mathcal{L}}^{\text{J}}_t(f)(\mathbf{X}) = \overrightarrow{\lambda}_t(\mathbf{X})\sum_{m=1}^N \int_{\mathbf{y}\in\mathbb{R}^{md}} f(\mathbf{Y})(\overrightarrow{K}_t(\mathbf{Y}|\mathbf{X}) - \delta_{\mathbf{X}}(\mathbf{Y}))\mathrm{d}\mathbf{y}.$$

By comparison with the approach to find the adjoint $\hat{\mathcal{K}}^*_t$, we also have $\hat{\mathcal{L}}^*_t = \hat{\mathcal{L}}^{\text{diff}*}_t + \hat{\mathcal{L}}^{\text{J}*}_t$ with

$$\hat{\mathcal{L}}^{\text{diff}*}_t(f)(\mathbf{X}) = -f(\mathbf{X})\nabla \cdot \overrightarrow{\mathbf{b}}_t(\mathbf{X}) - \nabla f(\mathbf{X}) \cdot \overrightarrow{\mathbf{b}}_t(\mathbf{X}) + \tfrac{1}{2}g_t^2\Delta f(\mathbf{X}).$$

In addition, we get

$$\hat{\mathcal{L}}^{\text{J}*}_t(f)(\mathbf{X}) = \sum_{m=1}^N \int_{\mathbf{y}\in\mathbb{R}^{md}} f(\mathbf{Y})\overrightarrow{\lambda}_t(\mathbf{Y})(\overrightarrow{K}_t(\mathbf{X}|\mathbf{Y}) - \delta_{\mathbf{X}}(\mathbf{Y}))\mathrm{d}\mathbf{y}.$$

Finally, $\mathcal{I}_{\text{ISM}}$ becomes

$$\mathcal{I}_{\text{ISM}}(\beta) = \int_0^T \mathbb{E}_{p_t(\mathbf{X}_t)}[\tfrac{\hat{\mathcal{L}}^{\text{diff}*}_t\beta_t(\mathbf{X}_t)}{\beta_t(\mathbf{X}_t)} + \hat{\mathcal{L}}^{\text{diff}}_t \log \beta_t(\mathbf{X}_t)]dt + \int_0^T \mathbb{E}_{p_t(\mathbf{X}_t)}[\tfrac{\hat{\mathcal{L}}^{\text{J}*}_t\beta_t(\mathbf{X}_t)}{\beta_t(\mathbf{X}_t)} + \hat{\mathcal{L}}^{\text{J}} \log \beta_t(\mathbf{X}_t)]dt$$
$$= \mathcal{I}^{\text{diff}}_{\text{ISM}}(\beta) + \mathcal{I}^{\text{J}}_{\text{ISM}}(\beta),$$

where we have named the two terms corresponding to the diffusion and jump part of the process as $\mathcal{I}^{\text{diff}}_{\text{ISM}}, \mathcal{I}^{\text{J}}_{\text{ISM}}$ respectively. For the diffusion part of the loss, we use the denoising form of the objective proven in Appendix E of [17] which is equivalent to $\mathcal{I}^{\text{diff}}_{\text{ISM}}$ up to a constant independent of $\theta$

$$\mathcal{I}^{\text{diff}}_{\text{ISM}}(\beta) = \int_0^T \mathbb{E}_{p_{0,t}(\mathbf{X}_0,\mathbf{X}_t)}[\tfrac{\hat{\mathcal{L}}^{\text{diff}}_t(p_{t|0}(\cdot|\mathbf{X}_0)\alpha_t(\cdot))(\mathbf{X}_t)}{p_{t|0}(\mathbf{X}_t|\mathbf{X}_0)\alpha_t(\mathbf{X}_t)} - \hat{\mathcal{L}}^{\text{diff}}_t \log(p_{t|0}(\cdot|\mathbf{X}_0)\alpha_t(\cdot))(\mathbf{X}_t)]\mathrm{d}t + \text{const}.$$

To simplify this expression, we first re-arrange $\hat{\mathcal{L}}^{\text{diff}}_t(h)$ for some general function $h : \mathcal{S} \to \mathbb{R}$.

$$\frac{\hat{\mathcal{L}}^{\text{diff}}_t(h)}{h} - \hat{\mathcal{L}}^{\text{diff}}_t(\log h) = \frac{\overrightarrow{\mathbf{b}}_t\cdot\nabla h}{h} + \tfrac{1}{2}g_t^2\frac{\Delta h}{h} - \overrightarrow{\mathbf{b}}_t \cdot \nabla \log h - \tfrac{1}{2}g_t^2\Delta \log h$$
$$= \tfrac{1}{2}g_t^2(\tfrac{\nabla\cdot\nabla h}{h} - \nabla \cdot \nabla \log h)$$
$$= \tfrac{1}{2}g_t^2\|\nabla \log h\|^2.$$

Setting $h = p_{t|0}(\cdot|\mathbf{X}_0)\alpha_t(\cdot)$, our diffusion part of the loss becomes

$$\mathcal{I}^{\text{diff}}_{\text{ISM}}(\beta) = \tfrac{1}{2}\int_0^T g_t^2\mathbb{E}_{p_{0,t}(\mathbf{X}_0,\mathbf{X}_t)}[\|\nabla \log p_{t|0}(\mathbf{X}_t|\mathbf{X}_0) + \nabla \log \alpha_t(\mathbf{X}_t)\|^2]\mathrm{d}t + \text{const}$$

We then directly parameterize $\nabla \log \alpha_t(\mathbf{X}_t)$ as $-s^\theta_t(\mathbf{X}_t)$

$$\mathcal{I}^{\text{diff}}_{\text{ISM}}(\beta) = \tfrac{1}{2}\int_0^T g_t^2\mathbb{E}_{p_{0,t}(\mathbf{X}_0,\mathbf{X}_t)}[\|\nabla \log p_{t|0}(\mathbf{X}_t|\mathbf{X}_0) - s^\theta_t(\mathbf{X}_t)\|^2]\mathrm{d}t + \text{const}.$$

We now focus on the expectation within the integral to re-write it in an easy to calculate form

$$\mathbb{E}_{p_{0,t}(\mathbf{X}_0,\mathbf{X}_t)}[\|\nabla \log p_{t|0}(\mathbf{X}_t|\mathbf{X}_0) - s_t^\theta(\mathbf{X}_t)\|^2]$$
$$= \mathbb{E}_{p_{0,t}(\mathbf{X}_0,\mathbf{X}_t)}[\|s_t^\theta(\mathbf{X}_t)\|^2 - 2s_t^\theta(\mathbf{X}_t)^T \nabla \log p_{0,t}(\mathbf{X}_0,\mathbf{X}_t)] + \text{const}$$

Now we note that we can re-write $\nabla \log p_{0,t}(\mathbf{X}_0,\mathbf{X}_t)$ using $M_t$ where $M_t$ is a mask variable $M_t \in \{0,1\}^{n_0}$ that is 0 for components of $\mathbf{X}_0$ that have been deleted to get to $\mathbf{X}_t$ and 1 for components that remain in $\mathbf{X}_t$.

$$\nabla \log p_{0,t}(\mathbf{X}_0,\mathbf{X}_t) = \frac{1}{p_{0,t}(\mathbf{X}_0,\mathbf{X}_t)} \nabla p_{0,t}(\mathbf{X}_0,\mathbf{X}_t)$$
$$= \frac{1}{p_{0,t}(\mathbf{X}_0,\mathbf{X}_t)} \nabla \sum_{M_t} p_{0,t}(\mathbf{X}_0,\mathbf{X}_t,M_t)$$
$$= \sum_{M_t} \frac{1}{p_{0,t}(\mathbf{X}_0,\mathbf{X}_t)} \nabla p_{0,t}(\mathbf{X}_0,\mathbf{X}_t,M_t)$$
$$= \sum_{M_t} \frac{p(n_t,M_t,\mathbf{X}_0)}{p_{0,t}(\mathbf{X}_0,\mathbf{X}_t)} \nabla p_{t|0}(\mathbf{x}_t|n_t,\mathbf{X}_0,M_t)$$
$$= \sum_{M_t} \frac{p(M_t|\mathbf{X}_0,\mathbf{X}_t)}{p(\mathbf{x}_t|n_t,\mathbf{X}_0,M_t)} \nabla p_{t|0}(\mathbf{x}_t|n_t,\mathbf{X}_0,M_t)$$
$$= \mathbb{E}_{p(M_t|\mathbf{X}_0,\mathbf{X}_t)}[\nabla \log p_{t|0}(\mathbf{x}_t|n_t,\mathbf{X}_0,M_t)]$$

Substituting this back in we get

$$\mathbb{E}_{p_{0,t}(\mathbf{X}_0,\mathbf{X}_t)}[\|\nabla \log p_{t|0}(\mathbf{X}_t|\mathbf{X}_0) - s_t^\theta(\mathbf{X}_t)\|^2]$$
$$= \mathbb{E}_{p_{0,t}(\mathbf{X}_0,\mathbf{X}_t)}[\|s_t^\theta(\mathbf{X}_t)\|^2 - 2s_t^\theta(\mathbf{X}_t)^T \mathbb{E}_{p(M_t|\mathbf{X}_0,\mathbf{X}_t)}[\nabla \log p_{t|0}(\mathbf{x}_t|n_t,\mathbf{X}_0,M_t)]] + \text{const}$$
$$= \mathbb{E}_{p_{0,t}(\mathbf{X}_0,\mathbf{X}_t,M_t)}[\|\nabla \log p_{t|0}(\mathbf{x}_t|n_t,\mathbf{X}_0,M_t) - s_t^\theta(\mathbf{X}_t)\|^2] + \text{const}.$$

Therefore, the diffusion part of $\mathcal{I}_{\text{ISM}}$ can be written as

$$\mathcal{I}_{\text{ISM}}^{\text{diff}}(\beta) = \frac{T}{2} \mathbb{E}_{\mathcal{U}(t;0,T)p_{0,t}(\mathbf{X}_0,\mathbf{X}_t,M_t)}[g_t^2 \|\nabla \log p_{t|0}(\mathbf{x}_t|n_t,\mathbf{X}_0,M_t) - s_t^\theta(\mathbf{X}_t)\|^2] + \text{const}.$$

We now focus on the jump part of the loss $\mathcal{I}_{\text{ISM}}^{\text{J}}$. We first substitute in $\hat{\mathcal{L}}_t^{\text{J}}$ and $\hat{\mathcal{L}}_t^{\text{J}*}$

$$\mathcal{I}_{\text{ISM}}^{\text{J}} = \int_0^T \mathbb{E}_{p_t(\mathbf{x}_t)}[\sum_m \int_{\mathbf{y}\in\mathbb{R}^{md}} \overrightarrow{\lambda}_t(\mathbf{Y}) \frac{\beta_t(\mathbf{Y})}{\beta_t(\mathbf{X}_t)}(\overrightarrow{K}_t(\mathbf{X}_t|\mathbf{Y}) - \delta_{\mathbf{Y}}(\mathbf{X}_t))d\mathbf{y}+$$
$$\overrightarrow{\lambda}_t(\mathbf{X}_t) \sum_{m=1}^N \int_{\mathbf{y}\in\mathbb{R}^{md}} \overrightarrow{K}_t(\mathbf{Y}|\mathbf{X}_t) \log \beta_t(\mathbf{Y})d\mathbf{y} - \overrightarrow{\lambda}_t(\mathbf{X}_t) \log \beta_t(\mathbf{X}_t)]dt. \tag{18}$$

Noting that $\beta_t(\mathbf{X}_t) = 1/\alpha_t(\mathbf{X}_t)$, we get

$$\frac{\beta_t(\mathbf{X}_t)}{\beta_t(\mathbf{Y})} = \frac{\overleftarrow{\lambda}_{T-t}^\theta(\mathbf{Y})\overleftarrow{K}_{T-t}^\theta(\mathbf{X}_t|\mathbf{Y})}{\overrightarrow{\lambda}_t(\mathbf{X}_t)\overrightarrow{K}_t(\mathbf{Y}|\mathbf{X}_t)} \quad \text{for } n_t \neq m \tag{19}$$

or swapping labels for $\mathbf{X}_t$ and $\mathbf{Y}$,

$$\frac{\beta_t(\mathbf{Y})}{\beta_t(\mathbf{X}_t)} = \frac{\overleftarrow{\lambda}_{T-t}^\theta(\mathbf{X}_t)\overleftarrow{K}_{T-t}^\theta(\mathbf{Y}|\mathbf{X}_t)}{\overrightarrow{\lambda}_t(\mathbf{Y})\overrightarrow{K}_t(\mathbf{X}_t|\mathbf{Y})} \quad \text{for } n_t \neq m \tag{20}$$

Substituting (19) into the second line and (20) into the first line of (18) and using the fact that $\overrightarrow{K}_t(\mathbf{X}_t|\mathbf{Y}) = 0$ for $n_t = m$, we obtain

$$\mathcal{I}_{\text{ISM}}^{\text{J}} = \int_0^T \mathbb{E}_{p_t(\mathbf{x}_t)}[\sum_{m=1\backslash n_t}^N \int_{\mathbf{y}\in\mathbb{R}^{md}} \overrightarrow{\lambda}_t(\mathbf{Y}) \frac{\overleftarrow{\lambda}_{T-t}^\theta(\mathbf{X}_t)\overleftarrow{K}_{T-t}^\theta(\mathbf{Y}|\mathbf{X}_t)}{\overrightarrow{\lambda}_t(\mathbf{Y})\overrightarrow{K}_t(\mathbf{X}_t|\mathbf{Y})}\overrightarrow{K}_t(\mathbf{X}_t|\mathbf{Y})d\mathbf{y}$$
$$- \sum_{m=1}^N \int_{\mathbf{y}\in\mathbb{R}^{md}} \overrightarrow{\lambda}_t(\mathbf{Y}) \frac{\beta_t(\mathbf{Y})}{\beta_t(\mathbf{X}_t)}\delta_{\mathbf{Y}}(\mathbf{X}_t)d\mathbf{y}$$
$$+ \overrightarrow{\lambda}_t(\mathbf{X}_t) \sum_{m=1\backslash n_t}^N \int_{\mathbf{y}\in\mathbb{R}^{md}} \overrightarrow{K}_t(\mathbf{Y}|\mathbf{X}_t)\{\log \beta_t(\mathbf{X}_t) - \log \overleftarrow{\lambda}_{T-t}^\theta(\mathbf{Y})$$
$$- \log \overleftarrow{K}_{T-t}^\theta(\mathbf{X}_t|\mathbf{Y}) + \log \overrightarrow{\lambda}_t(\mathbf{X}_t) + \log \overrightarrow{K}_t(\mathbf{Y}|\mathbf{X}_t)\}d\mathbf{y}$$
$$- \overrightarrow{\lambda}_t(\mathbf{X}_t) \log \beta_t(\mathbf{X}_t)]dt.$$

Hence, we have

$$\mathcal{I}_{\text{ISM}}^{\text{J}} = \int_0^T \mathbb{E}_{p_t(\mathbf{x}_t)}[\overleftarrow{\lambda}_{T-t}^\theta(\mathbf{X}_t) \sum_{m=1\backslash n_t}^N \int_{\mathbf{y}\in\mathbb{R}^{md}} \overleftarrow{K}_{T-t}^\theta(\mathbf{Y}|\mathbf{X}_t)d\mathbf{y} - \overrightarrow{\lambda}_t(\mathbf{X}_t) \frac{\beta_t(\mathbf{X}_t)}{\beta_t(\mathbf{X}_t)}+$$
$$\overrightarrow{\lambda}_t(\mathbf{X}_t) \sum_{m=1\backslash n_t}^N \int_{\mathbf{y}\in\mathbb{R}^{md}} \overrightarrow{K}_t(\mathbf{Y}|\mathbf{X}_t)\{-\log \overleftarrow{\lambda}_{T-t}^\theta(\mathbf{Y}) - \log \overleftarrow{K}_{T-t}^\theta(\mathbf{X}_t|\mathbf{Y})\}d\mathbf{y}]dt + \text{const}.$$

This can be rewritten as
$$\mathcal{I}_{\text{ISM}}^{\text{J}} = \int_0^T \mathbb{E}_{p_t(\mathbf{X}_t)}[\overleftarrow{\lambda}_{T-t}^{\theta}(\mathbf{X}_t) + \overrightarrow{\lambda}_t(\mathbf{X}_t)\mathbb{E}_{\overrightarrow{K}_t(\mathbf{Y}|\mathbf{X}_t)}[-\log\overleftarrow{\lambda}_{T-t}^{\theta}(\mathbf{Y}) - \log\overleftarrow{K}_{T-t}^{\theta}(\mathbf{X}_t|\mathbf{Y})]]\mathrm{d}t + \text{const.}$$

Therefore, we have
$$\mathcal{I}_{\text{ISM}}^{\text{J}} = T\mathbb{E}_{\mathcal{U}(t;0,T)p_t(\mathbf{X}_t)\overrightarrow{K}_t(\mathbf{Y}|\mathbf{X}_t)}[\overleftarrow{\lambda}_{T-t}^{\theta}(\mathbf{X}_t) - \overrightarrow{\lambda}_t(\mathbf{X}_t)\log\overleftarrow{\lambda}_{T-t}^{\theta}(\mathbf{Y}) - \overrightarrow{\lambda}_t(\mathbf{X}_t)\log\overleftarrow{K}_{T-t}^{\theta}(\mathbf{X}_t|\mathbf{Y})] + \text{const.}$$

Finally, using the definition of the forward and backward kernels, i.e. $\overrightarrow{K}_t(\mathbf{Y}|\mathbf{X}_t) = \sum_{i=1}^{n} K^{\text{del}}(i|n)\delta_{\text{del}(\mathbf{X},i)}(\mathbf{Y})$ and $\overleftarrow{K}_{T-t}^{\theta}(\mathbf{X}_t|\mathbf{Y}) = \int_{\mathbf{x}^{\text{add}}}\sum_{i=1}^{n} A_t^{\theta}(\mathbf{x}^{\text{add}},i|\mathbf{Y})\delta_{\text{ins}(\mathbf{Y},\mathbf{x}^{\text{add}},i)}(\mathbf{X}_t)\mathrm{d}\mathbf{x}^{\text{add}}$, we get
$$\mathcal{I}_{\text{ISM}}^{\text{J}} = T\mathbb{E}_{\mathcal{U}(t;0,T)p_t(\mathbf{X}_t)}[\sum_{m=1}^{N}\int_{\mathbf{y}\in\mathbb{R}^{md}}\sum_{i=1}^{n_t} K^{\text{del}}(i|n_t)\delta_{\text{del}(\mathbf{X}_t,i)}(\mathbf{Y})(\overleftarrow{\lambda}_{T-t}^{\theta}(\mathbf{X}_t) - \overrightarrow{\lambda}_t(\mathbf{X}_t)\log\overleftarrow{\lambda}_{T-t}^{\theta}(\mathbf{Y})$$
$$- \overrightarrow{\lambda}_t(\mathbf{X}_t)\log\overleftarrow{K}_{T-t}^{\theta}(\mathbf{X}_t|\mathbf{Y}))\mathrm{d}\mathbf{y}] + \text{const}$$

We get
$$\mathcal{I}_{\text{ISM}}^{\text{J}} = T\mathbb{E}_{\mathcal{U}(t;0,T)p_t(\mathbf{X}_t)K^{\text{del}}(i|n_t)\delta_{\text{del}(\mathbf{X}_t,i)}(\mathbf{Y})}$$
$$[\overleftarrow{\lambda}_{T-t}^{\theta}(\mathbf{X}_t) - \overrightarrow{\lambda}_t(\mathbf{X}_t)\log\overleftarrow{\lambda}_{T-t}^{\theta}(\mathbf{Y}) - \overrightarrow{\lambda}_t(\mathbf{X}_t)\log\overleftarrow{K}_{T-t}^{\theta}(\mathbf{X}_t|\mathbf{Y})] + \text{const.}$$

Therefore, we have
$$\mathcal{I}_{\text{ISM}}^{\text{J}} = T\mathbb{E}_{\mathcal{U}(t;0,T)p_t(\mathbf{X}_t)K^{\text{del}}(i|n_t)\delta_{\text{del}(\mathbf{X}_t,i)}(\mathbf{Y})}$$
$$[\overleftarrow{\lambda}_{T-t}^{\theta}(\mathbf{X}_t) - \overrightarrow{\lambda}_t(\mathbf{X}_t)\log\overleftarrow{\lambda}_{T-t}^{\theta}(\mathbf{Y}) - \overrightarrow{\lambda}_t(\mathbf{X}_t)\log A_{T-t}^{\theta}(\mathbf{x}^{\text{add}},i|\mathbf{Y})] + \text{const.}$$

Putting are expressions for $\mathcal{I}_{\text{ISM}}^{\text{diff}}$ and $\mathcal{I}_{\text{ISM}}^{\text{J}}$ together we obtain
$$\mathcal{I}_{\text{ISM}} = \frac{T}{2}\mathbb{E}[g_t^2\|\nabla\log p_{t|0}(\mathbf{x}_t|n_t,\mathbf{X}_0,M_t) - s_t^{\theta}(\mathbf{X}_t)\|^2]+$$
$$T\mathbb{E}[\overleftarrow{\lambda}_{T-t}^{\theta}(\mathbf{X}_t) - \overrightarrow{\lambda}_t(\mathbf{X}_t)\log\overleftarrow{\lambda}_{T-t}^{\theta}(\mathbf{Y}) - \overrightarrow{\lambda}_t(\mathbf{X}_t)\log A_{T-t}^{\theta}(\mathbf{x}^{\text{add}},i|\mathbf{Y})] + \text{const.}$$

We get that $-\mathcal{I}_{\text{ISM}}$ gives us our evidence lower bound on $\mathbb{E}_{p_{\text{data}}(\mathbf{X}_0)}[\log p_0^{\theta}(\mathbf{X}_0)]$ up to a constant that does not depend on $\theta$. In the main text we have used a time notation such that the backward process runs backwards from $t = T$ to $t = 0$. To align with the notation of time used in the main text we change $T - t$ to $t$ on subscripts for $\overleftarrow{\lambda}_{T-t}^{\theta}$ and $A_{T-t}^{\theta}$. We also will use the fact that $\overrightarrow{\lambda}_t(\mathbf{X}_t)$ depends only on the number of components in $\mathbf{X}_t$, $\overrightarrow{\lambda}_t(\mathbf{X}_t) = \overrightarrow{\lambda}_t(n_t)$.
$$\mathcal{L}(\theta) = -\frac{T}{2}\mathbb{E}[g_t^2\|s_t^{\theta}(\mathbf{X}_t) - \nabla_{\mathbf{x}_t}\log p_{t|0}(\mathbf{x}_t|n_t,\mathbf{X}_0,M_t)\|^2]+$$
$$T\mathbb{E}[-\overleftarrow{\lambda}_t^{\theta}(\mathbf{X}_t) + \overrightarrow{\lambda}_t(n_t)\log\overleftarrow{\lambda}_t^{\theta}(\mathbf{Y}) + \overrightarrow{\lambda}_t(n_t)\log A_t^{\theta}(\mathbf{x}^{\text{add}},i|\mathbf{Y})] + \text{const.}$$

**Tightness of the lower bound**   Now that we have derived the ELBO as in Proposition 2, we show that the maximizers of the ELBO are tight, i.e. that they close the variational gap. We do this by proving the general ELBO presented in [17] has this property and therefore ours, which is a special case of this general ELBO, also has that the optimum parameters close the variational gap.

To state our proposition, we recall the setting of [17]. The forward noising process is denoted $(\mathbf{Y}_t)_{t\geq 0}$ and associated with an infinitesimal generator $\hat{\mathcal{L}}$ its extension $(t,\mathbf{Y}_t)_{t\geq 0}$ is associated with the infinitesimal generator $\mathcal{L}$, i.e. $\mathcal{L} = \partial_t + \hat{\mathcal{L}}$. We also define the score-matching operator $\Phi$ given for any $f$ for which it is defined by
$$\Phi(f) = \mathcal{L}(f)/f - \mathcal{L}(\log(f)).$$

We recall that according to [17, Equation (8)] and under [17, Assumption 1, Assumption2], we have
$$\log p_T(\mathbf{Y}_0) \geq \mathbb{E}[\log p_0(\mathbf{Y}_T) - \int_0^T \mathcal{L}(v/\beta)/(v/\beta) + \mathcal{L}(\log\beta)\mathrm{d}t],$$
with $v_t = p_{T-t}$ for any $t \in [0,T]$. We define the *variational gap* Gap as follows
$$\text{Gap} = \mathbb{E}[\log p_T(\mathbf{Y}_0) - \log p_0(\mathbf{Y}_T) + \int_0^T \mathcal{L}(v/\beta)/(v/\beta) + \mathcal{L}(\log\beta)\mathrm{d}t].$$

In addition, using Itô Formula, we have that $\log v_T(\mathbf{Y}_T) - \log v_0(\mathbf{Y}_0) = \int_0^T \mathcal{L}(v)\mathrm{d}t$. Assuming that $\mathbb{E}[|\log v_T(\mathbf{Y}_T) - \log v_0(\mathbf{Y}_0)|] < +\infty$, we get
$$\text{Gap} = \mathbb{E}[\int_0^T -\mathcal{L}(\log v) + \mathcal{L}(v/\beta)/(v/\beta) + \mathcal{L}(\log\beta)\mathrm{d}t] = \mathbb{E}[\int_0^T \Phi(v/\beta)\mathrm{d}t].$$

In particular, using [17, Proposition 1], we get that Gap $\geq 0$ and Gap $= 0$ if and only if $\beta \propto v$. In addition, the ELBO is maximized if and only if $\beta \propto v$, see [17, Equation 10] and the remark that follows. Therefore, we have that: if we maximize the ELBO then the ELBO is tight. Combining this with the fact that the ELBO is maximized at the time reversal [17], then we have that when our jump diffusion parameters match the time reversal, our variational gap is 0.

**Other approaches.** Another way to derive the ELBO is to follow the steps of [15] directly, since [17] is a general framework extending this approach. The key formulae to derive the result and the ELBO is 1) a Feynman-Kac formula 2) a Girsanov formula. In the case of jump diffusions (with jump in $\mathbb{R}^d$) a Girsanov formula has been established by [22]. Extending this result to one-point compactification space would allow us to prove directly Proposition 2 without having to rely on the general framework of [17].

## A.4 Proof of Proposition 3

We start by recalling the form for the time reversal given in Proposition 1

$$\overleftarrow{\lambda}^*_t(\mathbf{X}) = \overrightarrow{\lambda}_t(n+1) \sum_{i=1}^{n+1} K^{\text{del}}(i|n+1) \int_{\mathbf{y}^{\text{add}}} p_t(\text{ins}(\mathbf{X}, \mathbf{y}^{\text{add}}, i)) \mathrm{d}\mathbf{y}^{\text{add}} / p_t(\mathbf{X}).$$

We then introduce a marginalization over $\mathbf{X}_0$

$$\overleftarrow{\lambda}^*_t(\mathbf{X}) = \overrightarrow{\lambda}_t(n+1) \sum_{i=1}^{n+1} K^{\text{del}}(i|n+1) \int_{\mathbf{y}^{\text{add}}} \sum_{n_0} \int_{\mathbf{x}_0} p_{0,t}(\mathbf{X}_0, \text{ins}(\mathbf{X}, \mathbf{y}^{\text{add}}, i)) \mathrm{d}\mathbf{x}_0 \mathrm{d}\mathbf{y}^{\text{add}} / p_t(\mathbf{X})$$

$$= \overrightarrow{\lambda}_t(n+1) \sum_{i=1}^{n+1} K^{\text{del}}(i|n+1) \int_{\mathbf{y}^{\text{add}}} \sum_{n_0} \int_{\mathbf{x}_0} \frac{p_0(\mathbf{X}_0)}{p_t(\mathbf{X})} p_{t|0}(\text{ins}(\mathbf{X}, \mathbf{y}^{\text{add}}, i)|\mathbf{X}_0) \mathrm{d}\mathbf{x}_0 \mathrm{d}\mathbf{y}^{\text{add}}$$

$$= \overrightarrow{\lambda}_t(n+1) \sum_{i=1}^{n+1} K^{\text{del}}(i|n+1) \int_{\mathbf{y}^{\text{add}}} \sum_{n_0} \int_{\mathbf{x}_0} \frac{p_{0|t}(\mathbf{X}_0|\mathbf{X})}{p_{t|0}(\mathbf{X}|\mathbf{X}_0)} p_{t|0}(\text{ins}(\mathbf{X}, \mathbf{y}^{\text{add}}, i)|\mathbf{X}_0) \mathrm{d}\mathbf{x}_0 \mathrm{d}\mathbf{y}^{\text{add}}$$

$$= \overrightarrow{\lambda}_t(n+1) \sum_{i=1}^{n+1} K^{\text{del}}(i|n+1) \int_{\mathbf{y}^{\text{add}}} \sum_{n_0} \int_{\mathbf{x}_0} \frac{p_{0|t}(n_0|\mathbf{X}) p_{0|t}(\mathbf{x}_0|\mathbf{X}, n_0)}{p_{t|0}(n|\mathbf{X}_0) p_{t|0}(\mathbf{x}|\mathbf{X}_0, n)} \times$$

$$p_{t|0}(n+1|\mathbf{X}_0) p_{t|0}(\mathbf{z}(\mathbf{X}, \mathbf{y}^{\text{add}}, i)|\mathbf{X}_0, n+1) \mathrm{d}\mathbf{x}_0 \mathrm{d}\mathbf{y}^{\text{add}}$$

where $(n+1, \mathbf{z}(\mathbf{X}, \mathbf{y}^{\text{add}}, i)) = \text{ins}(\mathbf{X}, \mathbf{y}^{\text{add}}, i)$. Now using the fact the forward component deletion process does not depend on $\mathbf{x}_0$, only $n_0$, we have $p_{t|0}(n|\mathbf{X}_0) = p_{t|0}(n|n_0)$ and $p_{t|0}(n+1|\mathbf{X}_0) = p_{t|0}(n+1|n_0)$. Using this result, we get

$$\overleftarrow{\lambda}^*_t(\mathbf{X}) = \overrightarrow{\lambda}_t(n+1) \sum_{n_0} \{ \frac{p_{t|0}(n+1|n_0)}{p_{t|0}(n|n_0)} p_{0|t}(n_0|\mathbf{X}) \times$$

$$\int_{\mathbf{x}_0} \frac{\sum_{i=1}^{n+1} K^{\text{del}}(i|n+1) \int_{\mathbf{y}^{\text{add}}} p_{t|0}(\mathbf{z}(\mathbf{X}, \mathbf{y}^{\text{add}}, i)|\mathbf{X}_0, n+1) \mathrm{d}\mathbf{y}^{\text{add}}}{p_{t|0}(\mathbf{x}|\mathbf{X}_0, n)} p_{0|t}(\mathbf{x}_0|\mathbf{X}, n_0) \mathrm{d}\mathbf{x}_0 \}. \quad (21)$$

We now focus on the probability ratio within the integral over $\mathbf{x}_0$. We will show that this ratio is 1. We start with the numerator, introducing a marginalization over possible mask variables between $\mathbf{X}_0$ and $(n+1, \mathbf{z})$, denoted $M^{(n+1)}$ with $M^{(n+1)}$ having $n+1$ ones and $n_0 - (n+1)$ zeros.

$$\sum_{i=1}^{n+1} K^{\text{del}}(i|n+1) \int_{\mathbf{y}^{\text{add}}} p_{t|0}(\mathbf{z}(\mathbf{X}, \mathbf{y}^{\text{add}}, i)|\mathbf{X}_0, n+1) \mathrm{d}\mathbf{y}^{\text{add}}$$

$$= \sum_{i=1}^{n+1} K^{\text{del}}(i|n+1) \sum_{M^{(n+1)}} \int_{\mathbf{y}^{\text{add}}} p_{t|0}(M^{(n+1)}, \mathbf{z}(\mathbf{X}, \mathbf{y}^{\text{add}}, i)|\mathbf{X}_0, n+1) \mathrm{d}\mathbf{y}^{\text{add}}$$

$$= \sum_{M^{(n+1)}} \sum_{i=1}^{n+1} K^{\text{del}}(i|n+1) p_{t|0}(M^{(n+1)}|\mathbf{X}_0, n+1) \int_{\mathbf{y}^{\text{add}}} p_{t|0}(\mathbf{z}(\mathbf{X}, \mathbf{y}^{\text{add}}, i)|\mathbf{X}_0, n+1, M^{(n+1)}) \mathrm{d}\mathbf{y}^{\text{add}}$$

Now, for our forward process we have

$$p_{t|0}(\mathbf{z}(\mathbf{X}, \mathbf{y}^{\text{add}}, i)|\mathbf{X}_0, n+1, M^{(n+1)}) = \prod_{j=1}^{n+1} \mathcal{N}(\mathbf{z}^{(j)}; \sqrt{\alpha_t} M^{(n+1)}(\mathbf{X}_0)^j, (1-\alpha_t) I_d)$$

where $\mathbf{z}$ is shorthand for $\mathbf{z}(\mathbf{X}, \mathbf{y}^{\text{add}}, i)$, $\mathbf{z}^{(j)}$ is the vector in $\mathbb{R}^d$ for the $j$th component of $\mathbf{z}$ and $M^{(n+1)}(\mathbf{X}_0)^j$ is the vector in $\mathbb{R}^d$ corresponding to the component in $\mathbf{X}_0$ corresponding to the $j$th one in the $M^{(n+1)}$ mask. Integrating out $\mathbf{y}^{\text{add}}$ we have

$$\int_{\mathbf{y}^{\text{add}}} p_{t|0}(\mathbf{z}(\mathbf{X}, \mathbf{y}^{\text{add}}, i)|\mathbf{X}_0, n+1, M^{(n+1)}) \mathrm{d}\mathbf{y}^{\text{add}} = \prod_{j=1}^{n} \mathcal{N}(\mathbf{x}^{(j)}; \sqrt{\alpha_t} M^{(n+1)\backslash i}(\mathbf{X}_0)^j, (1-\alpha_t) I_d),$$

where $M^{(n+1)\backslash i}$ denotes a mask variable obtained by setting the $i$th one of $M^{(n+1)}$ to zero. Hence, we have

$$\sum_{i=1}^{n+1} K^{\text{del}}(i|n+1) \int_{\mathbf{y}^{\text{add}}} p_{t|0}(\mathbf{z}(\mathbf{X}, \mathbf{y}^{\text{add}}, i)|\mathbf{X}_0, n+1) \mathrm{d}\mathbf{y}^{\text{add}}$$

$$= \sum_{M^{(n+1)}} \sum_{i=1}^{n+1} K^{\text{del}}(i|n+1) p_{t|0}(M^{(n+1)}|\mathbf{X}_0, n+1)$$

$$\prod_{j=1}^{n} \mathcal{N}(\mathbf{x}^{(j)}; \sqrt{\alpha_t} M^{(n+1)\backslash i}(\mathbf{X}_0)^j, (1-\alpha_t) I_d). \quad (22)$$

We now re-write the denominator from (21) introducing a marginalization over mask variables, $M^{(n)}$

$$p_{t|0}(\mathbf{x}|\mathbf{X}_0, n) = \sum_{M^{(n)}} p_{t|0}(M^{(n)}|\mathbf{X}_0, n) p_{t|0}(\mathbf{x}|M^{(n)}, \mathbf{X}_0, n). \quad (23)$$

We use the following recursion for the probabilities assigned to mask variables

$$p_{t|0}(M^{(n)}|\mathbf{X}_0, n) = \sum_{M^{(n+1)}} \sum_{i=1}^{n+1} \mathbb{I}\{M^{(n+1)\backslash i} = M^{(n)}\} K^{\text{del}}(i|n+1) p_{t|0}(M^{(n+1)}|\mathbf{X}_0, n+1).$$

Substituting this into (23) gives

$$p_{t|0}(\mathbf{x}|\mathbf{X}_0, n) = \sum_{M^{(n)}} \sum_{M^{(n+1)}} \sum_{i=1}^{n+1} \mathbb{I}\{M^{(n+1)\backslash i} = M^{(n)}\} K^{\text{del}}(i|n+1) \times$$
$$p_{t|0}(M^{(n+1)}|\mathbf{X}_0, n+1) p_{t|0}(\mathbf{x}|M^{(n)}, \mathbf{X}_0, n)$$
$$= \sum_{M^{(n)}} \sum_{M^{(n+1)}} \sum_{i=1}^{n+1} \mathbb{I}\{M^{(n+1)\backslash i} = M^{(n)}\} K^{\text{del}}(i|n+1)$$
$$\times p_{t|0}(M^{(n+1)}|\mathbf{X}_0, n+1) \prod_{j=1}^{n} \mathcal{N}(\mathbf{x}^{(j)}; \sqrt{\alpha_t} M^{(n)}(\mathbf{X}_0)^j, (1-\alpha_t) I_d)$$
$$= \sum_{M^{(n+1)}} \sum_{i=1}^{n+1} K^{\text{del}}(i|n+1) p_{t|0}(M^{(n+1)}|\mathbf{X}_0, n+1)$$
$$\times \prod_{j=1}^{n} \mathcal{N}(\mathbf{x}^{(j)}; \sqrt{\alpha_t} M^{(n+1)\backslash i}(\mathbf{X}_0)^j, (1-\alpha_t) I_d).$$

By comparing with (22), we can see that

$$p_{t|0}(\mathbf{x}|\mathbf{X}_0, n) = \sum_{i=1}^{n+1} K^{\text{del}}(i|n+1) \int_{\mathbf{y}^{\text{add}}} p_{t|0}(\mathbf{z}(\mathbf{X}, \mathbf{y}^{\text{add}}, i)|\mathbf{X}_0, n+1) \mathrm{d}\mathbf{y}^{\text{add}}.$$

This shows that the probability ratio in (21) is 1. Therefore, we have

$$\overleftarrow{\lambda}_t^*(\mathbf{X}) = \overrightarrow{\lambda}_t(n+1) \sum_{n_0} \left\{ \frac{p_{t|0}(n+1|n_0)}{p_{t|0}(n|n_0)} p_{0|t}(n_0|\mathbf{X}) \int_{\mathbf{x}_0} p_{0|t}(\mathbf{x}_0|\mathbf{X}, n_0) \mathrm{d}\mathbf{x}_0 \right\}$$
$$= \overrightarrow{\lambda}_t(n+1) \sum_{n_0} \frac{p_{t|0}(n+1|n_0)}{p_{t|0}(n|n_0)} p_{0|t}(n_0|\mathbf{X}),$$

which concludes the proof.

$p_{t|0}(n|n_0)$ can be analytically calculated when $\overrightarrow{\lambda}_t(n)$ is of a simple enough form. When $\overrightarrow{\lambda}_t(n)$ does not depend on $n$ then the dimension deletion process simply becomes a time inhomogeneous Poisson process. Therefore, we would have

$$p_{t|0}(n|n_0) = \frac{(\int_0^t \overrightarrow{\lambda}_s \mathrm{d}s)^{n_0-n}}{(n_0-n)!} \exp(-\int_0^t \overrightarrow{\lambda}_s \mathrm{d}s).$$

In our experiments we set $\overrightarrow{\lambda}_t(n=1) = 0$ to stop the dimension deletion process when we reach a single component. If we have $\overrightarrow{\lambda}_t(n) = \overrightarrow{\lambda}_t(m)$ for all $n, m > 1$ then we can still use the time inhomogeneous Poisson process formula for $n > 1$ and find the probability for $n = 1$, $p_{t|0}(n=1|n_0)$ by requiring $p_{t|0}(n|n_0)$ to be a valid normalized distribution. Therefore, for the case that $\overrightarrow{\lambda}_t(n) = \overrightarrow{\lambda}_t(m)$ for all $n, m > 1$ and $\overrightarrow{\lambda}_t(n=1) = 0$, we have

$$p_{t|0}(n|n_0) = \begin{cases} \frac{(\int_0^t \overrightarrow{\lambda}_s \mathrm{d}s)^{n_0-n}}{(n_0-n)!} \exp(-\int_0^t \overrightarrow{\lambda}_s \mathrm{d}s) & 1 < n \le n_0 \\ 1 - \sum_{m=2}^{n_0} \frac{(\int_0^t \overrightarrow{\lambda}_s \mathrm{d}s)^{n_0-m}}{(n_0-m)!} \exp(-\int_0^t \overrightarrow{\lambda}_s \mathrm{d}s) & n = 1 \end{cases}$$

In cases where $\overrightarrow{\lambda}_t(n)$ depends on $n$ not just for $n = 1$, $p_{t|0}(n|n_0)$ can become more difficult to calculate analytically. However, since the probability distributions are all 1-dimensional over $n$, it is very cheap to simply simulate the forward dimension deletion process many times and empirically estimate $p_{t|0}(n|n_0)$ although we do not need to do this for our experiments.

### A.5 The Objective is Maximized at the Time Reversal

In this section, we analyze the objective $\mathcal{L}(\theta)$ as a standalone object and determine the optimum values for $s_t^\theta$, $\overleftarrow{\lambda}_t^\theta$ and $A_t^\theta$ directly. This is in order to gain intuition directly into the learning signal of $\mathcal{L}(\theta)$ without needing to refer to stochastic process theory.

The definition of $\mathcal{L}(\theta)$ as in the main text is

$$\mathcal{L}(\theta) = -\frac{T}{2} \mathbb{E}[g_t^2 \| s_t^\theta(\mathbf{X}_t) - \nabla_{\mathbf{x}_t} \log p_{t|0}(\mathbf{x}_t|\mathbf{X}_0, n_t, M_t) \|^2] +$$
$$T \mathbb{E}[-\overleftarrow{\lambda}_t^\theta(\mathbf{X}_t) + \overrightarrow{\lambda}_t(n_t) \log \overleftarrow{\lambda}_t^\theta(\mathbf{Y}) + \overrightarrow{\lambda}_t(n_t) \log A_t^\theta(\mathbf{x}_t^{\text{add}}, i|\mathbf{Y})] + C.$$

with the expectations taken over $\mathcal{U}(t; 0, T) p_{0,t}(\mathbf{X}_0, \mathbf{X}_t, M_t) K^{\text{del}}(i|n_t) \delta_{\text{del}(\mathbf{X}_t, i)}(\mathbf{Y})$.

**Continuous optimum.** We start by analysing the objective for $s_t^\theta$. This part of $\mathcal{L}(\theta)$ can be written as

$$-\tfrac{1}{2}\int_0^T g_t^2 \mathbb{E}_{p_{0,t}(\mathbf{X}_0,\mathbf{X}_t,M_t)}[\|s_t^\theta(\mathbf{X}_t) - \nabla_{\mathbf{x}_t}\log p_{t|0}(\mathbf{x}_t|\mathbf{X}_0,n_t,M_t)\|^2]\mathrm{d}t$$

We now use the fact that the function that minimizes an $L_2$ regression problem $\min_f \mathbb{E}_{p(x,y)}[\|f(x) - y\|^2]$ is the conditional expectation of the target $f^*(x) = \mathbb{E}_{p(y|x)}[y]$. Therefore the optimum value for $s_t^\theta(\mathbf{X}_t)$ is

$$
\begin{aligned}
s_t^*(\mathbf{X}_t) &= \mathbb{E}_{p(M_t,\mathbf{X}_0|\mathbf{X}_t)}[\nabla_{\mathbf{x}_t}\log p_{t|0}(\mathbf{x}_t|\mathbf{X}_0,n_t,M_t)] \\
&= \sum_{M_t}\sum_{n_0=1}^N \int_{\mathbf{x}_0\in\mathbb{R}^{n_0 d}} p(M_t,n_0,\mathbf{x}_0|\mathbf{X}_t)\nabla_{\mathbf{x}_t}\log p_{t|0}(\mathbf{x}_t|\mathbf{x}_0,n_0,n_t,M_t)\mathrm{d}\mathbf{x}_0 \\
&= \sum_{M_t}\sum_{n_0=1}^N \int_{\mathbf{x}_0\in\mathbb{R}^{n_0 d}} \frac{p(M_t,n_0,\mathbf{x}_0|\mathbf{X}_t)}{p_{t|0}(\mathbf{x}_t|\mathbf{x}_0,n_0,n_t,M_t)}\nabla_{\mathbf{x}_t} p_{t|0}(\mathbf{x}_t|\mathbf{x}_0,n_0,n_t,M_t)\mathrm{d}\mathbf{x}_0 \\
&= \sum_{M_t}\sum_{n_0=1}^N \int_{\mathbf{x}_0\in\mathbb{R}^{n_0 d}} \frac{p(\mathbf{x}_0,n_0,n_t,M_t)}{p(n_t,\mathbf{x}_t)}\nabla_{\mathbf{x}_t} p_{t|0}(\mathbf{x}_t|\mathbf{x}_0,n_0,n_t,M_t)\mathrm{d}\mathbf{x}_0 \\
&= \frac{1}{p(n_t,\mathbf{x}_t)}\sum_{M_t}\sum_{n_0=1}^N \int_{\mathbf{x}_0\in\mathbb{R}^{n_0 d}} \nabla_{\mathbf{x}_t} p(\mathbf{x}_t,\mathbf{x}_0,n_0,n_t,M_t)\mathrm{d}\mathbf{x}_0 \\
&= \frac{1}{p(n_t,\mathbf{x}_t)}\nabla_{\mathbf{x}_t}\sum_{M_t}\sum_{n_0=1}^N \int_{\mathbf{x}_0\in\mathbb{R}^{n_0 d}} p(\mathbf{x}_t,\mathbf{x}_0,n_0,n_t,M_t)\mathrm{d}\mathbf{x}_0 \\
&= \frac{1}{p(n_t,\mathbf{x}_t)}\nabla_{\mathbf{x}_t} p(\mathbf{x}_t,n_t) = \nabla_{\mathbf{x}_t}\log p(\mathbf{X}_t).
\end{aligned}
$$

Therefore, the optimum value for $s_t^\theta(\mathbf{X}_t)$ is $\nabla_{\mathbf{x}_t}\log p(\mathbf{X}_t)$ which is the value that gives $\overleftarrow{\mathbf{b}}_t$ to be the time reversal of $\overrightarrow{\mathbf{b}}_t$ as stated in Proposition 1.

**Jump rate optimum.** The learning signal for $\overleftarrow{\lambda}_t^\theta$ comes from these two terms in $\mathcal{L}(\theta)$

$$T\mathbb{E}[-\overleftarrow{\lambda}_t^\theta(\mathbf{X}_t) + \overrightarrow{\lambda}_t(n_t)\log\overleftarrow{\lambda}_t^\theta(\mathbf{Y})] \tag{24}$$

This expectation is maximized when for each test input $\mathbf{Z}$ and test time $t$, we have the following expression maximized

$$-p_t(\mathbf{Z})\overleftarrow{\lambda}_t^\theta(\mathbf{Z}) + \sum_{i=}^{n_z+1}\int_{\mathbf{y}^{\mathrm{add}}} p_t(\mathrm{ins}(\mathbf{Z},\mathbf{y}^{\mathrm{add}},i))K^{\mathrm{del}}(i|n_z+1)\mathrm{d}\mathbf{y}^{\mathrm{add}} \times \overrightarrow{\lambda}_t(n_z+1)\log\overleftarrow{\lambda}_t^\theta(\mathbf{Z}),$$

because $p_t(\mathbf{Z})$ is the probability $\mathbf{Z}$ gets drawn as a full sample from the forward process and $\sum_{i=}^{n_z+1}\int_{\mathbf{y}^{\mathrm{add}}} p_t(\mathrm{ins}(\mathbf{Z},\mathbf{y}^{\mathrm{add}},i))K^{\mathrm{del}}(i|n_z+1)\mathrm{d}\mathbf{y}^{\mathrm{add}}$ is the probability that a sample one component bigger than $\mathbf{Z}$ gets drawn from the forward process and then a component is deleted to get to $\mathbf{Z}$. Therefore the first probability is the probability that test input $\mathbf{Z}$ and test time $t$ appear as the first term in (24) whereas the second probability is the probability that test input $\mathbf{Z}$ and test time $t$ appear as the second term in (24).

We now use the fact that, for constants $b$ and $c$,

$$\mathrm{argmax}_a \quad -ba + c\log a = \tfrac{c}{b}.$$

We therefore have the optimum $\overleftarrow{\lambda}_t^\theta(\mathbf{Z})$ as

$$\overleftarrow{\lambda}_t^*(\mathbf{Z}) = \overrightarrow{\lambda}_t(n_z+1)\frac{\sum_{i=}^{n_z+1}\int_{\mathbf{y}^{\mathrm{add}}} p_t(\mathrm{ins}(\mathbf{Z},\mathbf{y}^{\mathrm{add}},i))K^{\mathrm{del}}(i|n_z+1)\mathrm{d}\mathbf{y}^{\mathrm{add}}}{p_t(\mathbf{Z})}$$

which is the form for the time-reversal given in Proposition (1).

**Jump kernel optimum.** Finally, we analyse the part of $\mathcal{L}(\theta)$ for learning $A_t^\theta(\mathbf{x}_t^{\mathrm{add}},i|\mathbf{Y})$,

$$
\begin{aligned}
&T\mathbb{E}[\overrightarrow{\lambda}_t(n_t)\log A_t^\theta(\mathbf{x}_t^{\mathrm{add}},i|\mathbf{Y})] \\
&= \int_0^T \mathbb{E}_{p_t(\mathbf{X}_t)K^{\mathrm{del}}(i|n_t)\delta_{\mathrm{del}(\mathbf{X}_t,i)}(\mathbf{Y})}[\overrightarrow{\lambda}_t(n_t)\log A_t^\theta(\mathbf{x}_t^{\mathrm{add}},i|\mathbf{Y})]\mathrm{d}t \\
&= \int_0^T \mathbb{E}_{p_t(n_t)}[\overrightarrow{\lambda}_t(n_t)\mathbb{E}_{p_t(\mathbf{x}_t|n_t)K^{\mathrm{del}}(i|n_t)\delta_{\mathrm{del}(\mathbf{X}_t,i)}(\mathbf{Y})}[\log A_t^\theta(\mathbf{x}_t^{\mathrm{add}},i|\mathbf{Y})]]\mathrm{d}t.
\end{aligned}
$$

We now re-write the joint probability distribution that the inner expectation is taken with respect to,

$$p_t(\mathbf{x}_t|n_t)K^{\mathrm{del}}(i|n_t)\delta_{\mathrm{del}(\mathbf{X}_t,i)}(\mathbf{Y}) = \tilde{p}(\mathbf{Y}|n_t)p(\mathbf{x}_t^{\mathrm{add}},i|\mathbf{Y})\delta_{\mathbf{y}}(\mathbf{x}_t^{\mathrm{base}}).$$

with

$$\tilde{p}(\mathbf{Y}|n_t) = \sum_{i=1}^{n_t} \int_{\mathbf{x}_t} p_t(\mathbf{x}_t|n_t) K^{\text{del}}(i|n_t) \delta_{\text{del}(\mathbf{x}_t, i)}(\mathbf{Y}) \mathrm{d}\mathbf{x}_t,$$

and

$$p(\mathbf{x}_t^{\text{add}}, i|\mathbf{Y}) \propto p_t(\mathbf{x}_t|n_t) K^{\text{del}}(i|n_t),$$

and $\mathbf{x}_t^{\text{base}} \in \mathbb{R}^{(n_t-1)d}$ referring to the $n_t - 1$ components of $\mathbf{x}_t$, that are not $\mathbf{x}_t^{\text{add}}$ i.e. $\mathbf{X}_t = \text{ins}((\mathbf{x}_t^{\text{base}}, n_t - 1), \mathbf{x}_t^{\text{add}}, i)$. We then have

$$T\mathbb{E}[\overrightarrow{\lambda}_t(n_t) \log A_t^{\theta}(\mathbf{x}_t^{\text{add}}, i|\mathbf{Y})]$$

$$= \int_0^T \mathbb{E}_{p_t(n_t)}[\overrightarrow{\lambda}_t(n_t) \mathbb{E}_{\tilde{p}(\mathbf{Y}|n_t) p(\mathbf{x}_t^{\text{add}}, i|\mathbf{Y}) \delta_{\mathbf{y}}(\mathbf{x}_t^{\text{base}})}[\log A_t^{\theta}(\mathbf{x}_t^{\text{add}}, i|\mathbf{Y})]] \mathrm{d}t$$

$$= \int_0^T \mathbb{E}_{p_t(n_t)}[\overrightarrow{\lambda}_t(n_t) \mathbb{E}_{\tilde{p}(\mathbf{Y}|n_t) p(\mathbf{x}_t^{\text{add}}, i|\mathbf{Y}) \delta_{\mathbf{y}}(\mathbf{x}_t^{\text{base}})}[\log A_t^{\theta}(\mathbf{x}_t^{\text{add}}, i|\mathbf{Y})]] \mathrm{d}t$$

$$\qquad - \int_0^T \mathbb{E}_{p_t(n_t)}[\overrightarrow{\lambda}_t(n_t) \mathbb{E}_{\tilde{p}(\mathbf{Y}|n_t) p(\mathbf{x}_t^{\text{add}}, i|\mathbf{Y}) \delta_{\mathbf{y}}(\mathbf{x}_t^{\text{base}})}[\log p(\mathbf{x}_t^{\text{add}}, i|\mathbf{Y})]] \mathrm{d}t + \text{const}$$

$$= \int_0^T \mathbb{E}_{p_t(n_t)}[\overrightarrow{\lambda}_t(n_t) \mathbb{E}_{\tilde{p}(\mathbf{Y}|n_t) \delta_{\mathbf{y}}(\mathbf{x}_t^{\text{base}})}[-\text{KL}(p(\mathbf{x}_t^{\text{add}}, i|\mathbf{Y}) \,||\, A_t^{\theta}(\mathbf{x}_t^{\text{add}}, i|\mathbf{Y}))]] \mathrm{d}t + \text{const}.$$

Therefore, the optimum $A_t^{\theta}(\mathbf{x}_t^{\text{add}}, i|\mathbf{Y})$ which maximizes this part of $\mathcal{L}(\theta)$ is

$$A_t^*(\mathbf{x}_t^{\text{add}}, i|\mathbf{Y}) = p(\mathbf{x}_t^{\text{add}}, i|\mathbf{Y}) \propto p_t(\mathbf{X}_t) K^{\text{del}}(i|n_t).$$

which is the same form as given in Proposition 1.

## B  Training Objective

We estimate our objective $\mathcal{L}(\theta)$ by taking minibatches from the expectation $\mathcal{U}(t; 0, T) p_{0,t}(\mathbf{X}_0, \mathbf{X}_t, M_t) K^{\text{del}}(i|n_t) \delta_{\text{del}(\mathbf{X}_t, i)}(\mathbf{Y})$. We first sample $t \sim \mathcal{U}(t; 0, T)$ and then take samples from our dataset $\mathbf{X}_0 \sim p_{\text{data}}(\mathbf{X}_0)$. In order to sample $p_{t|0}(\mathbf{X}_t, M_t|\mathbf{X}_0)$ we need to both add noise, delete dimensions and sample a mask variable. Since the Gaussian noising process is isotropic, we can add a suitable amount of noise to all dimensions of $\mathbf{X}_0$ and then delete dimensions of that noised full dimensional value. More specifically, we first sample $\tilde{\mathbf{X}}_t = (n_0, \tilde{\mathbf{x}}_t)$ with $\tilde{\mathbf{x}}_t \sim \mathcal{N}(\tilde{\mathbf{x}}_t; \sqrt{\alpha_t}\mathbf{x}_0, (1 - \alpha_t)I_{n_0 d})$ for $\alpha_t = \exp\left(-\int_0^t \beta(s)\mathrm{d}s\right)$ using the analytic forward equations for the VP-SDE derived in [3]. Then we sample the number of dimensions to delete. This is simple to do when our rate function is independent of $n$ except for the case when $n = 1$ at which it is zero. We simply sample a Poisson random variable with mean parameter $\int_0^t \overrightarrow{\lambda}_s \mathrm{d}s$ and then clamp its value such that the maximum number of possible components that are deleted is $n_0 - 1$. This gives the appropriate distribution over $n$, $p_{t|0}(n|n_0)$ as given in Section A.4. To sample which dimensions are deleted, we can sample $K^{\text{del}}(i_1|n_0) K^{\text{del}}(i_2|n_0 - 1) \ldots K^{\text{del}}(i_{n_0 - n_t}|n_t + 1)$ from which we can create the mask $M_t$ and apply it to $\tilde{\mathbf{X}}_t$ to obtain $\mathbf{X}_t$, $\mathbf{X}_t = M_t(\tilde{\mathbf{X}}_t)$. When $K^{\text{del}}(i|n) = 1/n$ this is especially simple to do by simply randomly permuting the components of $\tilde{\mathbf{X}}_t$, and then removing the final $n_0 - n_t$ components.

As is typically done in standard diffusion models, we parameterize $s_t^{\theta}$ in terms of a noise prediction network that predicts $\epsilon$ where $\mathbf{x}_t = \sqrt{\alpha_t} M_t(\mathbf{x}_0) + \sqrt{1 - \alpha_t}\epsilon$, $\epsilon \sim \mathcal{N}(0, I_{n_t d})$. We then re-weight the score loss in time such that we have a uniform weighting in time rather than the 'likelihood weighting' with $g_t^2$ [3, 21]. Our objective to learn $s_t^{\theta}$ then becomes

$$-\mathbb{E}_{\mathcal{U}(t; 0, T) p_{\text{data}}(\mathbf{X}_0) p(M_t, n_t|\mathbf{X}_0) \mathcal{N}(\epsilon; 0, I_{n_t d})} \left[\|\epsilon_t^{\theta}(\mathbf{X}_t) - \epsilon\|^2\right]$$

with $\mathbf{x}_t = \sqrt{\alpha_t} M_t(\mathbf{x}_0) + \sqrt{1 - \alpha_t}\epsilon$, $s_t^{\theta}(\mathbf{X}_t) = \frac{-1}{\sqrt{1 - \alpha_t}} \epsilon_t^{\theta}(\mathbf{X}_t)$.

Further, by using the parameterization given in Proposition 3, we can directly supervise the value of $p_{0|t}^{\theta}(n_0|\mathbf{X}_t)$ by adding an extra term to our objective. We can treat the learning of $p_{0|t}^{\theta}(n_0|\mathbf{X}_t)$ as a standard prediction task where we aim to predict $n_0$ given access to $\mathbf{X}_t$. A standard objective for learning $p_{0|t}^{\theta}(n_0|\mathbf{X}_t)$ is then the cross entropy

$$\max_{\theta} \quad \mathbb{E}_{p_{0,t}(\mathbf{X}_0, \mathbf{X}_t)} \left[\log p_{0|t}^{\theta}(n_0|\mathbf{X}_t)\right]$$

Our augmented objective then becomes

$$\tilde{\mathcal{L}}(\theta) = T\mathbb{E}[-\frac{1}{2}\|\epsilon_t^\theta(\mathbf{X}_t)-\epsilon\|^2 - \overleftarrow{\lambda}_t^\theta(\mathbf{X}_t) + \overrightarrow{\lambda}_t(n_t)\log\overleftarrow{\lambda}_t^\theta(\mathbf{Y}) + \overrightarrow{\lambda}_t(n_t)\log A_t^\theta(\mathbf{x}^{\text{add}}, i|\mathbf{Y}) + \gamma\log p_{0|t}^\theta(n_0|\mathbf{X}_t)] \tag{25}$$

where the expectation is taken with respect to

$$\mathcal{U}(t; 0, T)p_{\text{data}}(\mathbf{X}_0)p(M_t, n_t|\mathbf{X}_0)\mathcal{N}(\epsilon; 0, I_{n_t d})K^{\text{del}}(i|n_t)\delta_{\text{del}(\mathbf{X}_t, i)}(\mathbf{Y})$$

where $\mathbf{x}_t = \sqrt{\alpha_t}M_t(\mathbf{x}_0) + \sqrt{1-\alpha_t}\epsilon$ and $\gamma$ is a loss weighting term for the cross entropy loss.

## C Trans-Dimensional Diffusion Guidance

To guide an unconditionally trained model such that it generates datapoints consistent with conditioning information, we use the reconstruction guided sampling approach introduced in [9]. Our conditioning information will be the values for some of the components of $\mathbf{X}_0$, and thus the guidance should guide the generative process such that the rest of the components of the generated datapoint are consistent with those observed components. Following the notation of [9], we denote the observed components as $\mathbf{x}^a \in \mathbb{R}^{n_a d}$ and the components to be generated as $\mathbf{x}^b \in \mathbb{R}^{n_b d}$. Our trained score function $s_t^\theta(\mathbf{X}_t)$ approximates $\nabla_{\mathbf{x}_t}\log p_t(\mathbf{X}_t)$ whereas we would like the score to approximate $\nabla_{\mathbf{x}_t}\log p_t(\mathbf{X}_t|\mathbf{x}_0^a)$. In order to do this, we will need to augment our unconditional score $s_t^\theta(\mathbf{X}_t)$ such that it incorporates the conditioning information.

We first focus on the dimensions of the score vector corresponding to $\mathbf{x}^a$. These can be calculated analytically from the forward process

$$\nabla_{\mathbf{x}_t^a}\log p(\mathbf{X}_t|\mathbf{x}_0^a) = \nabla_{\mathbf{x}_t^a}\log p_{t|0}(\mathbf{x}_t^a|\mathbf{x}_0^a, n_t)$$

with $p_{t|0}(\mathbf{x}_t^a|\mathbf{x}_0^a, n_t) = \mathcal{N}(\mathbf{x}_t^a; \sqrt{\alpha_t}\mathbf{x}_0^a, (1-\alpha_t)I_{n_a d})$. Note that we assume a correspondence between $\mathbf{x}_t^a$ and $\mathbf{x}_0^a$. For example, in video if we condition on the first and last frame, we assume that the first and last frame of the current noisy $\mathbf{x}_t$ correspond to $\mathbf{x}_0^a$ and guide them towards their observed values. For molecules, the point cloud is permutation invariant and so we can simply assume the first $n_a$ components of $\mathbf{x}_t$ correspond to $\mathbf{x}_0^a$ and guide them to their observed values.

Now we analyse the dimensions of the score vector corresponding to $\mathbf{x}^b$. We split the score as

$$\nabla_{\mathbf{x}_t^b}\log p(\mathbf{X}_t|\mathbf{x}_0^a) = \nabla_{\mathbf{x}_t^b}\log p(\mathbf{x}_0^a|\mathbf{X}_t) + \nabla_{\mathbf{x}_t^b}\log p_t(\mathbf{X}_t)$$

$p(\mathbf{x}_0^a|\mathbf{X}_t)$ is intractable to calculate directly and so, following [9], we approximate it with $\mathcal{N}(\mathbf{x}_0^a; \hat{\mathbf{x}}_0^{\theta a}(\mathbf{X}_t), \frac{1-\alpha_t}{\alpha_t}I_{n_a d})$ where $\hat{\mathbf{x}}_0^{\theta a}(\mathbf{X}_t)$ is a point estimate of $\mathbf{x}_0^a$ given from $s_t^\theta(\mathbf{X}_t)$ calculated as

$$\hat{\mathbf{x}}_0^{\theta a}(\mathbf{X}_t) = \frac{\mathbf{x}_t^a + (1-\alpha_t)s_t^\theta(\mathbf{X}_t)^a}{\sqrt{\alpha_t}}$$

where again we have assumed a correspondence between $\mathbf{x}_t^a$ and $\mathbf{x}_0^a$. Our approximation for $\nabla_{\mathbf{x}_t^b}\log p(\mathbf{x}_0^a|\mathbf{X}_t)$ is then

$$\nabla_{\mathbf{x}_t^b}\log p(\mathbf{x}_0^a|\mathbf{X}_t) \approx -\nabla_{\mathbf{x}_t^b}\frac{\alpha_t}{2(1-\alpha_t)}\|\mathbf{x}_0^a - \hat{\mathbf{x}}_0^{\theta a}(\mathbf{X}_t)\|^2$$

which can be calculated by differentiating through the score network $s_t^\theta$.

We approximate $\overleftarrow{\lambda}_t^*(\mathbf{X}_t|\mathbf{x}_0^a)$ and $A_t^*(\mathbf{y}^{\text{add}}, i|\mathbf{X}_t, \mathbf{x}_0^a)$, with their unconditional forms $\overleftarrow{\lambda}_t^\theta(\mathbf{X}_t)$ and $A_t^\theta(\mathbf{y}^{\text{add}}, i|\mathbf{X}_t)$. We find this approximation still leads to valid generations because the guidance of the score network $s_t^\theta$, results in $\mathbf{X}_t$ containing the conditioning information which in turn leads to $\overleftarrow{\lambda}_t^\theta(\mathbf{X}_t)$ guiding the number of components in $\mathbf{X}_t$ to be consistent with the conditioning information too as verified in our experiments. Further, any errors in the approximation for $A_t^\theta(\mathbf{y}^{\text{add}}, i|\mathbf{X}_t)$ are fixed by further applications of the guided score function, highlighting the benefits of our combined autoregressive and diffusion based approach.

## D Experiment Details

Our code is available at `https://github.com/andrew-cr/jump-diffusion`

### D.1 Molecules

#### D.1.1 Network Architecture

**Backbone** For our backbone network architecture, we used the EGNN used in [8]. This is a specially designed graph neural network applied to the point cloud treating it as a fully connected graph. A special equivariant update is used, operating only on distances between atoms. We refer to [8] for the specific details on the architecture. We used the same size network as used in [8]'s QM9 experiments, specifically there are 9 layers, with a hidden node feature size of 256. The output of the EGNN is fed into a final output projection layer to give the score network output $s_t^\theta(\mathbf{X}_t)$.

**Component number prediction** To obtain $p_{0|t}^\theta(n_0|\mathbf{X}_t)$, we take the embedding produced by the EGNN before the final output embedding and pass it through 8 transformer layers each consisting of a self-attention block and an MLP block applied channel wise. Our transformer model dimension is 128 and so we project the EGNN embedding output down to 128 before entering into the transformer layers. We then take the output of the transformer and take the average embedding over all nodes. This embedding is then passed through a final projection layer to give softmax logits over the $p_{0|t}^\theta(n_0|\mathbf{X}_t)$ distribution.

**Autoregressive Distribution** Our $A_t^\theta(\mathbf{y}^{\text{add}}, i|\mathbf{X}_t)$ network has to predict the position and features for a new atom when it is added to the molecule. Since the point cloud is permutation invariant, we do not need to predict $i$ and so we just need to parameterize $A_t^\theta(\mathbf{y}^{\text{add}}|\mathbf{X}_t)$. We found the network to perform the best if the network first predicts the nearest atom to the new atom and then a vector from that atom to the location of the new atom. To achieve this, we first predict softmax logits for a distribution over the nearest atom by applying a projection to the embedding output from the previously described transformer block. During training, the output of this distribution can be directly supervised by a cross entropy loss. Given the nearest atom, we then need to predict the position and features of the new atom to add. We do this by passing in the embedding generated by the EGNN and original point cloud features into a new transformer block of the same size as that used for $p_{0|t}^\theta(n_0|\mathbf{X}_t)$. We also input the distances from the nearest atom to all other atoms in the molecule currently as an additional feature. To obtain the position of the new atom, we will take a weighted sum of all the vectors between the nearest atom and other atoms in the molecule. This is to make it easy for the network to create new atoms 'in plane' with existing atoms which is useful for e.g. completing rings that have to remain in the same plane. To calculate the weights for the vectors, we apply an output projection to the output of the transformer block. The new atom features (atom type and charge) are generated by a separate output projection from the transformer block. For the position and features, $A_t^\theta(\mathbf{y}^{\text{add}}|\mathbf{X}_t)$ outputs both a mean and a standard deviation for a Gaussian distribution. For the position distribution, we set the standard deviation to be isotropic to remain equivariant to rotations. In total our model has around 7.3 million parameters.

#### D.1.2 Training

We train our model for 1.3 million iterations at a batch size of 64. We use the Adam optimizer with learning rate 0.00003. We also keep a running exponential moving average of the network weights that is used during sampling as is standard for training diffusion models [2, 3, 16] with a decay parameter of 0.9999. We train on the 100K molecules contained in the QM9 training split. We model hydrogens explicitly. Training a model requires approximately 7 days on a single GPU which was done on an Academic cluster.

In [8] the atom type is encoded as a one-hot vector and diffused as a continuous variable along with the positions and charge values for all atoms. They found that multiplying the one-hot vectors by 0.25 to boost performance by allowing the atom-type to be decided later on in the diffusion process. We instead multiply the one-hot vectors by 4 so that atom-type is decided early on in the diffusion process which improves our guided performance when conditioning on certain atom-types being

present. We found our model is robust to this change and achieves similar sample quality to [8] as shown in Table 2.

When deleting dimensions, we first shuffle the ordering of the nodes and then delete the final $n_0 - n_t$ nodes. The cross entropy loss weighting in (25) is set to 1.

Following [8] we train our model to operate within the center of mass (CoM) zero subspace of possible molecule positions. The means, throughout the forward and backward process, the average position of an atom is 0. In our transdimensional framework, this is achieved by first deleting any atoms required under the forward component deletion process. We then move the molecule such that its CoM is 0. We then add CoM free noise such that the noisy molecule also has CoM= 0. Our score model $s_t^\theta$ is parameterized through a noise prediction model $\epsilon_t^\theta$ which is trained to predict the CoM free noise that was added. Therefore, our score network learns suitable directions to maintain the process on the CoM= 0 subspace. For the position prediction from $A_t^\theta(\mathbf{y}^{\text{add}}|\mathbf{X}_t)$ we train it to predict the new atom position from the current molecules reference frame. When the new atom is added, we then update all atom positions such that CoM= 0 is maintained.

### D.1.3 Sampling

During sampling we found that adding corrector steps [3] improved sample quality. Intuitively, corrector steps form a process that has $p_t(\mathbf{X})$ as its stationary distribution rather than the process progressing toward $p_0(\mathbf{X})$. We use the same method to determine the corrector step size $\zeta$ as in [3]. For the conditional generation tasks, we also found it useful to include corrector steps for the component generation process. As shown in [29], corrector steps in discrete spaces can be achieved by simulating with a rate that is the addition of the forward and backward rates. We achieve this in the context of trans-dimensional modeling by first simulating a possible insertion using $\overleftarrow{\lambda}_t^\theta$ and then simulating a possible deletion using $\overrightarrow{\lambda}_t$. We describe our overall sampling algorithm in Algorithm 2.

---

**Algorithm 2:** Sampling the Generative Process with Corrector Steps

**Input:** Number of corrector steps $C$
$t \leftarrow T$
$\mathbf{X} \sim p_{\text{ref}}(\mathbf{X}) = \mathbb{I}\{n = 1\}\mathcal{N}(\mathbf{x}; 0, I_d)$
**while** $t > 0$ **do**
  **if** $u < \overleftarrow{\lambda}_t^\theta(\mathbf{X})\delta t$ with $u \sim \mathcal{U}(0, 1)$ **then**
    Sample $\mathbf{x}^{\text{add}}, i \sim A_t^\theta(\mathbf{x}^{\text{add}}, i|\mathbf{X})$
    $\mathbf{X} \leftarrow \text{ins}(\mathbf{X}, \mathbf{x}^{\text{add}}, i)$
  **end**
  $\mathbf{x} \leftarrow \mathbf{x} - \overleftarrow{\mathbf{b}}_t^\theta(\mathbf{X})\delta t + g_t\sqrt{\delta t}\epsilon$ with $\epsilon \sim \mathcal{N}(0, I_{nd})$
  **for** $c = [1, \dots, C]$ **do**
    $\mathbf{x} \leftarrow \mathbf{x} + \zeta s_{t-\delta t}^\theta(\mathbf{X}) + \sqrt{2\zeta}\epsilon$ with $\epsilon \sim \mathcal{N}(0, I_{nd})$
    **if** $u < \overleftarrow{\lambda}_{t-\delta t}^\theta(\mathbf{X})\delta t$ with $u \sim \mathcal{U}(0, 1)$ **then**
      Sample $\mathbf{x}^{\text{add}}, i \sim A_{t-\delta t}^\theta(\mathbf{x}^{\text{add}}, i|\mathbf{X})$
      $\mathbf{X} \leftarrow \text{ins}(\mathbf{X}, \mathbf{x}^{\text{add}}, i)$
    **end**
    **if** $u < \overrightarrow{\lambda}_{t-\delta t}(n)\delta t$ with $u \sim \mathcal{U}(0, 1)$ **then**
      $\mathbf{X} \leftarrow \text{del}(\mathbf{X}, i)$ with $i \sim K^{\text{del}}(i|n)$
    **end**
  **end**
  $\mathbf{X} \leftarrow (n, \mathbf{x}), t \leftarrow t - \delta t$
**end**

---

### D.1.4 Evaluation

**Unconditional** For our unconditional sampling evaluation, we start adding corrector steps when $t < 0.1T$ in the backward process and use 5 corrector steps without the corrector steps on the number of components. We set $\delta = 0.05$ for $t > 0.5T$ and $\delta = 0.001$ for $t < 0.5T$ such that the total number

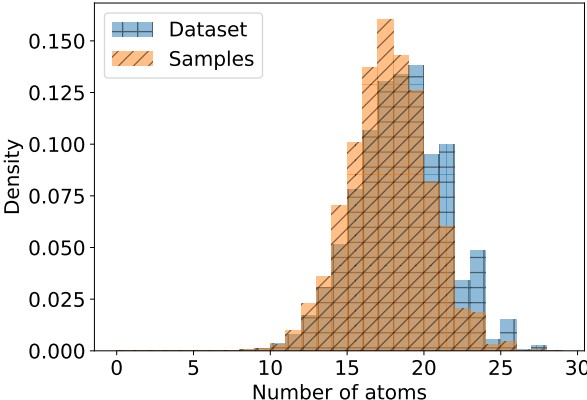

Figure 6: Distribution of the size of molecules in the QM9 dataset as measured through the number of atoms versus the distribution of the size of molecules generated by our unconditional model.

of network evaluations is $1000$. We show the distribution of sizes of molecules generated by our model in Figure 6 and show more unconditional samples in Figure 7. We find our model consistently generates realistic molecules and achieves a size distribution similar to the training dataset even though this is not explicitly trained and arises from sampling our backward rate $\overset{\leftarrow}{\lambda}{}^{\theta}_t$. Since we are numerically integrating a continuous time process and approximating the true time reversal rate $\overset{\leftarrow}{\lambda}{}^{*}_t$, some approximation error is expected. For this experiment, sampling all of our models and ablations takes approximately 2 GPU days on Nvidia 1080Ti GPUs.

**Conditional**  For evaluating applying conditional diffusion guidance to our model, we choose $10$ conditioning tasks that each result in a different distribution of target dimensions. The task is to produce molecules that include at least a certain number of target atom types. We then guide the first set of atoms generated by the model to have these desired atom types. The tasks chosen are given in Table 5. Molecules in the training dataset that meet the conditions in each task have a different distribution of sizes. The tasks were chosen so that we have an approximately linearly increasing mean number of atoms for molecules that meet the condition. We also require that there are at least 100 examples of molecules that meet the condition within the training dataset.

For sampling when using conditional diffusion guidance, we use 3 corrector steps throughout the backward process with $\delta t = 0.001$. For these conditional tasks, we include the corrector steps on the number of components. We show the distribution of dimensions for each task from the training dataset and from our generated samples in Figure 8. Our metrics are calculated by first drawing 1000 samples for each conditioning task and then finding the Hellinger distance between the size distribution generated by our method (orange diagonal hashing in Figure 8) and the size distribution for molecules in the training dataset that match the conditions of the task (green no hashing in Figure 8). We find that indeed our model when guided by diffusion guidance can automatically produce a size distribution close to the ground truth size distribution found in the dataset for that conditioning value. We show samples generated by our conditionally guided model in Figure 9. We can see that our model can generate realistic molecules that include the required atom types and are of a suitable size. For this experiment, sampling all of our models and ablations takes approximately 13 GPU days on Nvidia 1080Ti GPUs.

**Interpolations**  For our interpolations experiments, we follow the set up of [8] who train a new model conditioned on the polarizability of molecules in the dataset. We train a conditional version of our model which can be achieved by simply adding in the polarizability as an additional feature input to our backbone network and re-using all the same hyperparameters. We show more examples of interpolations in Figure 10.

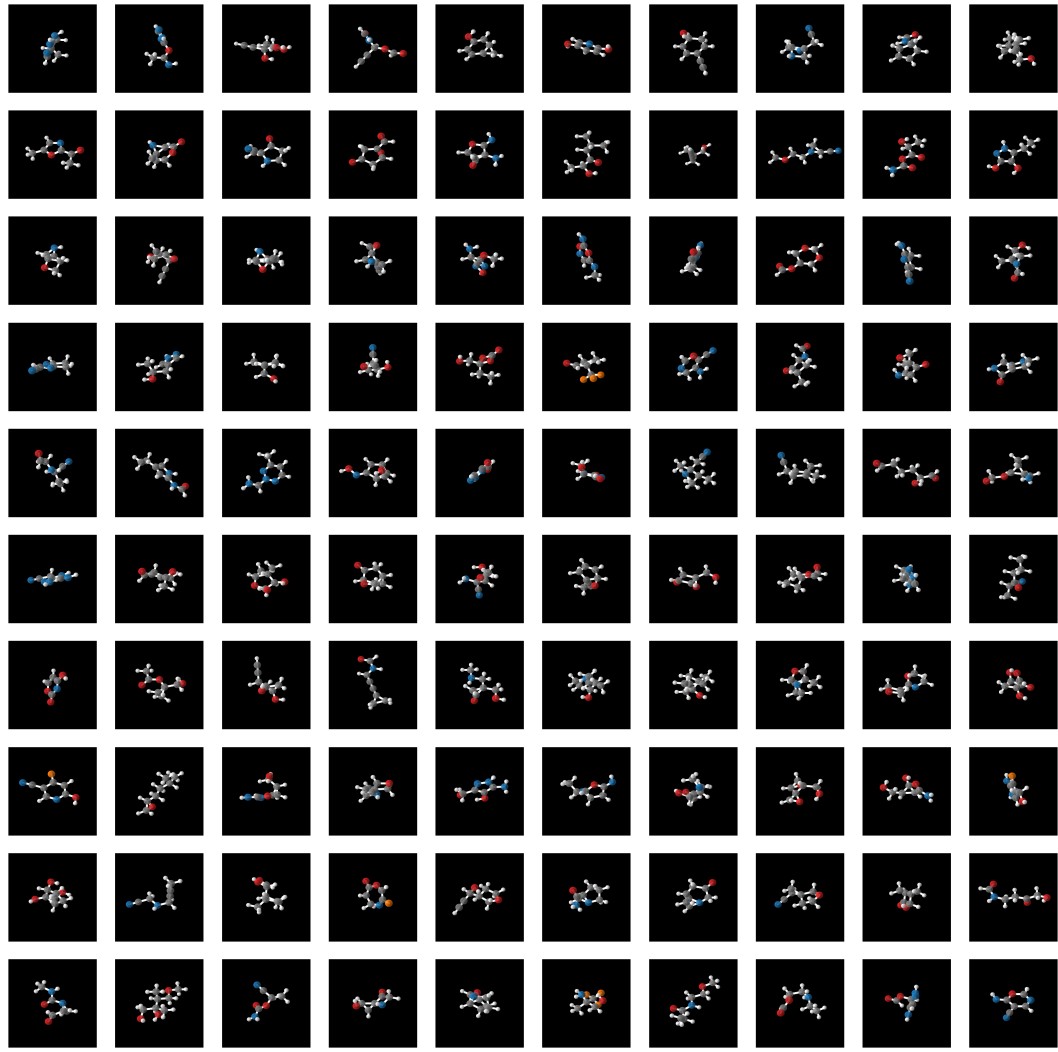

Figure 7: Unconditional samples from our model.

### D.1.5 Ablations

For our main model, we set $\overrightarrow{\lambda}_{t<0.1T} = 0$ to ensure that all dimensions are added with enough generation time remaining for the diffusion process to finalize all state values. To verify this setting, we compare its performance with $\overrightarrow{\lambda}_{t<0.03T} = 0$ and $\overrightarrow{\lambda}_{t<0.3T} = 0$. We show our results in Table 6. We find that the $\overrightarrow{\lambda}_{t<0.03T} = 0$ setting to generate reasonable sample quality but incur some extra dimension error due to the generative process sometimes observing a lack of dimensions near $t = 0$ and adding too many dimensions. We observed the same effect in the paper for when setting $\overrightarrow{\lambda}_t$ to be constant for all $t$ in Table 3. Further, the setting $\overrightarrow{\lambda}_{t<0.3T} = 0$ also results in increased dimension error due to there being less opportunity for the guidance model to supervise the number of dimensions. We find that $\overrightarrow{\lambda}_{t<0.1T} = 0$ to be a reasonable trade-off between these effects.

### D.1.6 Uniqueness and Novelty Metrics

We here investigate sample diversity and novelty of our unconditional generative models. We measure uniqueness by computing the chemical graph corresponding to each generated sample and measure what proportion of the 10000 produced samples have a unique chemical graph amongst this set of 10000 as is done in [8]. We show our results in Table 7 and find our TDDM method to have slightly lower levels of uniqueness when compared to the fixed dimension diffusion model baseline.

Table 5: The 10 conditioning tasks used for evaluation. The number of each atom type required for the task is given in columns $2-5$ whilst the average number of atoms in molecules that meet this condition in the training dataset is given in the 6th column.

| Task | Carbon | Nitrogen | Oxygen | Fluorine | Mean Number of Atoms |
|------|--------|----------|--------|----------|----------------------|
| 1 | 4 | 1 | 2 | 1 | 11.9 |
| 2 | 4 | 3 | 1 | 1 | 13.0 |
| 3 | 5 | 2 | 1 | 1 | 13.9 |
| 4 | 6 | 0 | 1 | 1 | 14.6 |
| 5 | 5 | 3 | 1 | 0 | 16.0 |
| 6 | 6 | 3 | 0 | 0 | 17.2 |
| 7 | 6 | 1 | 2 | 0 | 17.7 |
| 8 | 7 | 1 | 1 | 0 | 19.1 |
| 9 | 8 | 1 | 0 | 0 | 19.9 |
| 10 | 8 | 0 | 1 | 0 | 21.0 |

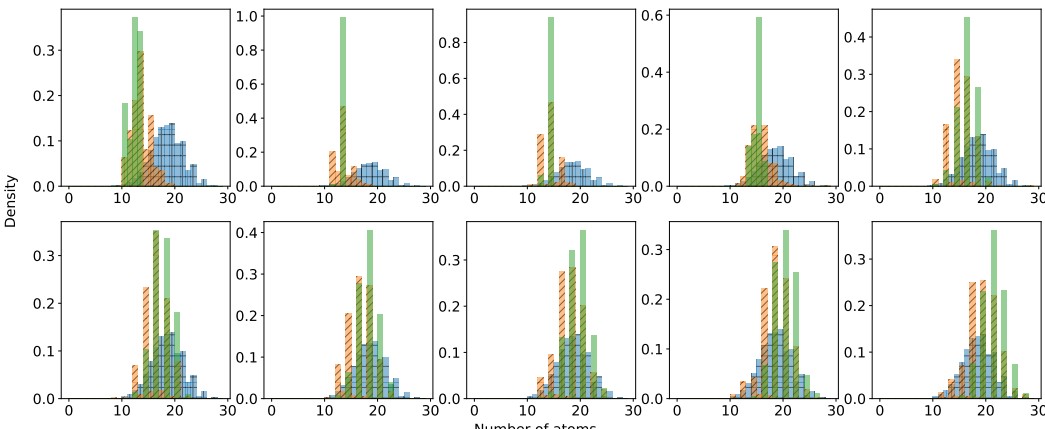

Figure 8: Distribution of molecule sizes for each conditioning task. Tasks $1-5$ are shown left to right in the top row and tasks $6-10$ are shown left to right in the bottom row. We show the unconditional size distribution from the dataset in blue vertical/horizontal hashing, the size distribution of our conditionally generated samples in orange diagonal hashing and finally the size distribution for molecules in the training dataset that match the conditions of each task (the ground truth size distribution) in green no hashing.

Measuring novelty on generative models trained on the QM9 dataset is challenging because the QM9 dataset contains an exhaustive enumeration of all molecules that satisfy certain predefined constraints [46], [8]. Therefore, if a novel molecule is produced it means the generative model has failed to capture some of the physical properties of the dataset and indeed it is found in [8] that during training, as the model improved, novelty decreased. Novelty is therefore not typically included in evaluating molecular diffusion models. For completeness, we include the novelty scores in Table 7 as a comparison to the results presented in [8] Appendix C. We find that our samples are closer to the statistics of the training dataset whilst still producing 'novel' samples at a consistent rate.

## D.2 Video

### D.2.1 Dataset

We used the $VP^2$ benchmark, which consists of $35\,000$ videos, each 35 frames long. The videos are evenly divided among seven tasks, namely: `push {red, green, blue} button, open {slide, drawer}, push {upright block, flat block} off table`. The 5000 videos for each task were collected using a scripted task-specific policy operating in the RoboDesk environment [40]. They sample an action vector at every step during data generation by adding i.i.d. Gaussian

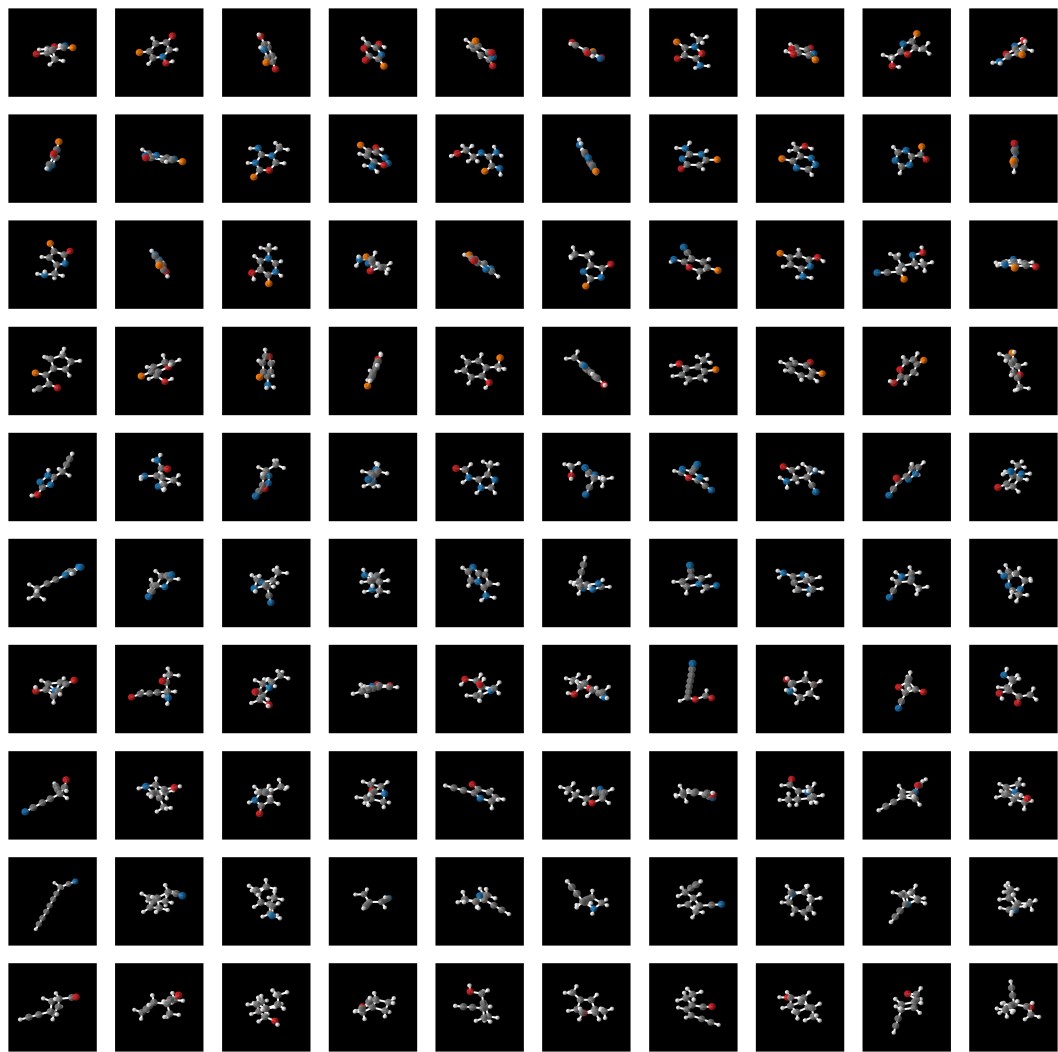

Figure 9: Samples generated by our model when conditional diffusion guidance is applied. Each row represents one task with task 1 at the top, down to task 10 at the bottom. For each task, 10 samples are shown in each row.

noise to each dimension of the action vector output by the scripted policy. For each task, they sample 2500 videos with noise standard deviation 0.1 and 2500 videos with standard deviation 0.2. We filter out the lower-quality trajectories sampled with noise standard deviation 0.2, and so use only the 17 500 videos (2500) per task with noise standard deviation 0.1. We convert these videos to $32 \times 32$ resolution and then, so that the data we train on has varying lengths, we create each training example by sampling a length $l$ from a uniform distribution over $\{2, \ldots, 35\}$ and then taking a random $l$-frame subset of the video.

### D.2.2 Forward Process

The video domain differs from molecules in two important ways. The first is that videos cannot be reasonably treated as a permutation-invariant set. This is because the order of the frames matters. Secondly, generating a full new component for the molecules with a single pass autoregressive network is feasible, however, a component for the videos is a full frame which is challenging for a single pass autoregressive network to generate. We design our forward process to overcome these challenges.

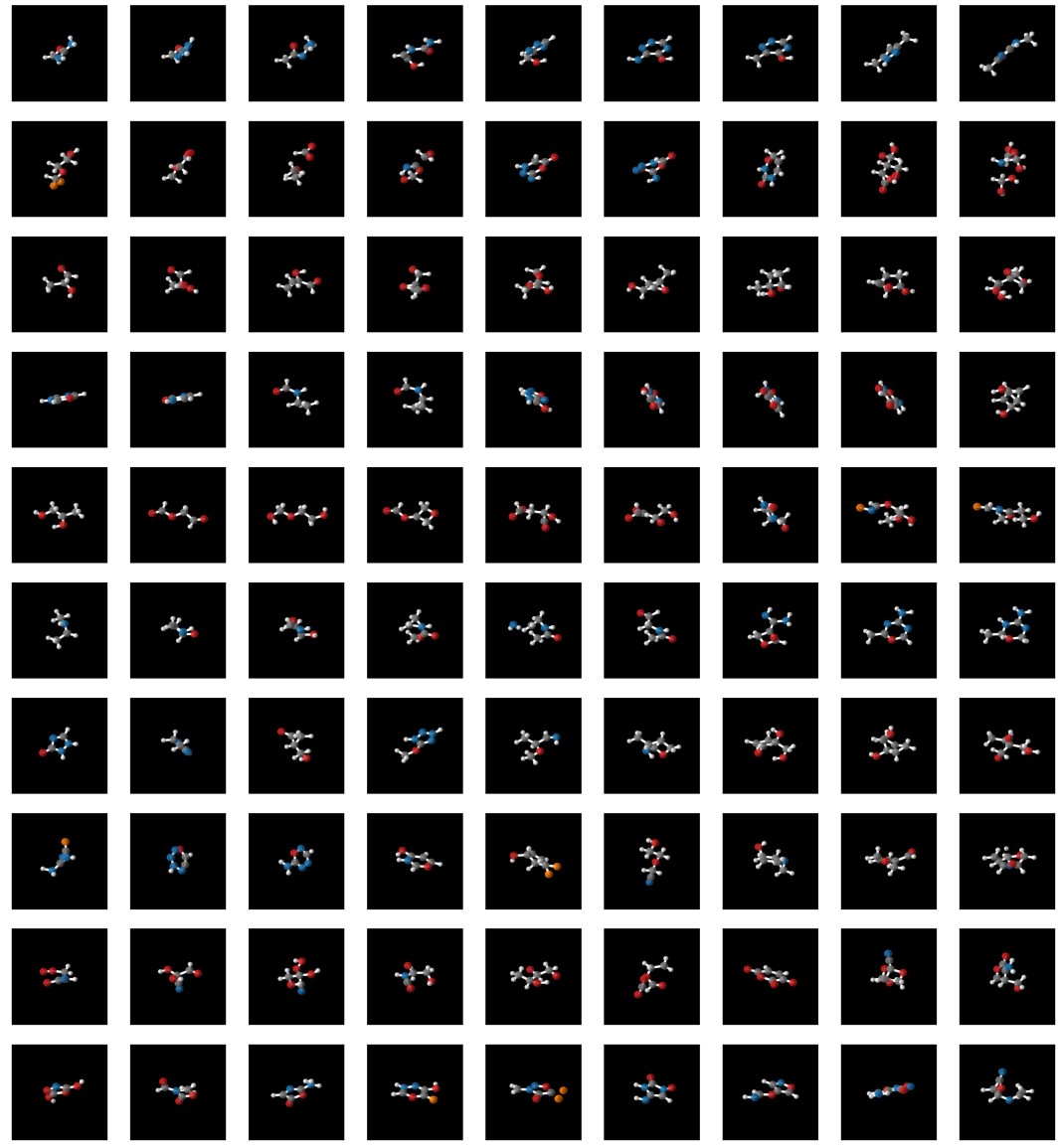

Figure 10: Interpolations showing a sequence of generations for linearly increasing polarizability from 39 Bohr$^3$ to 66 Bohr$^3$ with fixed random noise. Each row shows an individual interpolation with Bohr$^3$ increasing from left to right.

Table 6: Ablation of when to set the forward rate to $0$ on the conditional molecule generation task. We report dimension error as the average Hellinger distance between the generated and ground truth conditional dimension distributions as well as average sample quality metrics. Metrics are reported after 620k training iterations.

| Method | Dimension Error | % Atom stable | % Molecule Stable | % Valid |
|---|---|---|---|---|
| $\overrightarrow{\lambda}_{t<0.03T} = 0$ | $0.227_{\pm 0.16}$ | $91.5_{\pm 3.7}$ | $56.5_{\pm 9.8}$ | $72.0_{\pm 11}$ |
| $\overrightarrow{\lambda}_{t<0.1T} = 0$ | $0.162_{\pm 0.071}$ | $92.4_{\pm 2.8}$ | $53.9_{\pm 12}$ | $72.7_{\pm 9.6}$ |
| $\overrightarrow{\lambda}_{t<0.3T} = 0$ | $0.266_{\pm 0.11}$ | $92.0_{\pm 3.2}$ | $53.5_{\pm 13}$ | $66.6_{\pm 12}$ |

Table 7: Uniqueness and novelty metrics on unconditional molecule generation. We produce 10000 samples for each method and measure validity using RDKit. Uniquenss is judged as whether the chemical graph is unique amongst the 10000 produced samples. Amongst the valid and unique molecules, we then find the percentage that have a chemical graph not present in the training dataset.

| Method | % Valid | % Valid and Unique | Percentage of Valid and Unique Molecules that are Novel |
|---|---|---|---|
| FDDM [8] | 91.9 | 90.7 | 65.7 |
| TDDM (ours) | 92.3 | 89.9 | 53.6 |
| TDDM, const $\overrightarrow{\lambda}_t$ | 86.7 | 84.4 | 56.9 |
| TDDM, $\overrightarrow{\lambda}_{t<0.9T} = 0$ | 89.4 | 86.1 | 51.3 |
| TDDM w/o Prop. 3 | 87.1 | 85.9 | 63.3 |

We define our forward process to delete frames in a random order. This means that during generation, frames can be generated in any order in the reverse process, enabling more conditioning tasks since we can always ensure that whichever frames we want to condition on are added first. Further, we use a non-isotropic noise schedule by adding noise just to the frame that is about to be deleted. Once it is deleted, we then start noising the next randomly chosen frame. This is so that, in the backward direction, when a new frame is added, it is simply Gaussian noise. Then the score network will fully denoise that new frame before the next new frame is added. We now specify exactly how our forward process is constructed.

We enable random-order deletion by applying an initial shuffling operation occurring at time $t = 0$. Before this operation, we represent the video $\mathbf{x}$ as an ordered sequence of frames, $\mathbf{x}_0 = [\mathbf{x}_1, \mathbf{x}_2, \ldots, \mathbf{x}_{n_0}]$. During shuffling, we sample a random permutation $\pi$ of the integers $1, \ldots, n_0$. Then the frames are kept in the same order, but annotated with an index variable so that we have $\mathbf{x}_{0+} = [(\mathbf{x}_{0+}^{(1)}, \pi(1)), (\mathbf{x}_{0+}^{(2)}, \pi(2)), \ldots, (\mathbf{x}_{0+}^{(n_0)}, \pi(n_0))]$.

We will run the forward process from $t = 0$ to $t = 100N$. We will set the forward rate such we delete down from $n_t$ to $n_t - 1$ at time $(N - n_t + 1)100$. This is achieved heuristically by setting

$$\overrightarrow{\lambda}_t(n_t) = \begin{cases} 0 & \text{for } t < (N - n_t + 1)100, \\ \infty & \text{for } t \geq (N - n_t + 1)100. \end{cases}$$

We can see that at time $t = (N - n_t + 1)100$ we will quickly delete down from $n_t$ to $n_t - 1$ at which point $\overrightarrow{\lambda}_t(n_t)$ will become 0 thus stopping deletion until the process arrives at the next multiple of 100 in time. When we hit a deletion event, we delete the frame from $\mathbf{X}_t$ that has the current highest index variable $\pi(n)$. In other words

$$K^{\text{del}}(i|\mathbf{X}_t) = \begin{cases} 1 & \text{for } n_t = \mathbf{x}_t^{(i)}[2], \\ 0 & \text{otherwise} \end{cases}$$

where we use $\mathbf{x}_t^{(i)}[2]$ to refer to the shuffle index variable for the $i$th current frame in $\mathbf{x}_t$.

We now provide an example progression of the forward deletion process. Assume we have $n_0 = 4$, $N = 5$ and sample a permutation such that $\pi(1) = 3, \pi(2) = 2, \pi(3) = 4$, and $\pi(4) = 1$. Initially

the state is augmented to include the shuffle index. Then the forward process progresses from $t = 0$ to $t = 500$ with components being deleted in descending order of the shuffle index

$$\mathbf{x}_{0+} = [(\mathbf{x}_t^{(1)}, 3), (\mathbf{x}_t^{(2)}, 2), (\mathbf{x}_t^{(3)}, 4), (\mathbf{x}_t^{(4)}, 1)]$$

$$\mathbf{x}_{100+} = [(\mathbf{x}_t^{(1)}, 3), (\mathbf{x}_t^{(2)}, 2), (\mathbf{x}_t^{(3)}, 4), (\mathbf{x}_t^{(4)}, 1)]$$

$$\mathbf{x}_{200+} = [(\mathbf{x}_t^{(1)}, 3), (\mathbf{x}_t^{(2)}, 2), (\mathbf{x}_t^{(4)}, 1)]$$

$$\mathbf{x}_{300+} = [(\mathbf{x}_t^{(2)}, 2), (\mathbf{x}_t^{(4)}, 1)]$$

$$\mathbf{x}_{400+} = [(\mathbf{x}_t^{(4)}, 1)]$$

In this example, due to the random permutation sampled, the final video frame remained after all others had been deleted. Note that the order of frames is preserved as we delete frames in the forward process although the spacing between them can change as we delete frames in the middle.

Between jumps, we use a noising process to add noise to frames. The noising process is non-isotropic in that it adds noise to different frames at different rates such that the a frame is noised only in the time window immediately preceding its deletion. For component $i \in [1, \ldots, n_t]$, we set the forward noising process such that $p_{t|0}(\mathbf{x}_t^{(i)}|\mathbf{x}_0^{(i)}, M_t) = \mathcal{N}(\mathbf{x}_t^{(i)}; \mathbf{x}_0^{(i)}, \sigma_t(\mathbf{x}_t^{(i)})^2)$ where $\mathbf{x}_0^{(i)}$ is the clean frame corresponding to $\mathbf{x}_t^{(i)}$ as given by the mask $M_t$ and $\sigma_t(\mathbf{x}_t^{(i)})$ follows

$$\sigma_t(\mathbf{x}_t^{(i)}) = \begin{cases} 0 & \text{for } t < (N - \mathbf{x}_t^{(i)}[2])100, \\ 100 & \text{for } t > (N - \mathbf{x}_t^{(i)}[2])100, \\ t - (N - \mathbf{x}_t^{(i)}[2])100 & \text{for } (N - \mathbf{x}_t^{(i)}[2])100 \leq t \leq (N - \mathbf{x}_t^{(i)}[2] + 1)100 \end{cases}$$

where we again use $\mathbf{x}_t^{(i)}[2]$ for the shuffle index of component $i$. This is the VE-SDE from [3] applied to each frame in turn. We note that we only add noise to the state values on not the shuffle index itself. The SDE parameters that result in the VE-SDE are $\overrightarrow{\mathbf{b}}_t = 0$ and $\overrightarrow{g}_t = \sqrt{2t - 2(N - \mathbf{x}_t^{(i)}[2])100}$.

### D.2.3 Sampling the Backward Process

When $t$ is not at a multiple of 100, the forward process is purely adding Gaussian noise, and so the reverse process is also purely operating on the continuous dimensions. We use the Heun sampler proposed by [16] to update the continuous dimensions in this case, and also a variation of their discretisation of $t$ - specifically to update from e.g. $t = 600$ to $t = 500$, we use their discretization of $t$ as if the maximum value was 100 and then offset all values by 500.

To invert the dimension deletion process, we can use Proposition 3 to derive our reverse dimension generation process. We re-write our parameterized $\overleftarrow{\lambda}_t^\theta$ using Proposition 3 as

$$\overleftarrow{\lambda}_t^\theta(\mathbf{X}_t) = \overrightarrow{\lambda}_t(n_t + 1)\mathbb{E}_{p_{0|t}^\theta(n_0|\mathbf{X}_t)}\left[\frac{p_{t|0}(n_t + 1|n_0)}{p_{t|0}(n_t|n_0)}\right]$$

At each time multiple of 100 in the backward process, we will have an opportunity to add a component. At this time point, we estimate the expectation with a single sample $n_0 \sim p_{0|t}^\theta(n_0|\mathbf{X}_t)$. If $n_0 > n_t$ then $\overleftarrow{\lambda}_t^\theta(\mathbf{X}_t) = \infty$. The new component will then be added at which point $\overleftarrow{\lambda}_t^\theta(\mathbf{X}_t)$ becomes 0 for the remainder of this block of time due to $n_t$ becoming $n_t + 1$. If $n_0 = n_t$ then $\overleftarrow{\lambda}_t^\theta(\mathbf{X}_t) = 0$ and no new component is added. $\overleftarrow{\lambda}_t^\theta(\mathbf{X}_t)$ will continue to be 0 for the remainder of the backward process once an opportunity to add a component is not used.

When a new frame is added, we use $A_t^\theta(\mathbf{y}^{\text{add}}, i|\mathbf{X}_t)$ to decide where the frame is added and its initial value. Since when we delete a frame it is fully noised, $A_t^\theta(\mathbf{y}^{\text{add}}, i|\mathbf{X}_t)$ can simply predict Gaussian noise for the new frame $\mathbf{y}^{\text{add}}$. However, $A_t^\theta(\mathbf{y}^{\text{add}}, i|\mathbf{X}_t)$ will still learn to predict a suitable location $i$ to place the new frame such that backward process is the reversal of the forward.

We give an example simulation from the backward generative process in Figure 11.

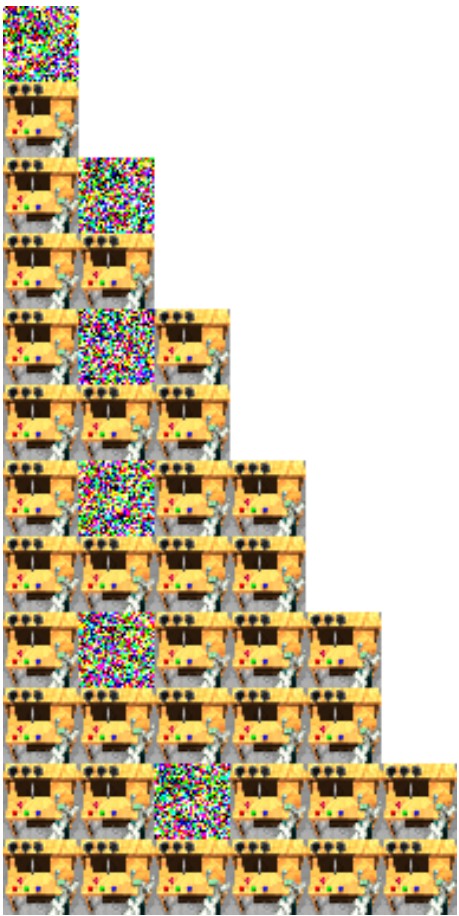

Figure 11: An example simulation of the backward generative process conditioned on the first and last frame. Note how the process first adds a new frame and then fully denoises it before adding the next frame. Since the first and last frame are very similar, the process produces a short video.

### D.2.4 Network Architecture

Our video diffusion network architecture is based on the U-net used by [42], which takes as input the index of each frame within the video, and uses the differences between these indices to control the interactions between frames via an attention mechanism. Since, during generation, we do not know the final position of each frame within the $\mathbf{x}_0$, we instead pass in its position within the ordered sequence $\mathbf{x}_t$.

One further difference is that, since we are perform non-isotropic diffusion, the standard deviation of the added noise will differ between frames. We adapt to this by performing preconditioning, and inputting the timestep embedding, separately for each frame $\mathbf{x}_t^{(i)}$ based on $\sigma_t(\mathbf{x}_t^{(i)})$ instead of basing them on the global diffusion timestep $t$. Our timestep embedding and pre- and post-conditioning of network inputs/outputs are as suggested by [16], other than being done on a per-frame basis. The architecture from [42] with these changes applied then gives us our score network $s_t^\theta$.

While it would be possible to train a single network that estimates the score and all quantities needed for modelling jumps, we chose to train two separate networks in order to factorize our exploration of the design space. These were the score network $s_t^\theta$, and the rate and index prediction network modeling $p_{0|t}^\theta(n_0|\mathbf{X}_t)$ and $A_t^\theta(i|\mathbf{X}_t)$. The rate and index prediction network is similar to the first half of the score network, in that it uses all U-net blocks up to and including the middle one. We then flatten the $512 \times 4 \times 4$ hidden state for each frame after this block such that, for an $n_t$ frame input, we obtain a $n_t \times 8192$ hidden state. These are fed through a 1D convolution with kernel size 2 and zero-padding of size 1 on each end, reducing the hidden state to $(n_t + 1) \times 128$, which is in turn fed

through a ReLU activation function. This hidden state is then fed into three separate heads. One head maps it to the parameters of $A_t^\theta(i|\mathbf{X}_t)$ via a 1D convolution of kernel size 3. The output of size $(n_t + 1)$ is fed through a softmax to provide the categorical distribution $A_t^\theta(i|\mathbf{X}_t)$. The second head averages the hidden state over the "frame" dimension, producing a 128-dimensional vector. This is fed through a single linear layer and a softmax to parameterize $p_{0|t}^\theta(n_0|\mathbf{X}_t)$. Finally, the third head consists of a 1D convolution of kernel size 3 with 35 output channels. The $(n_t + 1) \times 35$ output is fed through a softmax to parameterize distributions over the number of frames that were deleted from $\mathbf{X}_0$ which came before the first in $\mathbf{x}_t$, the number of frames from $\mathbf{X}_0$ which were deleted between each pair of frames in $\mathbf{x}_t$, and the number deleted after the last frame in $\mathbf{x}_t$. We do not use this head at inference-time but found that including it improved the performance of the other heads by helping the network learn better representations.

For a final performance improvement, we note that under our forward process there is only ever one "noised" frame in $\mathbf{x}_t$, while there are sometimes many clean frames. Since the cost of running our architecture scales with the number of frames, running it on many clean frames may significantly increase the cost while providing little improvement to performance. We therefore only feed into the architecture the "noised" frame, the two closest "clean" frames before it, and the two closest "clean" frames after it. See our released source code for the full implementation of this architecture.

### D.2.5 Training

To sample $t$ during training, we adapt the log-normal distribution suggested by [16] in the context of isotropic diffusion over a single image. To apply it to our non-isotropic video diffusion, we first sample which frames have been deleted, which exist with no noise, and which have had noise added, by sampling the timestep from a uniform distribution and simulating our proposed forward process. We then simply change the noise standard deviation for the noisy frame, replacing it with a sample from the log-normal distribution. The normal distribution underlying our log-normal has mean $-0.6$ and standard deviation $1.8$. This can be interpreted as sampling the timestep from a mixture of log-normal distributions, $\frac{1}{N} \sum_{i=0}^{N-1} \mathcal{LN}(t - 100i; -0.6, 1.8^2)$. Here, the mixture index $i$ can be interpreted as controlling the number of deleted frames.

We use the same loss weighting as [16] but, similarly to our use of preconditioning, compute the weighting separately for each frame $\mathbf{x}_t^{(i)}$ as a function of $\sigma_t(\mathbf{x}_t^{(i)})$ to account for the non-isotropic noise.

### D.2.6 Perceptual Quality Metrics

We now verify that our reverse process does not have any degradation in quality during the generation as more dimensions are added. We generate 10000 videos and throw away the 278 that were sampled to have only two frames. We then compute the FID score for individual frames in each of the remaining 9722 videos. We group together the scores for all the first frames to be generated in the reverse process and then for the second frame to be generated and so on. We show our results in Table 8. We find that when a frame is inserted has no apparent effect on perceptual quality and conclude that there is no overall degradation in quality as our sampling process progresses. We note that the absolute value of these FID scores may not be meaningful due to the RoboDesk dataset being far out of distribution for the Inception network used to calculate FID scores. We can visually confirm good sample quality from Figure 5.

Table 8: FID for video frames grouped by when they were inserted during sampling.

| 1st | 2nd | 3rd | 3rd last | 2nd last | last |
|------|------|------|----------|----------|------|
| 34.2 | 34.9 | 34.7 | 34.2 | 34.1 | 34.4 |

## E    Broader Impacts

In this work, we presented a general method for performing generative modeling on datasets of varying dimensionality. We have not focused on applications and instead present a generic method.

Along with other generic methods for generative modeling, we must consider the potential negative social impacts that these models can cause when inappropriately used. As generative modeling capabilities increase, it becomes simpler to generate fake content which can be used to spread misinformation. In addition to this, generative models are becoming embedded into larger systems that then have real effects on society. There will be biases present within the generations created by the model which in turn can reinforce these biases when the model's outputs are used within wider systems. In order to mitigate these harms, applications of generative models to real world problems must be accompanied with studies into their biases and potential ways they can be misused. Further, public releases of models must be accompanied with model cards [47] explaining the biases, limitations and intended uses of the model.

