# OpenReview forum: "Trans-Dimensional Generative Modeling via Jump Diffusion Models"
_NeurIPS.cc/2023/Conference — NeurIPS 2023 spotlight_

### Official Review · Reviewer_tNRG · 2023-07-05

**Soundness:** 3 good
**Presentation:** 3 good
**Contribution:** 3 good
**Rating:** 7
**Confidence:** 4

**Summary:**

This paper proposes a new diffusion model based on jump diffusion processes. Compared with previous discrete and continuous formulations, the model introduces the usage of a transition kernel, which models the jump process in a semantically meaningful manner. The method absorbs standard constructions like diffusion guidance and produces good results on molecule and robot arm video tasks.

**Strengths:**

* The proposed methodology is elegant and theoretically sound.
* Many constructions like diffusion guidance extend naturally.
* The proposed methodology is widely applicable, as many natural data types include a continuous portion (modeling the space) as well as a discrete but unknown component.

**Weaknesses:**

* For the molecule task, the metrics are based on molecular validity properties. This seems like it wouldn't account for overfitting from the model, which seems more relevant for a generative modeling task. In particular, can you report a diversity/non-memorization metric?
* The robot arm example, while sufficient for the paper, is rather toy.

**Questions:**

Nothing more than the weaknesses section.

**Limitations:**

Yes

---

> ### Author Rebuttal · Authors · 2023-08-09
>
> We would like to thank the reviewer for their review and positive feedback. We are very pleased to hear that our approach is considered to be elegant and theoretically sound with wide applicability. We address the specific comments from the review here.
>
> > *For the molecule task, the metrics are based on molecular validity properties. This seems like it wouldn't account for overfitting from the model, which seems more relevant for a generative modeling task. In particular, can you report a diversity/non-memorization metric?*
>
> Thank you for this suggestion, we have investigated sample diversity and novelty and will include these results in an update to the paper. On the molecule task, we can measure uniqueness by computing the chemical graph corresponding to each generated sample and measure what proportion of the 10000 produced samples have a unique chemical graph amongst this set of 10000 as is done by Hoogeboom et al. 2022. We show our results in Table 1 in the additional results pdf and find our method without any ablations has only slightly lower levels of uniqueness when compared to the fixed dimension diffusion model baseline. Measuring novelty on generative models trained on the QM9 dataset is challenging because the QM9 dataset contains an exhaustive enumeration of all molecules that satisfy certain predefined constraints as noted by Vignac et al. 2022 and Hoogeboom et al. 2022. Therefore, if a novel molecule is produced it means the generative model has failed to capture some of the physical properties of the dataset and indeed Hoogeboom et al. 2022 found that during training, as the model improved, novelty decreased. Novelty is therefore not typically included in evaluating molecular diffusion models. For completeness, we include the novelty scores in Table 1 as a comparison to the results presented in Hoogeboom et al. 2022 Appendix C. We find that our samples are closer to the statistics of the training dataset whilst still producing ‘novel’ samples at a consistent rate.
>
>
> ## References
> Emiel Hoogeboom, Vıctor Garcia Satorras, Clément Vignac, and Max Welling. *Equivariant diffusion for molecule generation in 3d.* International Conference on Machine Learning, 2022.
>
> Clement Vignac and Pascal Frossard. *Top-n: Equivariant set and graph generation without exchangeability.* International Conference on Learning Representations, 2022

---

> > ### Comment · Reviewer_tNRG · 2023-08-18
> > **Thank you for your response!**
> >
> > Thank you for clearing up any confusion that I had.

---

### Official Review · Reviewer_fiGg · 2023-07-06

**Soundness:** 3 good
**Presentation:** 3 good
**Contribution:** 3 good
**Rating:** 7
**Confidence:** 3

**Summary:**

This paper focuses on varying dimensional datasets and proposes a novel generative model to solve the varying dimensional problems. The proposed model is theoretically valid and has an interesting and novel contribution to extending the traditional score-based generative model by generating both state values and dimensions jointly during the generative process, which is idea-simple but effective. Experiments on molecule generation and video generation both show this model's effectiveness.

**Strengths:**

1. The idea of jointly modeling the state and dimension, in particular, the idea of using the intensity function to model the jump distribution, is interesting and novel.

2. The theoretical contribution is valid and the experiment is thorough in supporting the proposed model.

**Weaknesses:**

None.

**Questions:**

1. I am confused about how to properly define $K^{{del}}(i|n)$ in $\overrightarrow{K}_t(\mathbf{Y}|\mathbf{X})$? Any clarification about this issue will be helpful.

**Limitations:**

Yes, the authors have adequately addressed the limitations and potential negative societal impact of their work.

---

> ### Author Rebuttal · Authors · 2023-08-09
>
> We thank the reviewer for their review and we appreciate the praise for our method’s novelty and thorough experiments. We address your questions below.
>
> > *I am confused about how to properly define $K^\text{del}(i | n)$ in $\overrightarrow{K}_t(\mathbf{Y} | \mathbf{X})$ Any clarification about this issue will be helpful.*
>
> $K^\text{del}(i | n)$ will dictate the ordering in which dimensions are deleted in the forward noising process. For example if $K^\text{del}(i | n) = \mathbb{I} \\{ i = n \\}$ then it is always the final dimension of the datapoint that is deleted when a jump occurs. Alternatively, you can set $K^\text{del}(i|n) = 1/n$ so that the dimension to be deleted is chosen uniformly at random when a jump occurs.
>
> The choice of $K^\text{del}( i | n)$ impacts the reverse generative process. If $K^\text{del}(i | n) = \mathbb{I} \\{ i = n \\}$ then datapoints are constructed in an additive way, each new dimension is simply appended onto the end of the current datapoint. This is similar to how a text based autoregressive model builds sentences by generating new words and appending them onto the end of the current sentence. Instead, if $K^\text{del}(i|n) = 1/n$ then during the reverse generative process, the generative model first picks a suitable place to add a new dimension and then inserts one at that chosen point.
>
> The choice of $K^\text{del}( i | n)$ depends on the dataset and problem being tackled. In our case, we choose $K^\text{del}(i|n) = 1/n$ because for molecules there is no natural notion of a ‘final dimension’ due to permutation invariance of the point cloud. Further, for our videos we want to condition on the first and last frame thus an autoregressive appending style of generation would be unsuitable. We present our framework in a general manner so it can be easily applied to other problems where other choices of $K^\text{del}( i | n)$ could be more suitable.
>
> We will make this choice clearer in an update to the paper, thank you for raising this point of confusion.

---

> > ### Comment · Reviewer_fiGg · 2023-08-11
> >
> > Thanks for the detailed response!  The author's reply cleared up my confusion. For this reason, I improved my score further.

---

### Official Review · Reviewer_FTvW · 2023-07-06

**Soundness:** 4 excellent
**Presentation:** 4 excellent
**Contribution:** 4 excellent
**Rating:** 8
**Confidence:** 4

**Summary:**

This paper addresses the problem of modelling data of various dimensions. This is achieved by generalising diffusion models as jump diffusion processes, allowing the content and dimension of data to be jointly modelled. The forward process gradually corrupts the data with gaussian noise while also gradually deleting dimensions until a single normally distributed dimension remains. The reverse process learns both the score function, when to add a dimension, and what data to place in the newly created dimension. Experimental results on molecule and video generation tasks show that the approach can well represent the distribution of data dimensions, while outperforming/being competitive in terms of sample quality.

**Strengths:**

- The proposed approach of modelling the generative process as a jump diffusion process is interesting and is a sensible solution.
- The paper is very clear, easy to read, and to my understanding, the method is technically sound. I particularly like the intuitive explanation of the loss in Equations 3-4; and predicting the original data dimension (line 191) is a good idea to address the optimisation issues.
- Experimental results demonstrate the effectiveness of the proposed approach (Table 2), performing comparability to, or outperforming the baseline which sample dimension prior to sampling. I also like the evaluation of the impact of setting $\lambda=0$ (dimension deletion/insertion rate) towards the end of the diffusion process.
- It is shown that reconstruction guidance can be used to generate molecules with specified features (Table 3) much better than dimension independent approaches.
- The problem of jointly modelling content and dimension of data is an important one to address. In my opinion, the proposed solution is compelling and I believe it will be very useful to others.


**Weaknesses:**

- Predicting the content of the newly created dimensions with a gaussian distribution is limiting, potentially creating a discrepancy between train/test time and requiring more diffusion time to correct.
- Fixing $\lambda=0$ towards the end of the diffusion process to ensure added dimensions have time to diffuse is a weakness of the proposed approach. Particularly for more complex data, there is no guarantee that the remaining time will be sufficient.
- Finally, it would have been nice to see a perceptual quality metric for the video generation task, this would offset the above weaknesses.

**Questions:**

- Do the authors have any rationale/evidence that fixing $\lambda$ at $t<0.1$ is a reasonable choice? For instance, graphing metrics over a variety of limits.
- Similarly, is there rationale why predicting new dimensions with a gaussian is sufficient; or if for more complex data this is problematic, is there a more expressive extension?

**Limitations:**

Limitations are well discussed throughout the paper. The previously discussed weaknesses could be mentioned as well.

---

> ### Author Rebuttal · Authors · 2023-08-09
>
> We would like to thank the reviewer for their review and positive comments on the paper. We are especially happy to hear that our proposed method is considered to be a compelling solution to an important problem. You have raised important points regarding the new dimension distribution and the time needed to diffuse at the end of the process. Since these are related issues, we answer both here.
>
> We first emphasize that our framework can be seen as an integration of autoregressive models with diffusion models. In the limit of no diffusion and only jumps then we arrive at a pure autoregressive model whereas if we include diffusion but initialize each new dimension with $\mathcal{N}(0, I)$ and create all dimensions at the start of the reverse generative process then we arrive at a pure diffusion model. We show in the paper how there is a fertile middle ground where we can derive benefits from both types of model with the diffusion part giving good sample quality and the autoregressive part giving trans-dimensional generation capabilities.
>
> There is a lot of flexibility in the choice for the new dimension creation distribution which we refer to as the autoregressive distribution. We chose to parameterize it with a Gaussian in the paper for simplicity but it can be parameterized with any likelihood model because the training signal for learning the autoregressive distribution is simply a maximum likelihood objective for predicting the missing part of the data given the observed part. Therefore, depending on the task, we could use more expressive alternatives such as normalizing flows or a G-SchNet architecture for molecules (Gebauer et al. 2019) which predicts new atom positions using discrete probabilities over binned distances.
>
> In our experiments we found that a reasonably simple and effective approach is to have the network predict mean and standard deviation statistics for a Gaussian distribution and then refine this with the diffusion part of the process. In our preliminary experiments we found that $t=0.1$ to $t=0$ is sufficient time to clean up any errors and produce high quality samples whilst retaining the trans-dimensional nature of the model and avoiding instabilities near $t=0$.
>
> To demonstrate this effect, we have run a sweep over when to set $\lambda_t$ to $0$ on the molecule task, see Table 2 in the additional results pdf. We have found that the setting $\lambda_{t < 0.03T} = 0$ to generate reasonable sample quality but incur some extra dimension error due to the generative process sometimes observing a lack of dimensions near $t=0$ and adding too many dimensions. We observed the same effect in the paper (L276) when setting $\lambda_t$ to be constant for all $t$. Further, the setting $\lambda_{t < 0.3T}=0$ also results in increased dimension error due to there being less opportunity for the guidance model to supervise the number of dimensions. Hence, we believe the setting $\lambda_{t<0.1T}$ to be reasonable in our case. Note these models have not trained for as long as the ones in the paper due to time constraints during the rebuttal period.
>
> We agree that the specific balance could be different for different datasets and model architectures and ultimately this is a case of hyperparameter selection. Further, if it is found that a small amount of diffusion time is insufficient to generate high quality samples, then the expressivity of the autoregressive part can be arbitrarily increased, the only downsides being an increased computational cost and code complexity.
>
> Thank you for the suggestion of perceptual metrics on the video dataset. To investigate the relative quality of frames that are added near the start of the diffusion process versus those that are added near the end, we calculated the FID for individual frames, grouped by when they are added in the generative process. Our results can be found in Table 3 in the additional results pdf. We find no systematic trend meaning there is no degradation in quality for frames added near the end of the generative process. Note that the absolute value of these FIDs may not be meaningful due to the RoboDesk dataset being far out of distribution for the Inception network used to calculate FID scores. We can visually confirm good sample quality from e.g. Figure 5 in the paper.
>
> ## References
> Niklas W. A. Gebauer, Michael Gastegger, Kristof T. Schütt. *Symmetry-adapted generation of 3d point sets for the targeted discovery of molecules.* Advances in Neural Information Processing Systems, 2019.

---

> > ### Comment · Reviewer_FTvW · 2023-08-14
> > **Response to Authors**
> >
> > Thanks a lot to the authors for their thorough responses. The rebuttal addresses my concerns and I am still of the opinion that this is a strong paper and advocate acceptance.
> >
> > In particular, the provided FID scores added in the rebuttal pdf showing that the final video frames to be generated are of comparable quality, is compelling evidence that the Gaussian approximation is sufficient in this setting. It is also true that should it prove to be problematic, a more expressive model could be used. I also appreciate the extra experiment on $\lambda_t$ showing that $0.1T$ is a reasonable choice and the impact on sample properties when this value is changed, and agree that this is a reasonable additional hyperparameter to have.

---

### Official Review · Reviewer_EZPD · 2023-07-07

**Soundness:** 4 excellent
**Presentation:** 3 good
**Contribution:** 3 good
**Rating:** 7
**Confidence:** 4

**Summary:**

This paper proposes jump diffusion, which is a novel diffusion model to handle data with varying dimensions. The proposed method is derived from a special forward process that contains a jump part that changes the dimension of the generated samples. The corresponding backward process and the learning objective are derived, with two more components that need to be learned other than the standard diffusion drift. Numerical issues are properly handled to make the mathematical model work in practice. Experiments over various application scenarios are conducted to demonstrate the flexibility and versatility of the proposed framework.

**Strengths:**

1. The proposed method targets at a very important problem. It is a great contribution to invent theoretically sound diffusion models to deal with data with varying dimensions. This kind of problem frequently happens in real-world applications.

2. The motivation is strong, the idea is reasonable, and the mathematical derivations makes the idea a solid framework.

3. Applications on molecule and video generation showcases the potential of the proposed framework in reality.

**Weaknesses:**

1. Although I do think the proposed framework is interesting, generation with varying dimensionality can be addressed easily by

(1) learning a distribution over the dimension numbers and sample dimension number from the learned distribution (2) sampling initial random noises according to the sampled dimension number at the beginning of the generation and pad the unused dimensions with 0.

This simple method has little modification over existing diffusion models, and the only extra efforts one need to do is to add paddings during training.

It would be very nice if I can hear comments from the authors on this simple baseline. I do appreciate the theoretical contribution of this paper, and it would be better if the authors can compare with this simple baseline .

2.In L84, should the summation be taken over m<n? Because there is only deletion in the forward process.

**Questions:**

I have one question on the proposed diffusion model and I would appreciate answers from the authors.

In Algorithm Box 1, what is the initial distribution? Is it a random standard Gaussian noise with only 1 dimension? If so, how do we decide which dimension is the initial dimension? For example, if we have x = (3, x_1, x_2, x_3), what is the initial random noise? (1, N(0, I), 0, 0), (1, 0, N(0, I), 0), or (1, 0, 0,  N(0, I) ? Or are they uniformly sampled?

Moreover, if the initial number of dimension is always 1, how do we guarantee that in setting the forward process? I feel like there must be some constraints on the scheduling of rate function $\lambda(t)$.

Maybe I missed some parts of the derivation in the paper.

**Limitations:**

The limitations and broader impacts are properly discussed.

---

> ### Author Rebuttal · Authors · 2023-08-09
>
> We thank the reviewer for their engagement with our proposed methodology and thoughtful questions. We are grateful that our work is considered to be a solid framework to tackle a very important problem. We answer questions and respond to comments below.
>
> > *It would be very nice if I can hear comments from the authors on this simple baseline.*
>
> We focussed in this work on training flexible unconditional models that jointly model all the relevant aspects of the data: the state and the dimension. This is in contrast to the mentioned baseline that models the state and dimension separately with two separate models. We showcase the benefit of the joint modelling approach by using the task of diffusion guidance where an unconditional model is first trained and then, at test time, different researchers with different goals can condition this model on their task of interest using diffusion guidance, for example Weiss et al.  2023 and Crowson 2021. This has the advantage that, should the unconditional model be powerful enough, any conditional generation task can be easily accomplished with a limited computational budget because no re-training is required, the only thing needed is a guidance model. In the case of the baseline that you mention, complete knowledge of the final generation task is needed before training because the dimension prediction model and score model both need to be conditioned on this task information. Therefore this type of approach is not applicable to our intended application where the end user does not have the resources for a complete re-training of the models for the tasks they are interested in.
>
> Motivated by this deficiency of models that treat dimensions and state values separately, we developed a generative model that jointly models dimension and state meaning it is powerful enough to be used as an unconditional model that can then be guided at test time for different generation tasks. In this case where the end user is guiding the model on their task of interest, the fixed dimensional model fails to capture the correct dimension information whereas our method can produce more accurate dimensions statistics for each desired task as we show in Table 3 in the paper.
>
> > *In L84, should the summation be taken over m<n? Because there is only deletion in the forward process.*
>
> Thank you for pointing this out, you are correct that for our forward process $K_t( m , \mathbf{y} | \mathbf{X})$ will be zero for $m \geq n$ due to the forward process only deleting dimensions. For the backward process we would instead have $K_t(m , \mathbf{y} | \mathbf{X})$ being $0$ for $m \leq n$. On L84 we introduce jump diffusions for the first time and so try to keep the formulation general enough to cover both the forward and backward cases. However, given that this introduction is in the forward process section this could lead to confusion and we will make this clearer in a revision to the paper.
>
> > *In Algorithm Box 1, what is the initial distribution?*
>
> The initial distribution is indeed a single dimension of standard Gaussian noise. However, we do not need to pre-determine which dimension in the final generated datapoint it corresponds to. We treat it genuinely as only a single dimension i.e. the initial sample is $(1, \mathcal{N}(0, I))$ with no baked in knowledge of the length of the final datapoint to create nor the initial dimension’s place in that final datapoint. Then, when a new dimension is added it could be added to the left or to the right becoming $(2, z, \mathcal{N}(0, I) )$ or $(2, \mathcal{N}(0, I), z)$ where $z$ is the new value to be added. This process is repeated until we reach the terminal time $t=0$ of the reverse generative process. Therefore, that initial dimension could correspond to any of the dimensions in the final generated datapoint depending on where new dimensions are inserted. Furthermore, the final number of dimensions also varies depending on how many jumps occurred during generation. In your example e.g. $(1, 0, \mathcal{N}(0, I), 0)$ this assumes the final dimension is 3 but we make no such assumption in our method.
>
> > *Moreover, if the initial number of dimension is always 1, how do we guarantee that in setting the forward process? I feel like there must be some constraints on the scheduling of rate function*
>
> This is a good point and indeed this constraint influences the choice of forward process rate function, $\lambda_t$. We set it such that, with high probability, the number of dimensions at the end of the forward process is $1$. This is the same idea as in fixed dimension diffusion models where the noising process is such that the corrupted data at the end of the noising process is approximately distributed according to $\mathcal{N}( \mathbf{x}; 0, I)$. Here, we have a noising process such that the corrupted data at the end of the noising process is approximately distributed according to $\mathbb{I} \\{n = 1\\} \mathcal{N}(\mathbf{x}; 0, I)$. In practice we achieve this by setting $\lambda_t$ large enough such that there is a high probability of enough jumps occurring to remove all but one of the dimensions in the datapoint. Further, we set $\lambda_t$ to $0$ when the number of dimensions is equal to $1$ so that we can’t delete down to $0$ dimensions.
>
> ## References
>
> Tomer Weiss, Luca Cosmo, Eduardo Mayo Yanes, Sabyasachi Chakraborty, Alex M Bronstein,and Renana Gershoni-Poranne. *Guided diffusion for inverse molecular design.* ChemrXiv, 2023
>
> Katherine Crowson. *Clip guided diffusion.* Web Demo, 2021

---

> > ### Comment · Reviewer_EZPD · 2023-08-14
> > **Thank you**
> >
> > Thank you for your comments comparing between your model and my simple baseline! The joint modeling of both dimensionality and state is indeed important. I agree. Other replies also clearly answer my questions. Although the initial distribution in the backward and forward process has a little bit mismatch, I think the way the authors handle it can mitigate the influence.
> >
> > Thank you for the reply again!

---

### Author Rebuttal · Authors · 2023-08-09

We would like to thank all the reviewers for their analysis of our paper and very helpful reviews. We were pleased that reviewers considered our work to be a novel and theoretically sound method for tackling the important problem of modeling data with varying dimensionality.

We address comments made by each reviewer in individual responses and attach an additional results pdf here.

---

### Decision · Program_Chairs · 2023-09-21

**Decision:**

Accept (spotlight)

**Comment:**

Authors  propose in this paper a new diffusion model based on jump diffusion processes that handles varying dimensions. All reviews agreed on the novelty and the soundness of the work. The proposed methods was validated on molecules, videos and robotics applications.